## OPEN
# Mitochondrial electron transport chain is necessary for NLRP3 inflammasome activation

Leah K. Billingham[1], Joshua S. Stoolman [ID][1], Karthik Vasan[1], Arianne E. Rodriguez[1], Taylor A. Poor[1], Marten Szibor [ID][2,3,4], Howard T. Jacobs [ID][2,4], Colleen R. Reczek[1], Aida Rashidi[5], Peng Zhang[5], Jason Miska[5] and Navdeep S. Chandel[1,6] ✉

**The NLRP3 inflammasome is linked to sterile and pathogen-dependent inflammation, and its dysregulation underlies many chronic diseases. Mitochondria have been implicated as regulators of the NLRP3 inflammasome through several mechanisms including generation of mitochondrial reactive oxygen species (ROS). Here, we report that mitochondrial electron transport chain (ETC) complex I, II, III and V inhibitors all prevent NLRP3 inflammasome activation. Ectopic expression of *Saccharomyces cerevisiae* NADH dehydrogenase (NDI1) or *Ciona intestinalis* alternative oxidase, which can complement the functional loss of mitochondrial complex I or III, respectively, without generation of ROS, rescued NLRP3 inflammasome activation in the absence of endogenous mitochondrial complex I or complex III function. Metabolomics revealed phosphocreatine (PCr), which can sustain ATP levels, as a common metabolite that is diminished by mitochondrial ETC inhibitors. PCr depletion decreased ATP levels and NLRP3 inflammasome activation. Thus, the mitochondrial ETC sustains NLRP3 inflammasome activation through PCr-dependent generation of ATP, but via a ROS-independent mechanism.**

The NLRP3 inflammasome is activated by viral and bacterial infections as well as noninfectious stimuli including uric acid crystals, asbestos, imiquimod, nigericin and ATP signaling through the P2X7 receptor. Aberrant NLRP3 activation is linked to development of type II diabetes, atherosclerosis, autoimmunity and neurodegenerative diseases. NLRP3 inflammasome activation occurs in macrophages primed with LPS and subsequently exposed to a second stimuli, which can be dependent on (for example, ATP, nigericin) or independent of (imiquimod) potassium (K⁺) efflux[1–3]. The NLRP3 inflammasome consists of the receptor protein NLRP3, the adapter protein ASC and the cysteine protease pro-caspase-1 (ref. [4]). NLRP3 contains a pyrin domain (PYD), a NACHT domain and a leucine-rich repeat domain. The NLRP3 inflammasome requires ATP hydrolysis at the NACHT domain to assume an active conformation[5]. ASC contains a PYD domain and a caspase-recruitment domain (CARD). Upon activation of this inflammasome, NLRP3 and ASC oligomerize through PYD–PYD interactions, forming filamentous aggregates known as ASC specks. ASC binds, in turn, to pro-caspase-1 through CARD–CARD domain interactions. The clustering of pro-caspase-1 at ASC specks results in autocleavage into its active form, caspase-1. Caspase-1 then processes pro-IL-1β, IL-18 and gasdermin D, resulting in secretion of IL-1β and IL-18 (refs. [1,6–8]).

Several studies have linked mitochondrial ETC function to NLRP3 activation[3,9,10]. Pharmacologic studies have linked the mitochondrial ETC to NLRP3 inflammasome activation through ROS[11,12]. Some studies suggest that ETC inhibition can either increase or decrease NLRP3 activation, while others point to NLRP3 inflammasome activation being independent of ETC function[2,3,9,10]. Transcriptional, translational and metabolic changes occur rapidly post LPS treatment[13–15]. Previous studies have demonstrated that the tricarboxylic acid cycle intermediate succinate accumulates during lipopolysaccharide (LPS) stimulation of bone marrow-derived macrophages (BMDMs) in vitro[16–18]. Furthermore, dimethyl malonate (DMM), an inhibitor of mitochondrial complex II (succinate dehydrogenase (SDH)) or loss of mitochondrial complex II subunit SDHB, which prevents succinate oxidation, attenuates LPS induction of *Il1b* mRNA and IL-1β protein levels at 24–48 h in vitro[19]. Using a combination of several ETC inhibitors as well as genetic perturbations that modify ETC function, we directly tested whether ATP-dependent NLRP3 inflammasome activation depends on ETC function.

## Results

**Mitochondrial complex II is required for NLRP3 inflammasome activation.** Mitochondrial complex II is the best described ETC complex linked to LPS-dependent induction of IL-1β. Thus, we initiated our studies by testing whether ETC inhibition at mitochondrial complex II decreases LPS priming and/or NLRP3 inflammasome activation early in the LPS response. In our studies, BMDMs were primed with LPS for 5.5 h. Subsequently, we measured intracellular cleaved caspase 1 (p20 fragment) protein levels at 10 min and IL-1β protein in the supernatant at 30 min after extracellular ATP addition (activation of NLRP3 inflammasome). The p20 fragment of intracellular caspase-1 is the final inactive product of caspase-1 activation[20]. Mitochondrial ETC inhibitors in our study were administered 30 min before treatment with LPS, that is, the priming step (Extended Data Fig. 1a).

DMM treatment decreased oxygen consumption rates (OCR) in primary mouse BMDMs (Extended Data Fig. 1b,c). Furthermore, LPS treatment induced significant changes in certain metabolites

[1]Department of Medicine, Northwestern University Feinberg School of Medicine, Chicago, IL, USA. [2]Faculty of Medicine and Health Technology, Tampere University, Tampere, Finland. [3]Department of Cardiothoracic Surgery, Center for Sepsis Control and Care (CSCC), Jena University Hospital, Jena, Germany. [4]Department of Environment and Genetics, La Trobe University, Melbourne, Victoria, Australia. [5]Department of Neurological Surgery, Lou and Jean Malnati Brain Tumor Institute, Northwestern University Feinberg School of Medicine, Chicago, IL, USA. [6]Department of Biochemistry and Molecular Genetics, Northwestern University Feinberg School of Medicine, Chicago, IL, USA. ✉e-mail: nav@northwestern.edu

that were abrogated by DMM (Extended Data Fig. 1d). DMM also increased succinate levels without altering the NAD⁺/NADH ratio (Extended Data Fig. 1e,f). Despite these metabolic changes, DMM did not decrease LPS induction of *Il1b*, *Tnf* or *Il10* mRNA (Extended Data Fig. 2a–c). However, DMM did attenuate release of secreted IL-1β protein in BMDMs treated with LPS and extracellular ATP without altering intracellular pro-IL-1β levels during priming (Extended Data Fig. 2d,e). DMM did not alter LPS induction of secreted TNFα protein (Extended Data Fig. 2f). DMM decreased the level of intracellular cleaved caspase-1 protein levels without altering intracellular pro-caspase-1 protein following treatment with LPS plus extracellular ATP (Extended Data Fig. 2g,h). These data demonstrate that mitochondrial complex II is necessary for caspase-1 activation and IL-1β protein production but not for early LPS induction of *Il1b* mRNA expression.

**Mitochondrial complex I is required for NLRP3 inflammasome activation.** Mitochondrial ETC complexes I and II transfer electrons from NADH and succinate, respectively, to ubiquinone (CoQ), reducing it to ubiquinol (CoQH₂) (Fig. 1a). During the inflammatory response, succinate levels increase, and the CoQ pool can become reduced[19]. This results in reverse electron transport (RET) from CoQH₂ to NAD⁺ at mitochondrial complex I (Fig. 1a)—a process that generates high levels of superoxide ($O_2^{\bullet-}$)[21]. RET-generated $O_2^{\bullet-}$ has been implicated in perpetuating the inflammatory response after 24 h of LPS administration alone in vitro[19]. Mitochondrial complex I inhibitors, such as rotenone and piericidin A, block $O_2^{\bullet-}$ generation by RET[21] and attenuate LPS induction of *Il1b* mRNA[22]. We investigated the necessity of mitochondrial complex I function for NLRP3 activation by using piericidin A, which decreases OCR and the NAD⁺/NADH ratio (Extended Data Fig. 3a–c). Piericidin A abolished LPS-induced metabolite changes, including an increase in succinate (Extended Data Fig. 3d,e). Piericidin A did not diminish LPS induction of *Il1b*, *Tnf*, or *Il10* mRNA after 4 h (Extended Data Fig. 4a–c). Furthermore, piericidin A did not diminish pro-IL-1β protein or pro-caspase-1 protein levels (Extended Data Fig. 4d,e). However, piericidin A did decrease secreted IL-1β protein levels and intracellular cleaved caspase-1 protein levels upon LPS plus ATP stimulation (Extended Data Fig. 4f,g). Piericidin A also decreased secreted IL-1β protein levels in BMDMs treated with the NLRP3 inflammasome activator nigericin (Extended Data Fig. 4h). Importantly, Piericidin A did not diminish LPS induction of secreted TNFα protein levels (Extended Data Fig. 4i) These data suggest that mitochondrial complex I function is not required for the LPS induction of *Il1b* mRNA but is for caspase-1 activation and production of secreted IL-1β protein.

**Reverse electron transport is not required for NLRP3 inflammasome activation.** To test the specificity of piericidin A as a mitochondrial complex I inhibitor in our studies, we used BMDMs that express *Saccharomyces cerevisiae* NADH dehydrogenase (NDI1)[23]. Mammalian mitochondrial complex I transfers electrons from NADH to CoQ while pumping protons across the inner mitochondrial membrane. By contrast, NDI1 transfers electrons from NADH to CoQ but does not pump protons and is unable by itself to generate RET-induced $O_2^{\bullet-}$ (refs. [23,24]). Importantly, NDI1 is resistant to piericidin A and other mitochondrial complex I inhibitors[23,25] (Fig. 1a). Thus, treating NDI1-expressing cells with piericidin A allows for NADH oxidation to support downstream electron flow to mitochondrial complexes III, IV and molecular oxygen (respiration), but not complex I-dependent proton pumping or RET-induced $O_2^{\bullet-}$ production. Recently, we generated a transgenic mouse line that contains a lox-stop-lox-NDI1 targeting construct in the Rosa26 locus[26]. To generate BMDMs that express NDI1, we crossed NDI1ˡˢˡ/ʷᵗ mice with Vav-iCre mice, resulting in mice that express NDI1 in hematopoietic lineages including monocytes, here denoted as NDI1. Mice containing Vav-iCre without the lox-stop-lox-NDI1 are denoted as WT.

As expected, BMDMs generated from NDI1 mice expressed *NDI1* mRNA (Fig. 1b). NDI1-expressing BMDMs do not exhibit changes in OCR coupled to ATP production (Fig. 1c). Piericidin A decreased OCR and the NAD⁺/NADH ratio in WT but not in NDI1-expressing BMDMs (Fig. 1d,e) and decreased RET-generated H₂O₂ to a similar extent in both WT and NDI1-expressing BMDMs (Fig. 1f). These results indicate that any rescue effects of piericidin A observed in NDI1-expressing cells are independent of RET-generated H₂O₂ and are due to restoration of NADH oxidation. To understand more broadly any changes in metabolism conferred by the presence of NDI1, we performed metabolomics on BMDMs from WT and NDI1 mice. Metabolites significantly altered in LPS-stimulated WT BMDMs in the presence of piericidin A remained unchanged in NDI1 BMDMs (Fig. 1g). Of note, succinate levels following LPS stimulation were maintained in NDI1 BMDMs in the presence of piericidin A (Fig. 1h).

Next, we determined whether piericidin A inhibition of NLRP3 inflammasome activation was due to inhibition of mitochondrial complex I. NDI1-expressing mice do not have altered IL-1β protein in vivo 2 h post LPS administration (Fig. 2a), indicating that expression of NDI1 itself is not inflammatory. Furthermore, principal component analysis of transcriptional patterns, based on RNA-seq, demonstrated that, both at baseline and in response to LPS, WT and NDI1 BMDMs are largely similar (Extended Data Fig. 5a). However, piericidin A-induced changes in the transcriptional response to LPS treatment were largely abolished by the expression of NDI1 (Extended Data Fig. 5b). Piericidin A attenuated the production of secreted IL-1β protein in WT but not NDI1 BMDMs (Fig. 2b). In contrast, DMM decreased IL-1β protein in both WT and NDI BMDMs, consistent with the proposition that forward electron transport is required to activate the inflammasome (Fig. 2c). Piericidin A also decreased intracellular cleaved caspase-1 protein levels in WT but not NDI1 BMDMs (Fig. 2d). Piericidin A did not decrease intracellular

**Fig. 1 | NDI1 expression confers resistance to mitochondrial complex I inhibitor piericidin A. a,** Schematic of the mitochondrial electron transport chain in WT (top) and NDI1-expressing (bottom) BMDMs during LPS stimulation. Piericidin A inhibition of mitochondrial complex I on electron flow is rescued by NDI1 expression. IMM, inner mitochondrial membrane; RET, reverse electron transport. **b,** *NDI1* mRNA levels (ΔΔC$_t$) in WT and NDI1 BMDMs (*n* = 5 WT; *n* = 12 NDI1). **c,** Coupled OCR in WT and NDI1 BMDMs (*n* = 9 for each genotype). **d,** Basal OCR in WT and NDI1 BMDMs after 1 h treatment with 100 nM or 500 nM piericidin A (*n* = 13 vehicle for each genotype; *n* = 9 100 nM piericidin A for each genotype; *n* = 4 500 nM piericidin A for each genotype). **e,** NAD⁺/NADH ratio in WT and NDI1 BMDMs after 4 h treatment with or without LPS (100 ng ml⁻¹) in the presence or absence of piericidin A (500 nM) (*n* = 3 WT LPS + piercidin A; *n* = 4 all other treatments). **f,** Rate of H₂O₂ production in WT and NDI1 BMDMs in the presence of succinate (500 μM) with or without piericidin A treatment (500 nM) (*n* = 9). **g,** Heatmap of significantly altered metabolites in WT and NDI1 BMDMs treated with LPS (100 ng ml⁻¹) alone, piericidin A alone (500 nM) or both LPS and piericidin A for 4 h. The relative abundance of each metabolite is depicted as *z* score across rows (red, high; blue, low) (*n* = 5 for all treatments). **h,** Arbitrary units of succinate in WT and NDI1 BMDMs with or without LPS (100 ng ml⁻¹) and piericidin A (500 nM) for 4 h (*n* = 5 for all treatments). Data are mean ± s.e.m. *$P < 0.05$, two-tailed *t*-test (**b**, $P = 0.0001$), ANOVA with Tukey's post hoc test for multiple comparisons (**d**, *$P = 0.0008$ WT UT/100 nM, *$P = 0.0047$ WT UT/500 nM; **e**, *$P = 0.006$ WT UT/WT piericidin A, *$P = 0.0034$ WT LPS/WT LPS + piericidin A; **f**, *$P = 0.0465$ WT succinate/WT succinate + piericidin A, *$P = 0.0493$ NDI1 succinate/NDI1 succinate + piericidin A; **h**, *$P = 0.0478$), or ANOVA with Fisher's LSD (**g**). *n* indicates number of individual mice. Parts of this figure were created with BioRender.com.

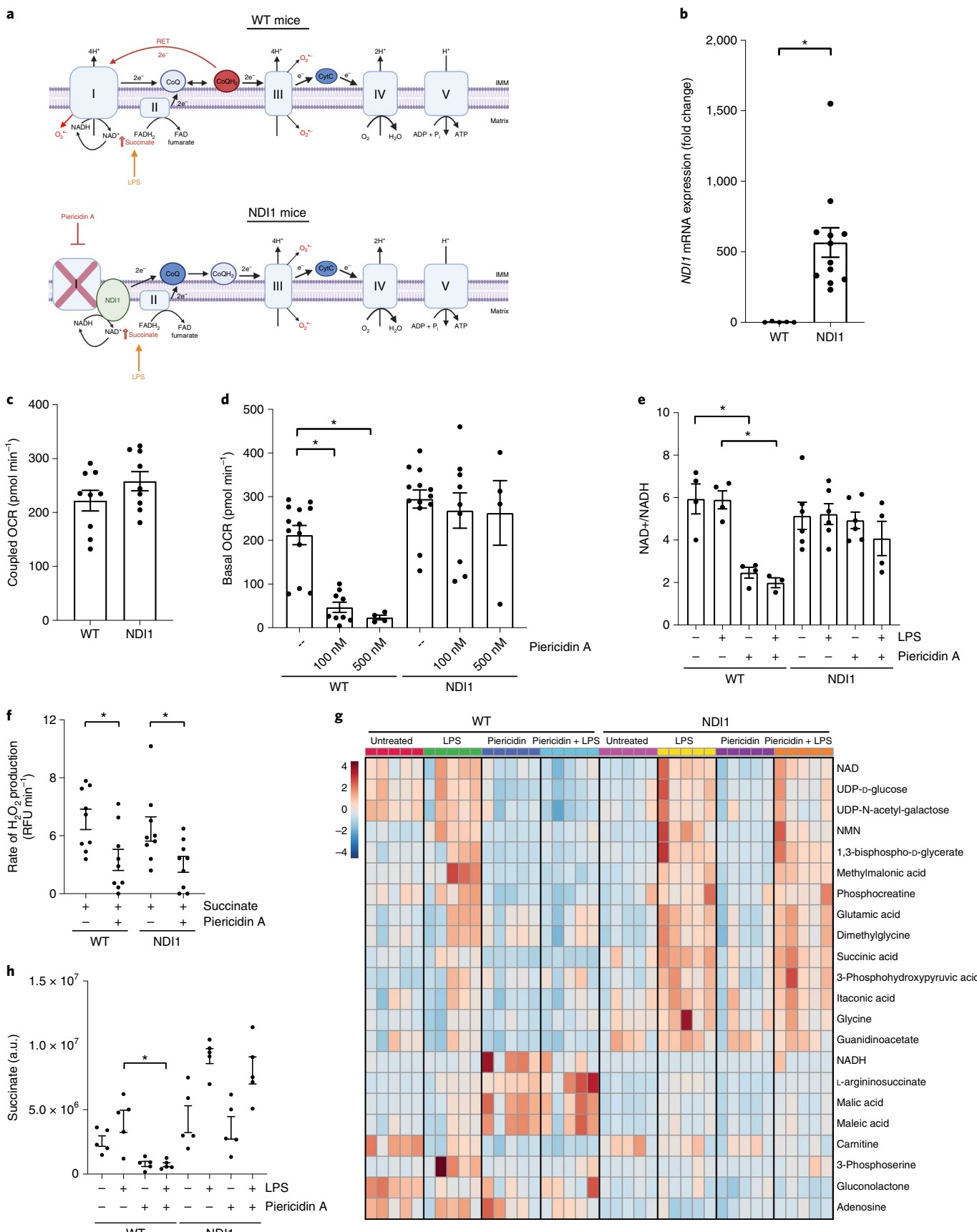

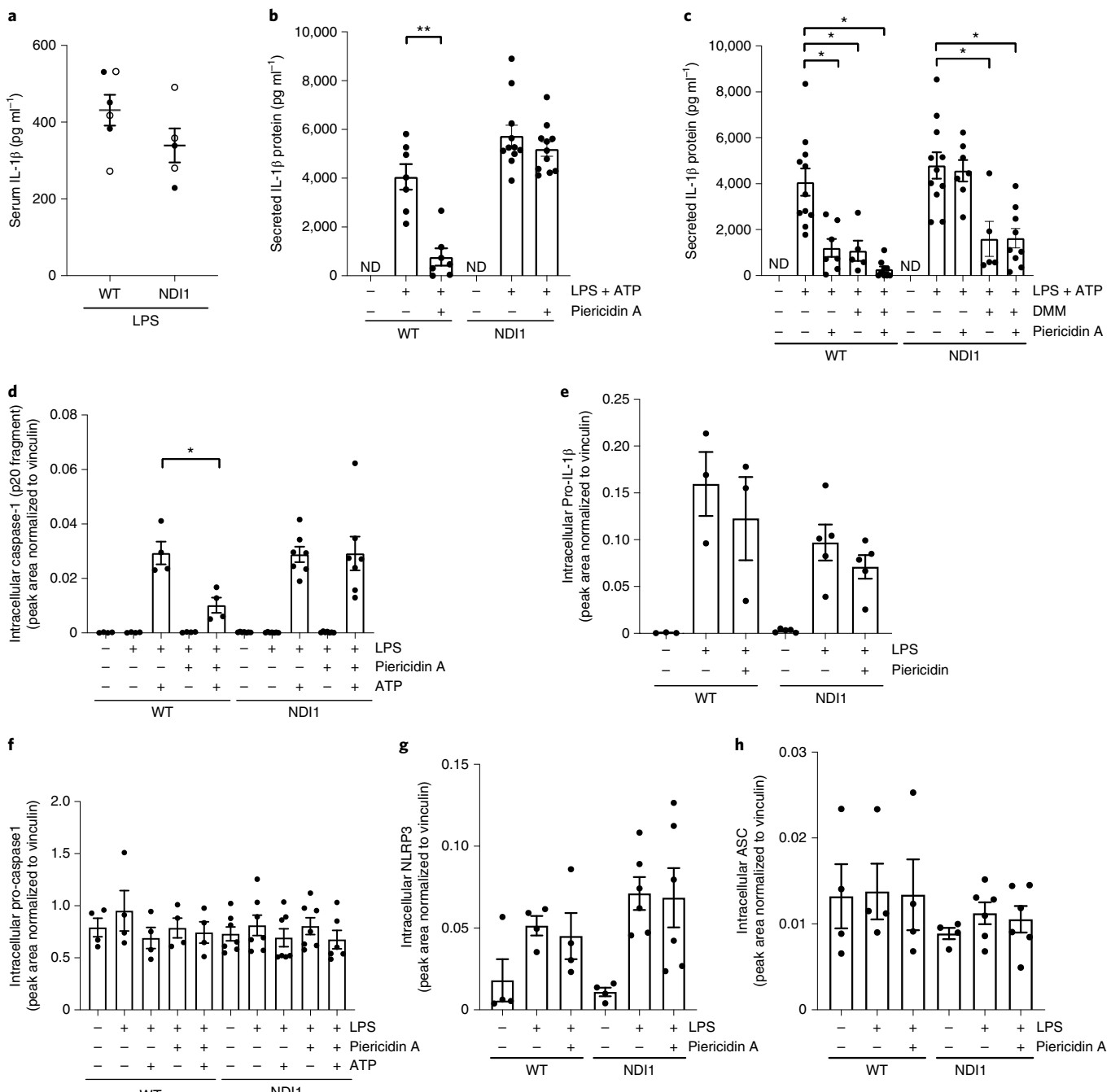

**Fig. 2 | Reverse electron transport is not required for NLRP3 inflammasome activation. a**, IL-1β protein levels in the serum of WT and NDI1 mice 2 h post i.p. injection of 100 mg kg⁻¹ crude LPS (*n* = 6 WT; *n* = 5 NDI1; symbols indicate independent experiments). **b**, IL-1β protein levels in cell culture supernatant of WT and NDI1 BMDMs treated with LPS (100 ng ml⁻¹) and ATP (5 mM) with or without piericidin A (100 nM) (*n* = 7 WT; *n* = 11 NDI1). **c**, IL-1β protein levels in cell culture supernatant of WT and NDI1 BMDMs treated with LPS (100 ng ml⁻¹) and ATP (5 mM), with or without piericidin A (100 nM) and/or DMM (10 mM) (*n* = 10 LPS + DMM + piericidin A for each genotype; *n* = 11 LPS for each genotype; *n* = 7 LPS + piericidin A for each genotype; *n* = 5 LPS + DMM for each genotype). **d**, Intracellular Caspase-1 (p20 fragment) protein expression in cell lysates from WT and NDI1 BMDMs treated as in **c** (*n* = 4 WT; *n* = 7 NDI1). **e**, Pro-IL-1β protein expression in cell lysates from WT and NDI1 BMDMs treated with LPS (100 ng ml⁻¹) with or without piericidin A (100 nM) (*n* = 3 WT; *n* = 5 NDI1). **f**, Pro-Caspase-1 protein expression in cell lysates from WT and NDI1 BMDMs treated with LPS (100 ng ml⁻¹) and ATP (5 mM) with or without piericidin A (100 nM) (*n* = 4 WT; *n* = 7 NDI1). **g**, NLRP3 protein expression in cell lysates from WT and NDI1 BMDMs treated with LPS (100 ng ml⁻¹) (*n* = 4 WT all treatments, NDI1 UT; *n* = 6 NDI1 + LPS, NDI1 + LPS + piericidin A). **h**, ASC protein expression in cell lysates from WT and NDI1 BMDMs treated as in **e** (*n* = 4 WT all treatments, NDI1 UT; *n* = 6 NDI1 + LPS, NDI1 + LPS + piericidin A). Data are means ± s.e.m. *P < 0.05, ANOVA with Tukey's post hoc test for multiple comparisons (**b**, *P < 0.0001; **c**, *P = 0.0038 WT LPS + ATP/ WT LPS + ATP + Piericidin A, *P = 0.0083 WT LPS + ATP/WT LPS + ATP + DMM, *P < 0.0001 WT LPS + ATP/WT LPS + ATP + DMM + Piericidin A, *P = 0.0037 NDI1 LPS + ATP/NDI1 LPS + ATP + DMM, *P = 0.0003 NDI1 LPS + ATP/NDI1 LPS + ATP + DMM + Piericidin A; **d**, *P = 0.0129). ND, not detected.

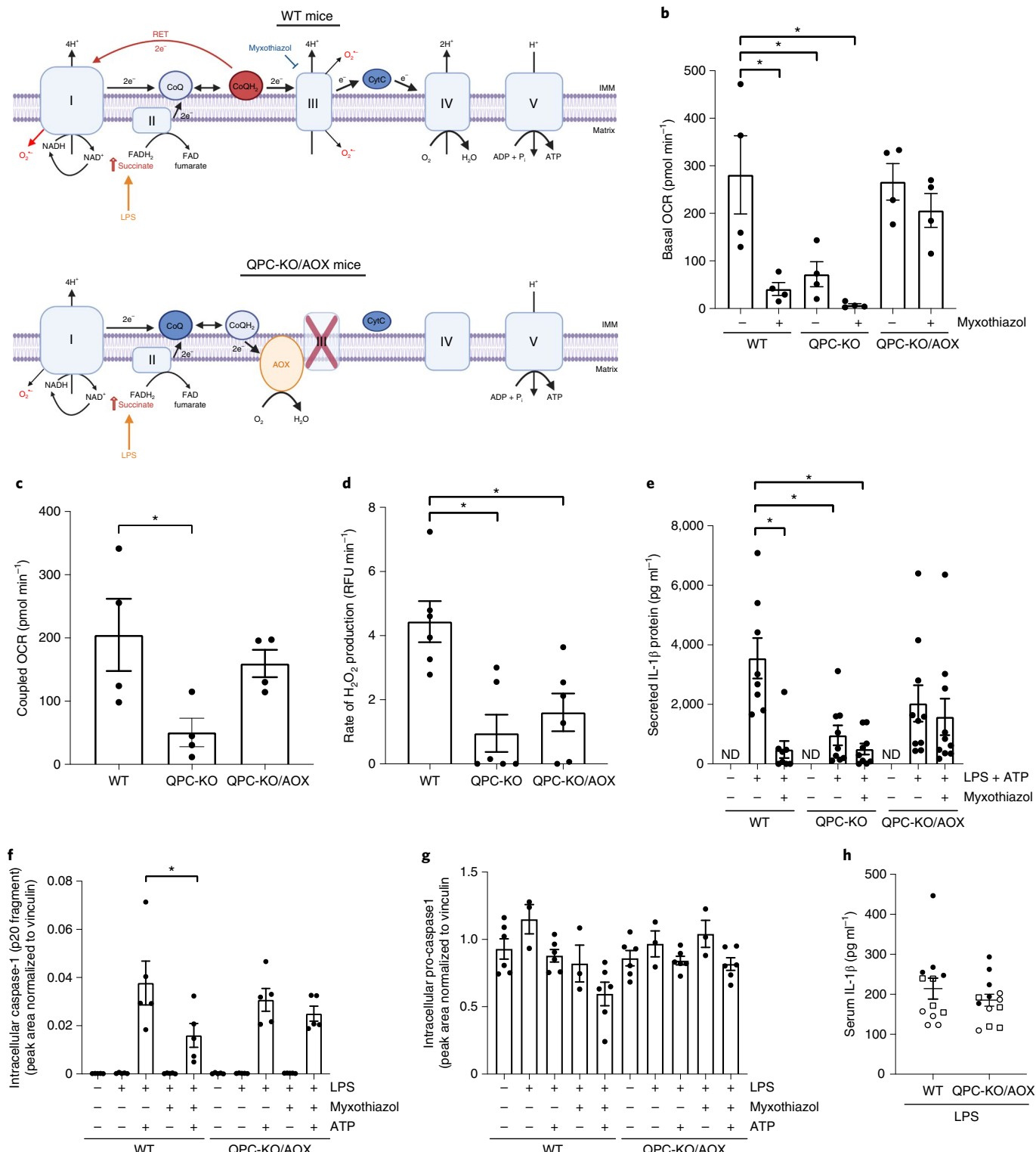

pro-IL-1β, pro-caspase-1 protein, NLRP3 or ASC protein levels under all conditions (Fig. 2e–h). As NDI1-expressing BMDMs in the presence of piericidin A cannot perform RET, these results imply that mitochondrial complex I is necessary for NLRP3 activation due to NADH oxidation, which supports forward electron transport to other downstream ETC complexes.

**Mitochondrial production of $H_2O_2$ is not required for NLRP3 inflammasome activation.** Mitochondrial complexes I and II

donate electrons to CoQ, which transfers electrons to mitochondrial complex III. Subsequently, mitochondrial complex III transfers electrons from CoQH2 to cytochrome *c*, which then donates electrons to cytochrome *c* oxidase (complex IV) and ultimately to molecular oxygen (Fig. 3a). Mitochondrial complex III also pumps protons and is one of the major sites of $O_2^{\bullet-}$ production. To determine whether mitochondrial complex III is required for inflammasome activation, we treated BMDMs with the inhibitor myxothiazol, which diminished OCR as expected (Extended Data Fig. 6a). As with DMM and

**Fig. 3 | Mitochondrial-generated $H_2O_2$ production is not required for NLRP3 inflammasome activation. a**, Schematic of the mitochondrial ETC in WT (top) and QPC-KO/AOX BMDMs (bottom). In WT BMDMs, myxothiazol inhibition of complex III blocks onward electron flow to oxygen. In QPC-KO/ AOX BMDMs, AOX accepts electrons from reduced CoQ, maintaining electron flow but without generating $O_2^{\bullet}$. Mitochondrial complex I pumps proton to generate a proton motive force to sustain ATP levels in QPC-KO/AOX. **b**, OCR in WT, QPC-KO and QPC-KO/AOX BMDMs with or without 100 nM myxothiazol ($n = 4$ for each genotype). **c**, Coupled OCR in WT and QPC-KO/AOX BMDMs after 1 h treatment with 100 nM myxothiazol ($n = 4$ for each genotype). **d**, Rate of $H_2O_2$ production in WT, QPC-KO and QPC-KO/AOX BMDMs in the presence of 500 µM succinate ($n = 6$ for each genotype). **e**, IL-1β protein levels in cell culture supernatant of WT, QPC-KO and QPC-KO/AOX BMDMs treated with LPS (100 ng ml⁻¹) and ATP (5 mM), with or without myxothiazol (100 nM) ($N = 8$ WT both treatments; $n = 9$ QPC-KO both treatments; $n = 10$ QPC-KO/AOX both treatments). **f**, Pro-caspase-1 protein expression in cell lysates from WT and QPC-KO/AOX BMDMs treated with LPS (100 ng ml⁻¹) and ATP (5 mM), with or without myxothiazol (100 nM) ($n = 6$ WT UT, WT LPS + ATP, WT LPS + myxothiazol + ATP, QPC-KO/AOX UT, QPC-KO/AOX LPS + ATP, QPC-KO/AOX + myxothiazol + ATP; $n = 3$ WT LPS, WT LPS + myxothiazol, QPC-KO/AOX LPS, QPC-KO/AOX LPS + myxothiazol). **g**, Intracellular caspase-1 (p20 fragment) protein expression in cell lysates from WT and QPC-KO/AOX BMDMs treated with as in **f** ($n = 5$ for each treatment and genotype). **h**, IL-1β protein levels in the serum of WT and QPC-KO/AOX mice 2 h post i.p. injection of 50 mg kg⁻¹ crude LPS ($n = 12$ WT; $n = 13$ QPC-KO/AOX; symbols indicate distinct independent experiments). Data are means ± s.e.m. *$P < 0.05$, one-way ANOVA with Tukey's post hoc test for multiple comparisons (**b**, *$P = 0.0077$ WT UT/WT Myxothiazol, *$P = 0.0231$ WT UT/QPC-KO UT, $P = 0.0023$ WT UT/QPC-KO Myxothiazol; **c**, *$P = 0.042$, **d**, *$P = 0.0027$ WT/QPC-KO, *$P = 0.0124$ WT/QPC-KO/AOX; **e**, *$P = 0.0022$ WT LPS + ATP/WT LPS + ATP + Myxothiazol, *$P = 0.0108$ WT LPS + ATP/QPC-KO LPS + ATP, *$P = 0.0017$ WT LPS + ATP + Myxothiazol/ QPC-KO LPS + ATP + Myxothiazol; **f**, *$P = 0.0063$ WT LPS + ATP/WT LPS + Myxothiazol+ATP). Parts of this figure were created with BioRender.com.

piericidin A treatment, myxothiazol did not affect the LPS induction of *Il1b*, *Tnf* or *Il10* mRNA expression at 4 h (Extended Data Fig. 6b–d). However, myxothiazol did decrease secreted IL-1β protein levels from BMDMs treated with LPS plus extracellular ATP without altering pro-IL-1β protein levels (Extended Data Fig. 6e,f). NDI1 expression in BMDMs did not rescue intracellular cleaved caspase-1 or secreted IL-1β protein levels when cells were treated with both piericidin A and myxothiazol, indicating that NDI1 expression only rescues the effects of piericidin A (Extended Data Fig. 6g–i). These results suggest that mitochondrial complex III is also required for NLRP3 inflammasome activation.

To distinguish the role of mitochondrial complex III in electron transport from its ability to generate $O_2^{\bullet-}$ and proton pump, we adopted an equivalent approach to the use of cells from NDI1 mice by using BMDMs from mice expressing the *Ciona intestinalis* alternative oxidase (AOX). AOX transfers electrons from $CoQH_2$ directly to oxygen without proton pumping or $O_2^{\bullet-}$ production[27–29]. In the absence of mitochondrial complex III function, AOX allows mitochondrial complexes I and II to transfer electrons to CoQ, thus regenerating $NAD^+$ and FAD without generation of $O_2^{\bullet-}$ at complex III (ref. [30]). Moreover, ectopic AOX expression in mammalian cells has been shown to prevent overreduction of the CoQ pool to diminish RET-induced $O_2^{\bullet-}$ (refs. [21,31]) (Fig. 3a). In the absence of mitochondrial complex III, AOX expressing cells are able to generate the mitochondrial complex I-dependent proton motive force needed for mitochondrial ATP production, that is, coupled respiration (Fig. 3a)[32].

To genetically abrogate mitochondrial complex III- and RET-generated $O_2^{\bullet-}$, we generated mice that conditionally express AOX[33] in myeloid cells lacking the mitochondrial complex III subunit VII (QPC)[34] (QPCfl/fl; AOXlsl/ Lyz2-Cre, here denoted QPC-KO/ AOX). We also used QPC-KO (QPCfl/fl Lyz2-Cre) mice. Control mice were heterozygous for QPC in myeloid cells without AOX (here denoted WT). QPC-KO have diminished OCR compared with WT and QPC-KO/AOX (Fig. 3b). Myxothiazol inhibited OCR in BMDMs from WT mice but not from QPC-KO/AOX BMDMs, confirming the specificity of myxothiazol as a mitochondrial complex III inhibitor (Fig. 3b). Importantly, the rate of coupled respiration in WT and QPC-KO/AOX BMDMs was similar, indicating that QPC-KO/AOX BMDMs can generate mitochondrial ATP (Fig. 3c). As expected, both QPC-KO and QPC-KO/AOX BMDMs produced less $H_2O_2$ than WT BMDMs (Fig. 3d). QPC-KO BMDMs exhibited a decrease in secreted IL-1β protein levels compared with WT (Fig. 3e). QPC-KO/AOX BMDMs did not exhibit significant differences in secreted IL-1β or intracellular cleaved caspase-1 protein levels compared with WT BMDMs (Fig. 3e,f), and these were unaffected by myxothiazol (Fig. 3e,f). Importantly, intracellular pro-caspase 1 protein levels were similar in WT and QPC-KO/AOX BMDMs (Fig. 3g). Finally, LPS induced similar levels of secreted IL-1β protein in serum from WT and QPC-KO/AOX mice (Fig. 3h). Collectively, these data indicate that $O_2^{\bullet-}$ generated at mitochondrial complex III or by RET is not required for NLRP3 inflammasome activation in vitro or in vivo. Nevertheless, NLRP3 inflammasome activation requires forward electron flow through the ETC, producing ATP

**Fig. 4 | Mitochondrial-generated PCr during priming supports NLRP3 inflammasome activation. a**, PCr levels (a.u.) in cells treated with or without cyclocreatine (10 mM), with or without LPS for 4 h ($n = 5$ for each treatment). **b**, Intracellular ATP levels (a.u.) in cells treated for 4 h with piericidin A (100 nM) or cyclocreatine (10 µM), with or without LPS (100 ng ml⁻¹) or nigericin (20 µM) ($n = 19$ LPS alone; $n = 12$ LPS + nigericin; $n = 8$ piericidin A, piericidin A + LPS, piericidin A + LPS + nigericin; $n = 17$ cyclocreatine; $n = 13$ cyclocreatine + LPS, cyclocreatine + LPS + nigericin). **c**, IL-1β protein levels in cell culture supernatant of BMDMs treated LPS (100 ng ml⁻¹) and nigericin (20 µM) with or without cyclocreatine (10 µM) ($n = 6$ for all treatments). **d**, IL-1β protein levels in cell culture supernatant of BMDMs treated LPS (100 ng ml⁻¹) and ATP (5 mM) with or without cyclocreatine (10 µM) ($n = 6$ for all treatments). **e**, Intracellular pro-caspase-1 protein expression in cell lysates from WT BMDMs treated with LPS (100 ng ml⁻¹) and ATP (5 mM) with or without cyclocreatine (10 µM) ($n = 5$ for all treatments). **f**, Intracellular caspase-1 (p20 fragment) protein expression in cell lysates from WT BMDMs treated as in **e** ($n = 4$ WT; $n = 7$ NDI1). **g**, Intracellular pro-caspase-1 protein expression in cell lysates from BMDMs transfected with vehicle control or siRNA against *Ckb* treated or not with LPS (100 ng ml⁻¹) and ATP (5 mM) ($n = 5$ independent experiments). **h**, Intracellular caspase-1 (p20 fragment) protein expression in cell lysates from BMDMs transfected with vehicle control or siRNA against *Ckb* treated or not with LPS (100 ng ml⁻¹) and ATP (5 mM) ($n = 5$ independent experiments). **i**, IL-1β protein levels in the serum of mice administered cyclocreatine before i.p. administration of 50 mg kg⁻¹ crude LPS. Serum samples were collected 2 h post LPS injection ($n = 13$ $H_2O$ + PBS; $n = 11$ cyclocreatine + cyclocreatine; symbols indicate independent experiments). Data are means ± s.e.m. *$P < 0.05$, one-way ANOVA with Tukey test for multiple comparisons (**a**, *$P = 0.0037$ UT/cyCr, *$P = <0.0001$ UT/ LPS + Cycr; **b**, *$P < 0.0001$ LPS/LPS + Nigericin, *$P = 0.0404$ LPS + Nigericin/LPS + Piericidin+Nigericin, *$P = 0.0313$ LPS + Nigericin/LPS + CyCr+Nigericin; **f**, *$P = 0.0007$; **h**, *$P = 0.0086$), two-tailed *t*-test (**c**, *$P = 0.0008$; **d**, *$P = 0.0008$; **i**, *$P = 0.0153$), one-sample *t*-test (**b**, *$P = 0.0097$ UT/Piericidin A, *$P < 0.0001$ UT/cyCr).

above a threshold level that is met either by mitochondrial complexes III and IV alone (in NDI1 BMDMs) or by mitochondrial complex I alone (in QPC-KO/AOX BMDMs).

**NLRP3 inflammasome activation is not linked to change in mitochondrial membrane potential.** Next, we tested whether changes in mitochondrial membrane potential (MPP) was necessary for NLRP3 activation. High or low MMP triggers increase or decrease in ETC-linked superoxide production, respectively[35]. We treated BMDMs with either oligomycin—an inhibitor of mitochondrial

complex V (ATP synthase)—or the protonophore carbonyl cyanide-p-trifluoromethoxyphenylhydrazone (FCCP). Oligomycin increases both NADH levels and MMP (Extended Data Fig. 7a–c). By contrast, FCCP allows efficient NADH oxidation but decreases MMP (Extended Data Fig. 7a–c). Treatment with oligomycin diminished oxygen consumption, as expected (Extended Data Fig. 8a). Although oligomycin did not significantly diminish the LPS-dependent increase in *Il1b* mRNA expression or intracellular pro-IL-1β protein levels (Extended Data Fig. 8b,c), it did attenuate the LPS-dependent increase in secreted IL-1β and intracellular cleaved caspase-1 protein

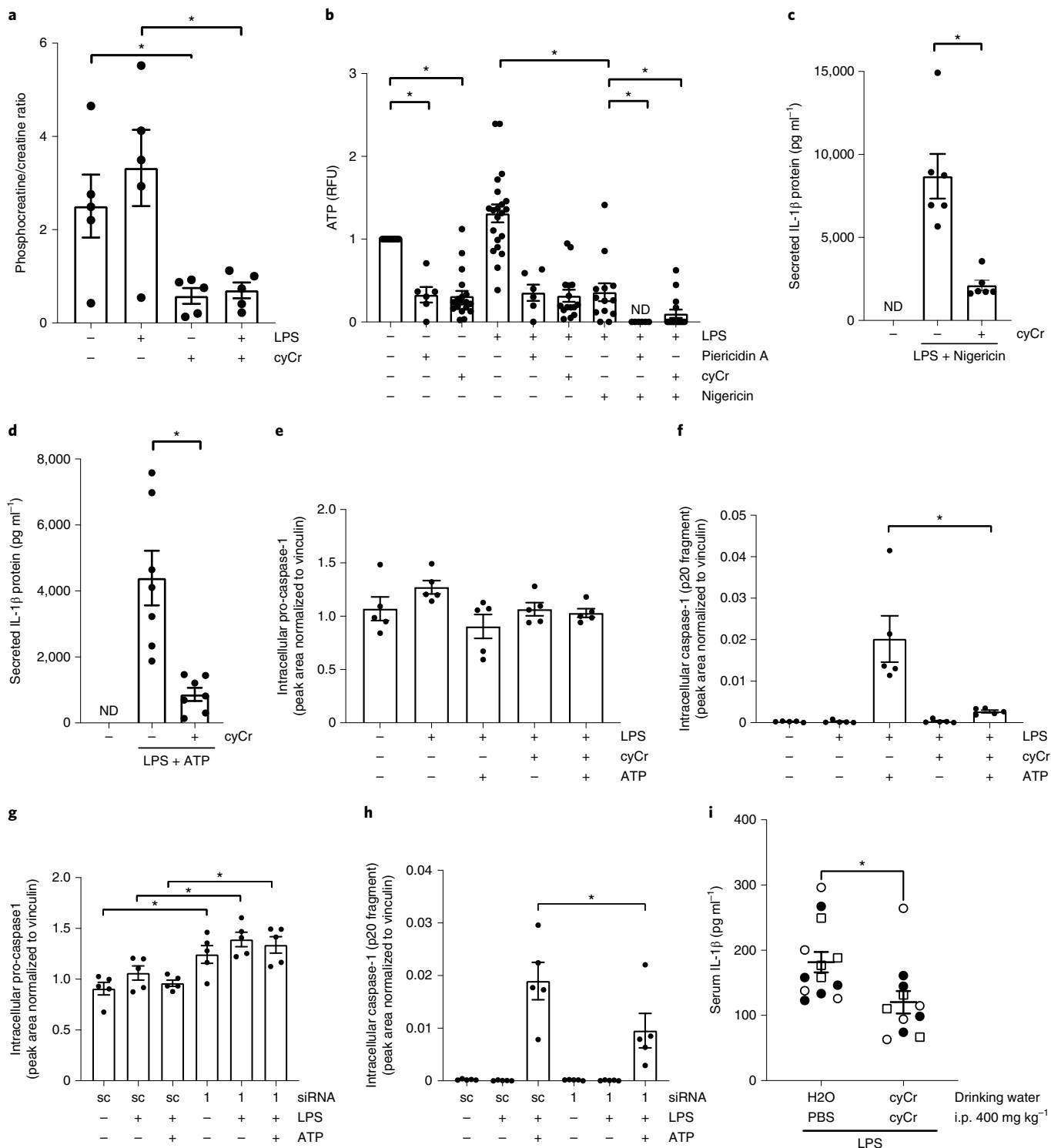

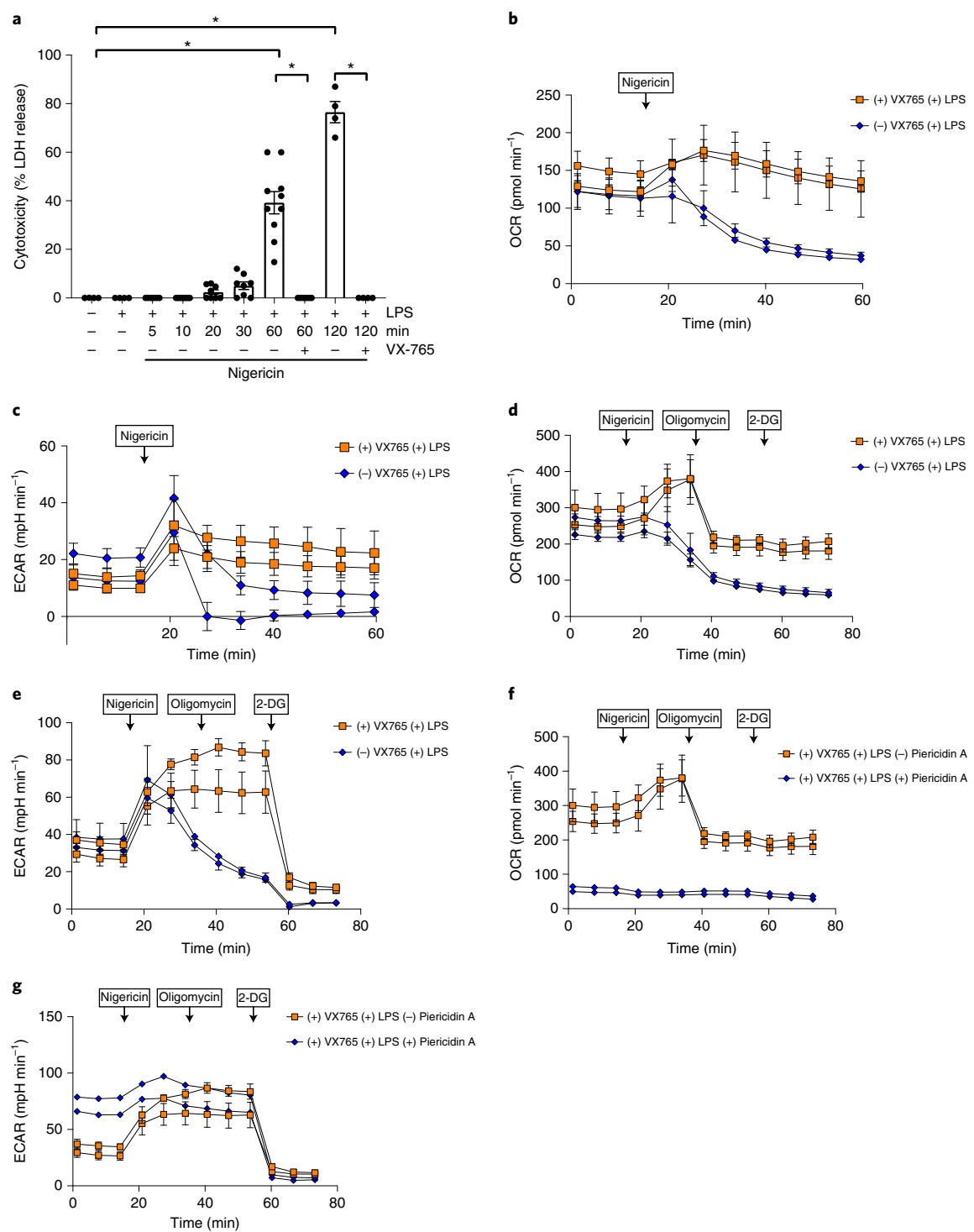

**Fig. 5 | Nigericin decreases OCR in an active caspase-1-dependent manner. a**, Percentage LDH release from BMDMs treated with LPS (100 ng ml⁻¹) and Nigericin (20 µM), with or without VX-765 (20 µg ml⁻¹) ($n=10$ UT, LPS, LPS + 60 min nigericin; $n=8$ LPS + 10 min nigericin, LPS + 20 min nigericin, LPS + 30 min + nigericin; $n=6$ LPS + 5 min nigericin, LPS + 60 min nigericin + VX-765; $n=4$ LPS + 120 min nigericin, LPS + 120 min nigericin + VX-765). **b**, OCR of BMDMs treated for 6 h with LPS in the presence or absence of VX-765 (20 µg ml⁻¹). Nigericin was added (final concentration 20 µM) at indicated time ($n=2$; error bars s.d. of four technical replicates). **c**, ECAR of BMDMs treated as in **b** ($n=2$; error bars represent s.d. of four technical replicates). **d**, OCR of BMDMs treated for 6 h with LPS (100 ng ml⁻¹), with or without VX-765 (20 µg ml⁻¹). Nigericin (final concentration 20 µM), Oligomycin (final concentration 2 µM), and 2DG (final concentration 50 mM) were added at indicated timepoints ($n=2$, representative of eight mice in four independent experiments). **e**, ECAR of BMDMs treated as in **d** ($n=2$, representative of eight mice in four independent experiments). **f**, OCR of BMDMs treated for 6 h with LPS (100 ng ml⁻¹) and VX-765 (20 µg ml⁻¹) with or without piericidin A (500 nM) ($n=2$, representative of eight mice in four independent experiments). **g**, ECAR of BMDMs treated as in **f** ($n=2$, representative of eight mice in four independent experiments). Data are means ± s.e.m. (**a**) or s.d. (**b–g**). *$P < 0.0001$, one-way ANOVA with Turkey's post hoc test for multiple comparisons.

levels in BMDMs treated with extracellular ATP (Extended Data Fig. 8d,e). Oligomycin did not affect intracellular pro-caspase-1 levels (Extended Data Fig. 8f). FCCP did not diminish the LPS-dependent increase in *Il1b* mRNA expression or intracellular pro-IL-1β protein levels (Extended Data Fig. 8g,h). FCCP did attenuate the LPS-dependent increase in secreted IL-1β and intracellular cleaved caspase-1 protein levels in BMDMs treated with extracellular ATP (Extended Data Fig. 8i, j). FCCP did not affect intracellular pro-caspase-1 levels (Extended Data Fig. 8k). FCCP and oligomycin have distinct effects on the MMP, yet both decrease intracellular cleaved caspase-1 protein levels. Thus, changes in the MMP are not linked to NLRP3 inflammasome activation.

**Mitochondrial-generated PCr supports NLRP3 inflammasome activation.** To identify one or more common metabolites altered upon inhibition of mitochondrial complexes I, II, III and V and disruption of the MMP, we inspected metabolomics data from cells treated with DMM, piericidin A, myxothiazol, oligomycin or FCCP (Fig. 1g, Extended Data Fig. 1c and Extended Data Fig. 9a,b). PCr was a common metabolite that increased during LPS priming and was diminished by all five inhibitors. The piericidin A-induced decrease in PCr was abrogated by expression of NDI1 (Fig. 1g). PCr is generated from creatine (Cr) and ATP by creatine kinase (CKMT2) in the mitochondria, then released into the cytosol where it is converted back to creatine by cytosolic CKB, transferring the phosphate group to ADP, thus generating cytosolic ATP (Extended Data Fig. 10a). This PCr shuttle provides readily available ATP for energy-consuming processes throughout the rest of the cell[36,37]. To deplete PCr from the cytosol, we treated BMDMs with cyclocreatine (cyCr)—a creatine analog. Cyclocreatine is readily phosphorylated by creatine kinase (CK) to produce phosphocyclocreatine, which is an inefficient donor of phosphate to ADP for ATP generation[38,39].

Treatment of BMDMs with cyCr decreased PCr/Cr levels (Fig. 4a). We measured intracellular ATP levels in BMDMs treated with cyCr to decrease ATP supply via the PCr shuttle or with piericidin A to inhibit mitochondrial complex I. Nigericin administration to LPS-primed BMDMs diminished the level of ATP, which was further decreased by piericidin A or cyCr (Fig. 4b). Cyclocreatine or RNAi against cytosolic CKB also decreased the level of secreted IL-1β in LPS-primed BMDMs treated with extracellular ATP or nigericin (Fig. 5c,d and Extended Data Fig. 9b,c). Cyclocreatine or RNAi against cytosolic CKB decreased intracellular cleaved caspase-1 protein levels without decreasing pro-caspase-1 protein levels (Fig. 4e–h). The administration of cyCr in vivo diminished LPS induced IL-1β protein in serum (Fig. 5i).

NLRP3 requires ATP hydrolysis for inflammasome activation[40,41]. The widely used NLRP3 inhibitor MCC950 interacts with the Walker B motif within the NLRP3 NACHT domain to prevent ATP hydrolysis[41]. We hypothesized that mitochondrial ETC-generated PCr is required to sustain the cytosolic store of ATP during NLRP3 inflammasome activation. However, LPS stimulation of BMDMs is thought to primarily stimulate glycolysis to sustain ATP levels[42]. Thus, we used extracellular acidification rate (ECAR) to assess glycolytic flux during NLRP3 activation. Nigericin induces cell death in LPS-treated BMDMs after 20 min in a caspase-1 dependent manner (Fig. 5a). Nigericin stimulated ECAR and OCR over 20 min in the presence of the caspase-1 inhibitor VX-765 (Fig. 5b–e). Notably, BMDMs treated with piericidin A during LPS plus nigericin stimulation (which cannot activate the NLRP3 inflammasome) are highly glycolytic (Fig. 5f,g). Thus, glycolysis-generated ATP is not sufficient to support NLRP3 inflammasome activation in the absence of mitochondrial ATP. Importantly, hypoxic cells can generate mitochondrial ATP[43], which may explain previous work indicating that NLRP3 inflammasome can also be activated under hypoxia (1% $O_2$). Collectively, these data indicate that NLRP3 inflammasome activation depends on mitochondria-derived ATP, initially generated by forward respiratory electron flow and supplied via the PCr shuttle.

**NLRP3 inflammasome activation by CL097 requires inhibition of mitochondrial complex I.** Typically, activation of the NLRP3 inflammasome requires $K^+$ efflux[2], which occurs upon extracellular ATP or Nigericin administration in LPS-primed BMDMs. However, the NLRP3 inflammasome can also be activated in a $K^+$ efflux-independent manner. Notably, $K^+$ efflux is dispensable for activation of NLRP3 inflammasome by imiquimod and the related molecule CL097 (ref. [3]). It has been proposed that these molecules inhibit the quinone oxidoreductases NQO2 and mitochondrial complex I to trigger ROS production, which stimulates NLRP3 inflammasome activation. We tested the necessity of mitochondrial complex I inhibition for CL097-dependent inflammasome activation by using our NDI1-expressing BMDMs. CL097 caused cell death within 20 min in an active caspase-1-dependent manner in LPS-primed BMDMs (Fig. 6a). However, CL097 decreased OCR in the presence of the caspase-1 inhibitor VX-765, indicating that the decrease in OCR was not due to cell death (Fig. 6b). This is consistent with the observation that CL097 decreases OCR in cells lacking inflammasome components[3]. NDI1 expression prevented CL097- or piericidin A-induced decrease in OCR, indicating that CL097 indeed inhibits mitochondrial complex I (Fig. 6c,d). NDI1 expression prevented CL097-dependent secreted IL-1β and intracellular cleaved caspase-1 protein levels in LPS-primed BMDMs without altering intracellular pro-caspase-1 levels (Fig. 6e–g). Next, we tested whether mitochondrial complex I inhibitor piericidin A or other ETC inhibitors, like CL097, are also sufficient to trigger inflammasome activation in LPS-primed BMDMs. None of the ETC inhibitors increased secreted IL-1β

**Fig. 6 | Mitochondrial complex I inhibition is necessary for CL097 activation of NLRP3 inflammasome. a,** Percent LDH release from BMDMs treated with LPS (100 ng ml⁻¹) and CL097 (70 μM), with or without VX-765 (20 μg ml–1); *n* = 4. **b,** OCR of BMDMs treated with LPS (100 ng ml⁻¹), with or without VX-765 (20 μg ml⁻¹). Data is shown as a percent of the basal OCR of untreated. CL097 (70 μM), Oligomycin (2 μM) and 2-deoxy-ᴅ-glucose (2DG) (50 mM) were added at indicated timepoints; *n* = 4. **c,d,** OCR of WT and NDI1 BMDMs treated with LPS (100 ng ml⁻¹). Data are shown as a percentage of the basal OCR of untreated. Piericidin A (500 nM) (**c**) or CL097 (70 μM) (**d**), Oligomycin (2 μM) and 2DG (50 mM) were added as indicated; *n* = 3 for each genotype. **e,** IL-1β protein levels in cell culture supernatant of WT or NDI1 BMDMs treated LPS (100 ng ml⁻¹) and CL097 (70 μM); *n* = 6. **f,** Intracellular caspase-1 (p20 fragment) protein expression in cell lysates from WT and NDI1 BMDMs treated as in Fig. **e** (70 μM); *n* = 6. **g,** Intracellular pro-caspase-1 expression in cell lysates treated as in **e**; *n* = 6. **h,** IL-1β protein levels in cell culture supernatant of WT BMDMs treated with LPS (100 ng ml⁻¹) and CL097 (70 μM), piericidin A (100 nM), myxothiazol (100 nM), antimycin A (100 nM) or oligomycin (50 nM); *n* = 4. **i,** IL-1β protein levels in cell culture supernatant of BMDMs treated with LPS (100 ng ml⁻¹) and CL097 (70 μM), with or without piericidin A (500 nM); *n* = 4. **j,** Intracellular pro-caspase-1 protein expression in cell lysates from BMDMs treated as in **i**; *n* = 4. **k,** Intracellular caspase-1 protein expression in cell lysates from BMDMs treated as in **i**; *n* = 4. **l,** IL-1β protein levels in cell culture supernatant of BMDMs treated with LPS (100 ng ml⁻¹) and CL097 (70 μM), with or without CyCr (10 μM) (*n* = 4). **m,** Intracellular caspase-1 protein expression in cell lysates treated as in **l**; *n* = 4. **n,** Intracellular pro-caspase-1 protein expression in cell lysates from BMDMs treated as in **l**; *n* = 4. Data are means ± s.e.m. *P < 0.05, two-tailed *t*-test (**i**, *P = 0.007; **l**, *P = 0.0127) one-way ANOVA with Tukey's post hoc test for multiple comparisons (**a**, *P = 0.0399 UT/LPS + 20 min; *P < 0.0001 UT/LPS + 30 min, UT/LPS + 60 min, LPS + 30 min/LPS + 30 min+VX-765, LPS + 60 min/LPS + 60 min + VX-765; *P = 0.0013 (**e**); *P < 0.0001 (**f**); *P = 0.0273 (**j**); *P < 0.0001 (**m**).

levels (Fig. 6h) indicating that CL097, in addition to inhibiting mitochondrial complex I, has other targets that are necessary for NLRP3 inflammasome activation, perhaps endolysomoal effects[3]. Although piericidin A cannot serve as an inflammasome activator, we tested whether administration of piericidin A or cyclocreatine during LPS priming would diminish CL097 activation of the NLRP3 inflammasome. Indeed, both piericidin A and cyclocreatine administered during LPS priming diminished secreted IL-1β levels and intracellular cleaved caspase-1 protein levels upon

CL097 administration (Fig. 6i–n). Thus, mitochondrial-generated ATP to sustain PCr levels during LPS priming is also necessary for CL097 activation of the NLRP3 inflammasome.

CL097 inhibition also triggers ROS production[3]. We tested whether increasing mitochondrial ROS in NDI1-expressing BMDMs, which are resistant to CL097, would rescue NLRP3 inflammasome activation. Antimycin is a well-described generator of mitochondrial superoxide production at complex III (ref. [44]). Antimycin releases superoxide from mitochondrial complex III

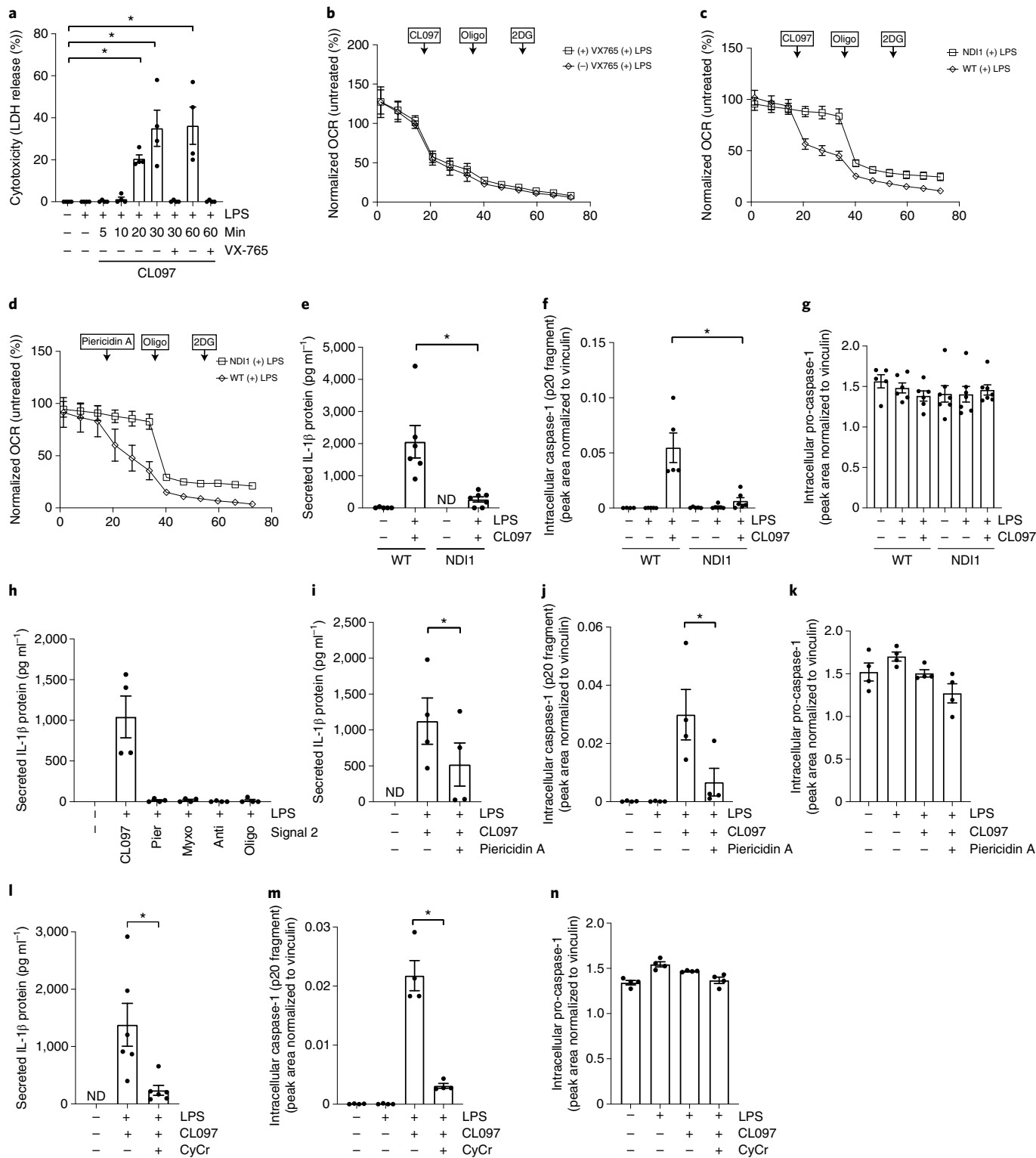

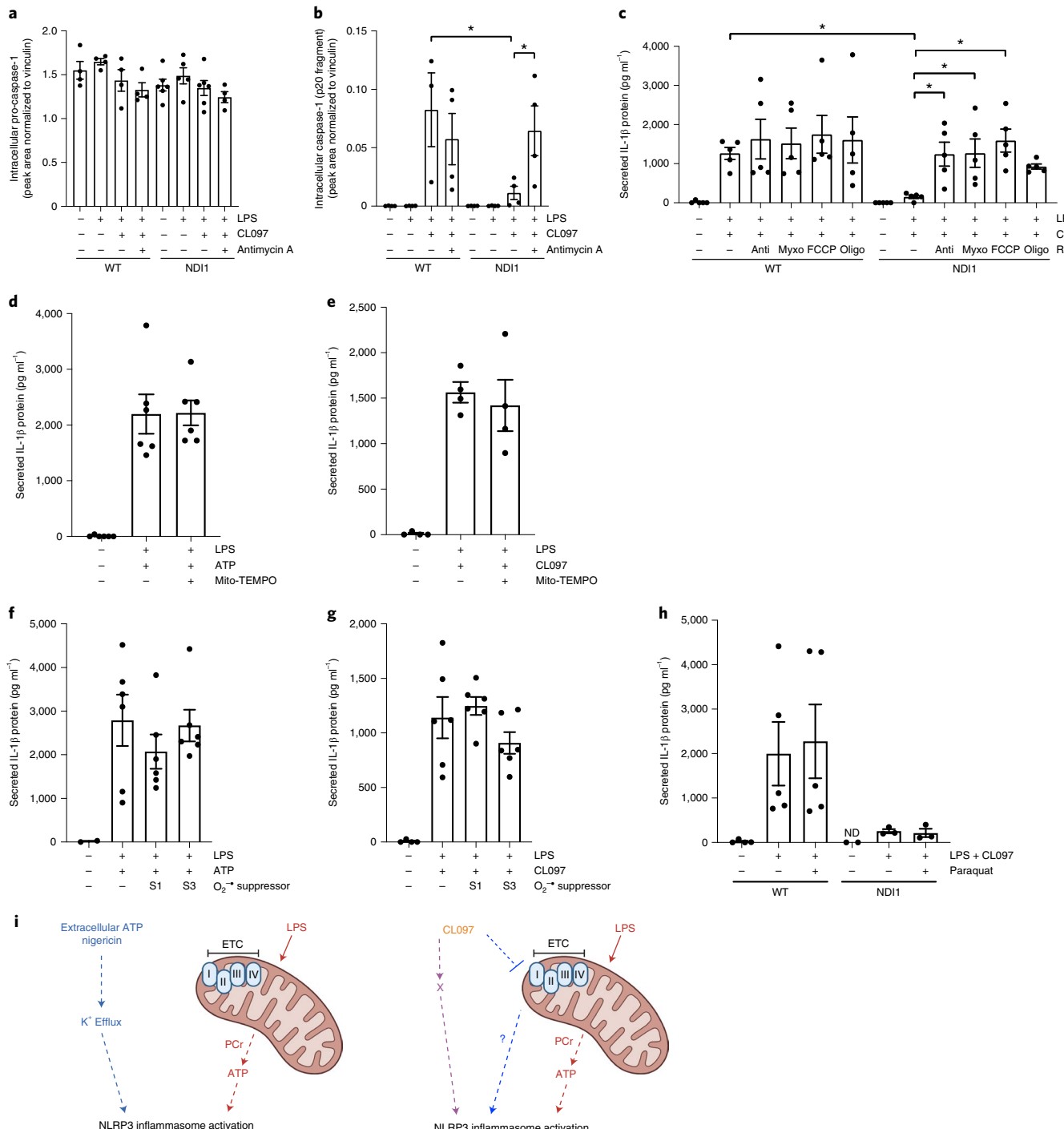

both in the mitochondrial matrix and intermembrane space[44]. By contrast, myxothiazol inhibits mitochondrial superoxide production at complex III[45]. Indeed, antimycin rescued intracellular cleaved caspase-1 (Fig. 7a,b) and secreted IL-1β protein levels (Fig. 7c) in NDI1-expressing LPS-primed BMDMs treated with CL097. Surprisingly, myxothiazol also increased secreted IL-1β levels in NDI1-expressing LPS-primed BMDMs treated with CL097. Moreover, oligomycin and FCCP, which have opposite effects on MMP and superoxide production, also increased secreted IL-1β levels in NDI1-expressing BMDMs primed with LPS and treated with CL097 (Fig. 7c). FCCP, unlike oligomycin and other ETC inhibitors, allows for efficient NAD⁺ regeneration (Extended Data Fig. 7a)[46]. These results suggest that the rescue effects observed

here by ETC inhibitors and FCCP are independent from ROS production or NAD⁺ regeneration. To directly test whether suppressing or scavenging mitochondrial superoxide could prevent CL097 or extracellular ATP activation of the NLRP3 inflammasome, we administered the mitochondrial-targeted superoxide dismutase mimetic MitoTEMPO. We also administered SEQEL1 (S1) or SEQEL3 (S3), which can suppress mitochondrial complex I- or III-generated superoxide production, respectively[47,48]. MitoTEMPO, S1 and S3 did not prevent CL097 or extracellular ATP activation of NLRP3 inflammasome (Fig. 7d–g). We used MitoTEMPO, S1 and S3 concentrations that do not inhibit OCR and have previously shown efficacy in other cell systems[47–50]. Finally, we tested whether increasing ROS production could rescue secreted IL-1β levels

**Fig. 7 | Mitochondrial ROS is not necessary for CLO97-dependent NLRP3 inflammasome activation. a**, Intracellular pro-caspase-1 expression in cell lysates from WT and NDI1 BMDMs treated with LPS (100 ng; ml⁻¹) and CL097 (70 μM); $N = 4$ for all treatments. **b**, Intracellular caspase-1 (p20 fragment) protein expression in WT and NDI1 BMDMs treated as in **a** ($n = 4$ NDI1 all treatments, WT UT, WT LPS, WT LPS + CL097 + antimycin A; $n = 3$ WT LPS + CL097). **c**, IL-1β protein levels in cell culture supernatant of WT and NDI1 BMDMs treated with LPS (100 ng ml⁻¹) and CL097 (70 μM). Antimycin (Anti; 100 nM), myxothiazol (Myxo; 100 nM), FCCP (10 μM) or oligomycin (Oligo; 50 nM) were added 30 min before inflammasome activation ($n = 5$ for all treatments). **d**, IL-1β protein levels in cell culture supernatant of BMDMs treated with LPS (100 ng ml⁻¹) and ATP (5 mM). MitoTempo (500 μM) was added 30 min before ATP ($n = 6$ for each condition). **e**, IL-1β protein levels in cell culture supernatant of BMDMs treated with LPS (100 ng ml⁻¹) and CL097 (70 μM). MitoTempo (500 μM) was added 30 min before the addition of CL097 ($n = 6$ for each condition). **f**, IL-1β protein levels in cell culture supernatant of BMDMs treated with LPS (100 ng ml⁻¹) and ATP (5 mM). S1QEL (S1; 1 μM) or S3QEL (S3; 10 μM) was added 30 min before ATP ($n = 6$ for each condition). **g**, IL-1β protein levels in cell culture supernatant of BMDMs treated with LPS (100 ng ml⁻¹) and CL097 (70 μM). S1QEL (S1; 1 μM) or S3QEL (S3; 10 μM) was added 30 min before CL097 ($n = 6$ for each condition). **h**, IL-1β protein levels in cell culture supernatant of BMDMs treated with LPS (100 ng ml⁻¹) and CL097 (70 μM). Paraquat (25 μM) was added 30 min before CL097 ($n = 5$ WT all treatments; $n = 3$ NDI1 all treatments). **i**, Schematic of K⁺ efflux-dependent (left) and independent (right) NLRP3 inflammasome activation. Error bars represent means ± s.e.m. *$P < 0.05$, one-way ANOVA with Tukey's post hoc test for multiple comparisons (**b**, *$P = 0.0454$ WT LPS + CL097/NDI1 LPS + CL097, *$P = 0.0229$ NDI1 LPS + CL097/NDI1 LPS + CL097 + Antimycin A; **c**, *$P < 0.0001$ WT LPS + CL097/NDI1 LPS + CL097, *$P = 0.0434$ NDI1 LPS/CL097/NDI1 LPS + CL097 + Anti, *$P = 0.0373$ NDI1 LPS + CL097/NDI1 LPS + CL097 + Myxo, *$P = 0.0053$ NDI1 LPS + CL097/NDI1 LPS + CL097 + FCCP). Parts of this figure were created with BioRender.com.

in NDI1-expressing LPS-primed BMDMs treated with CL097. Paraquat—a known generator of superoxide production[51]—failed to increase secreted IL-1β levels (Fig. 7h). Collectively, our data indicate that CL097 requires inhibition of mitochondrial complex I to trigger NLRP3 inflammasome activation through an unidentified mitochondria-dependent mechanism (Fig. 7i).

## Discussion

Our studies on LPS-primed BMDMs activated with extracellular ATP or CL097 have revealed three important aspects of mitochondrial ETC in controlling NLRP3 inflammasome activation. First, LPS priming increases mitochondrial ATP-dependent PCr levels that sustain NLRP3 inflammasome activation by both extracellular ATP and CL097. Importantly, mitochondrial ETC inhibitors maximally activate glycolysis, which is not able to sustain NLRP3 inflammasome activation. Second, mitochondrial ETC inhibitors are not sufficient to trigger NLRP3 inflammasome activation in LPS-primed BMDMs, consistent with previous findings[3]. Nevertheless, CL097 inhibits mitochondrial complex I to activate NLRP3 inflammasome in LPS-primed BMDMs, suggesting that CL097 targets mitochondrial complex I and some other unknown target(s) to activate the NLRP3 inflammasome. Moreover, the mechanism by which CL097 inhibition of mitochondrial complex I is necessary for NLRP3 inflammasome activation is not clear. Third, we find no evidence that mitochondrial ROS are necessary for NLRP3 inflammasome activation by extracellular ATP or CL097, although we cannot exclude nonmitochondrial ROS sources as potential inputs into NLRP3 activation. Collectively, our studies establish the necessity of the ETC to sustain NLRP3 inflammasome activation by both K⁺ efflux-dependent, that is, extracellular ATP, and K⁺ efflux-independent, that is, CL097, stimuli.

It is important to note that we examined here only one critical aspect of inflammation—the production of IL-1β by the canonical NLRP3 inflammasome. Other cytokines that are linked to mitochondrial ETC function, such as tumor necrosis factor alpha (TNF-α) and interleukin (IL)-6, may depend on mitochondrially generated ROS[52–54]. Mitochondrial DNA (mtDNA), known to activate the cGAS-STING pathway for induction of type I interferons, is potentially another input into NLRP3 inflammasome activation[55,56]. Our studies do not address or refute this mechanism. However, it is important to note that experimental strategies that deplete mtDNA also disable ETC function and thereby diminish mitochondrial ATP production. Thus, it is possible that depletion of mtDNA by TFAM ablation or cytidine monophosphate kinase 2 impairs NLRP3 inflammasome activation[55,57], in part due to diminished mitochondrial ATP.

Our present studies do not address whether the mechanism described here would apply to stimuli, such as serum amyloid, that require longer period (24 h) of exposure[58,59]. Furthermore, we did

not address whether ETC is necessary for the activation of other inflammasomes. Nevertheless, the genetic tools used in this study could be helpful in elucidating the necessity of mitochondrial ETC for different NLRP3 inflammasome stimuli as well as other distinct inflammasomes, such as AIM2 and NLRC4.

## Online content

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

## Methods

**Mice.** Male and female mice were used at 8–14 weeks of age. Littermate controls were used for all experiments. WT C57BL/6J mice were obtained from Jackson Laboratories and bred inhouse at Northwestern University. We used a previously published transgenic mouse line, which contains a lox-stop-lox-NDI1 targeting construct in the Rosa26 locus[26]. To generate mice that express NDI1 in hematopoietic cells, we bred mixed C57Bl/6J/N Rosa26[NDI1-lsl/wt] mice with B6.Cg-*Commd10*[Tg(Vav1-icre)A2Kio]/J mice from Jackson Laboratories to generate the conditional NDI1 transgenic mouse. Mice expressing AOX in myeloid cells were generated by breeding B6.129P2-*Lyz2*[tm1(cre)Ifo]/J mice from Jackson Laboratories with previously published C57Bl/6N Rosa26[AOX-lsl/wt] mice given to us by M. Szibor[28]. Previously published QPC[fl/fl] mice are C57Bl/6J background. All mouse lines were maintained at Northwestern University under specific pathogen-free conditions in ventilated microisolator cages with automatic water access. Teklad LM-485 mouse/rat sterilizable diet chow (Envigo, catalog no. 7912) was provided ad libitum. Housing rooms had standard 12-h light/dark cycles and an ambient temperature of 23 °C. We complied with all relevant ethical regulations in accordance with Federal and University guidelines and protocols approved by IACUC and Northwestern University, protocol number IS00014481.

**BMDM isolation and cell culture.** Bone marrow was isolated from mice and plated in 10 cm Primaria tissue culture plates (ThermoFisher, catalog no. 25382-701). To induce differentiation into macrophages, cells were cultured in RPMI medium containing 11 mM glucose, 10% fetal+ serum (Atlas Biologics, catalog no. P16E19A1), 1 mM methyl pyruvate (Sigma, catalog no. 371173), 400 μM uridine (Sigma, catalog no. U3003), 1% antibiotic/mycotic (ThermoFisher, catalog no. 15-240-062) 1% Hepes (ThermoFisher, catalog no. MT25060CI) and 4 mM glutamine (Gibco, catalog no.11965-126) supplemented with 20 ng ml$^{-1}$ M-CSF (Peprotech, catalog no.315-02) at 37 °C with 5% CO$_2$. The medium was changed every 3 days, and BMDMs were harvested by scraping on day 6 and plated in 12-well plates (2 million cells per well), 24-well plates (940,000 cells per well), 48-well plates (300,000 cells per well) or 96-well plates (150,000 cells per well).

**Quantitative PCR with reverse transcription.** Quantitative PCR reverse transcription (qRT–PCR) was performed on BMDMs treated with 100 ng ml$^{-1}$ ultrapure O5:B55 LPS (Invivogen, catalog no. tlrl-pb5lps) for 4 h. Before stimulation, BMDMs were treated for 30 min with piericidin A (Cayman Chemical, catalog no. 15379), myxothiazol (Sigma, catalog no. T5580), FCCP (Sigma, catalog no. C2920), oligomycin (Sigma, catalog no. 75351) or dimethyl malonate (Sigma, catalog no. 136441), as indicated in figures and figure legends.

RNA was extracted from peritoneal macrophages or BMDMs using the Omega E.Z.N.A. RNA Isolation Kit (Omega Biologicals, catalog no. R6834-02). RNA was quantified using a Nanodrop 2000 UV-visible spectrophotometer, and 300 ng of RNA was reverse transcribed using RETROscript first-strand synthesis kit (ThermoFisher, catalog no. AM17-10). Real-time PCR was performed on a BioRadCFX using iQ SYBR green Supermix (Bio-Rad, catalog no. 1708880). The following primers were used: IL-1β (forward 5′- TGGCAACTGTTCCTG-3′; reverse 3′-GGAAGCAGCCCTTCATCTTT-5′); NDI1 (forward 5′-GCCGAAGA AGTCCAAATTCAC-3′; reverse 3′- CGACAGCCGTTCTCAGAT-5′); b-Actin (forward 5′-CTAAGGCCAACCGTGAAAA-3′; reverse 3′-ACCAGAGGCATACA GGGACA-5′). Fold changes in gene expression relative to untreated control were calculated by the ΔΔC$_t$ method using mouse actin as an endogenous control for mRNA expression.

**Oxygen consumption rate.** The OCR was measured in a XF96 extracellular flux analyzer (Agilent Bioscience). BMDMs were plated at $0.15 \times 10^6$ cells per well of a XF96 plate, allowed to adhere overnight. Pretreated cells were treated with myxothiazol, piericidin A or oligomycin for 30 min, or dimethyl malonate for 3 h before OCR measurement. LPS-treated cells were primed with 100 ng ml$^{-1}$ LPS for 6 h. At 1 h before OCR measurement, the medium was exchanged for Seahorse base RPMI (Agilent, catalog no. 103335-100, supplemented with glucose, methyl pyruvate, glutamine and uridine) in the presence or absence of 20 μg ml$^{-1}$ VX-765 (Invivogen, catalog no. inh-vx765i-5) before initiation of the assay. Injection of ETC inhibitors and inflammasome activators occurred at the timepoints indicated in the figures. Final concentrations of drugs are included in the figure legends. Basal OCR was assessed by subtracting nonmitochondrial oxygen consumption (measured in the presence of 1 μM antimycin A (Sigma, catalog no. A8674) and 1 μM piericidin A) from the baseline OCR. Coupled OCR was assessed as the difference between basal OCR and OCR after addition of oligomycin.

**NAD$^+$:NADH ratio.** BMDMs were plated at $0.15 \times 10^6$ cells per well in 96-well plates and allowed to adhere overnight. The NAD$^+$/NADH ratio was measured in BMDMs treated with or without LPS for 4 h after pretreatment with or without of piericidin A or dimethyl malonate using the Promega NAD$^+$/NADH Glo-Kit (Promega, catalog no. G9071) according to the manufacturer's instructions.

**Inflammasome activation.** BMDMs were plated at $0.15 \times 10^6$ cells per well and allowed to adhere overnight. Cells were treated with metabolic inhibitors for 30 min before priming with 100 ng ml$^{-1}$ ultrapure O5:B55 LPS (Invivogen, catalog

no. tlrl-pb5lps) for 5.5 h. To activate the NLRP3 inflammasome, 5 mM ATP (Sigma, catalog no. A2383) or 70 μM CL097 (Invivogen, catalog no. tlrl-c97) was added for 30 min, or 20 μM nigericin (Sigma, catalog no. N7143) was added for 1 h.

To avoid oxidation of cyclocreatine, we avoided repeated freeze–thaws of the powder and solution. Cell supernatant was collected and used for IL-1β and TNFα measurement via enzyme-linked immunosorbent assay (ELISA) (R&D Duoset, catalog nos. DY401-05 and DY410-05). ELISA kits were used according to manufacturer's instructions.

To assess capase-1 cleavage and pro-IL-1β levels, BMDMs were plated at 2 million cells per well in a 12-well plate and allowed to adhere overnight. BMDMs were primed with LPS (100 ng ml$^{-1}$) for 5.5 h. ATP (5 mM) or CL097 (70 μM) was added for 10 min to activate the NLRP3 inflammasome. Cells were lysed with NP40 cell lysis buffer (ThermoFisher, catalog no. FNN0021) with Halt protease and phosphatase inhibitor (ThermoFisher, catalog no. 78442) and protein concentration was quantified via bicinchoninic acid (BCA) assay. Immunoblot was performed on the Wes (ProteinSimple) according to the manufacturer's instructions for pro-caspase-1/caspase-1 (Adipogen, catalog no. AG-20B-0042; dilution 1:200), pro-IL-1β/IL-1β (R&D systems, catalog no. AF-401-NA; dilution 1:100), anti-ASC (Novus Biologics, catalog no. NBP1-78977SS; 1:50 dilution), anti-NLRP3 (Novus Biologics, catalog no. NBP2-03948SS, clone 25N10E9; 1:100 dilution) and vinculin (Cell Signaling, catalog no. 13901; dilution 1:500). Chemiluminescence of all proteins was quantified on Compass software (v.5.0.1; ProteinSimple). Relative protein expression levels were quantified as peak area of IL-1β, pro-IL-1β, pro-caspase-1 or caspase-1 over the peak area of vinculin.

**H$_2$O$_2$ measurement in permeabilized BMDMs.** BMDMs were centrifuged at 500$g$ and washed first with KHEB buffer (120 mM KCL, 5 mM HEPES, 1 mM EGTA, 0.3% BSA, pH 7.4 with KOH), followed by a wash with KHEB + 100 μg ml$^{-1}$ saponin (Sigma, catalog no. 47036). A total of 150,000 cells were incubated in 96-well, clear-bottom plates (Corning, catalog no. 3603) in 50 μl KHEB + saponin for 15 min at room temperature. Following incubation, an additional 50 μl of KHEB buffer was added to each well along with 100 μl superoxide sensing solution (1.5 U ml$^{-1}$ HRP (ThermoFisher), 25 KU ml$^{-1}$ superoxide dismutase (Sigma, catalog no. S5395) and 25 μM Amplex Red (Invitrogen, catalog no. A222188)) was added to each well. Additional treatments were added to superoxide sensing solution as follows: 500 μM L-succinate (Sigma, catalog no. 224731), 500 nM piericidin A. Immediately following addition of superoxide sensing solution, the plate was placed in a SpectraMax M2 (Molecular Devices) plate reader set to mix plate and take readings every minute with excitation (544 nm) and emission (590 nm). For analysis, the fluorescent readings from the linear range of the reaction (15–30 min) were used to calculate the slope (RFU min$^{-1}$). RFU min$^{-1}$ readings from the BMDMs treated with superoxide sensing solution alone are subtracted as background readings.

**ATP assay.** A total of 2 million BMDMs were plated in a 12-well plate, as indicated above, and allowed to adhere overnight. BMDMs were treated with 100 nM piericidin A or 10 mM cyclocreatine for 30 min before the addition of 100 ng ml$^{-1}$ LPS for 4 h where indicated. Nigericin (20 μM) was added where indicated for 20 min to allow for the initiation of NLRP3 inflammasome activation before cell death. Cells were harvested with ATP assay buffer (ATP Assay kit (Colorimetric/Fluorometric), Abcam, catalog no. ab83355) and centrifuged at 14,000$g$ for 5 min. The assay was performed according to the manufacturer's instructions using the fluorometric protocol. Fluorescent readings from each treated sample replicate are shown as relative to the fluorescent reading of the corresponding untreated sample.

**Metabolomics.** Two million BMDMs were allowed to adhere overnight in 12-well plates. The cells were treated with 50 nM oligomycin, 500 nM piericidin A, 100 nM myxothiazol or 10 mM cyclocreatine for 30 min, or 10 mM dimethyl malonate for 3 h, before stimulation with 100 ng ml$^{-1}$ LPS for 4 h. To extract metabolites, 1 ml HPLC-grade methanol in water (80/20, v/v) cooled to –80 °C. Cells went through three complete freeze–thaw cycles in liquid nitrogen and a 37 °C waterbath before high-speed centrifugation at 4 °C. The supernatants, which contained metabolites, were collected and stored at –80 °C. The supernatants were dried in a SpeedVac concentrator (Thermo Savant). The dried metabolites were reconstituted in acetonitrile in analytical-grade water (50/50, v/v) and centrifuged to remove debris. A 10 μl aliquot of the sample was used for high-resolution HPLC-tandem mass spectrometry. High-resolution HPLC-tandem mass spectrometry was performed on a Q-Exactive (ThermoFisher Scientific) in line with an electrospray source and an UltiMate 3000 (ThermoFisher Scientific) series HPLC consisting of a binary pump, degasser and autosampler outfitted with a XBridge Amide column (Waters; 4.6 mm × 100 mm dimension and a 3.5 μm particle size). Mobile phase A contained water and acetonitrile (95/5, v/v), 10 mM ammonium hydroxide and 10 mM ammonium acetate (pH 9.0). Mobile phase B was 100% acetonitrile. The gradient was set to 0 min, 15% A; 2.5 min, 30% A; 7 min, 43% A; 16 min, 62% A; 16.1–18 min, 75% A; 18–25 min, 15% A, with a flow rate of 400 μl min$^{-1}$. The capillary of the electrospray ionization source was set to 275 °C, with sheath gas at 45 arbitrary units, auxiliary gas at 5 arbitrary units and the spray voltage at 4.0 kV. A mass/charge ratio scan ranging from 70 to 850 was used in positive/negative polarity switching mode. MS1 data were collected at a resolution

of 70,000. The automatic gain control (AGC) target was set at $1 \times 10^6$, with a maximum injection time of 200 ms. The top five precursor ions were fragmented using the higher-energy collisional dissociation cell with normalized collision energy of 30% in MS2 at a resolution of 17,500. Data were acquired with Xcalibur software (v.4.1; ThermoFisher Scientific). The resulting data were analyzed using MetaboAnalyst (v.4.0), normalized by total ion current. Significantly different metabolites between treatment groups were identified by one-way analysis of variance (ANOVA) with Fisher's least significant difference (LSD) post hoc analysis and then plotted as a heatmap. Peak areas of individual metabolites (that is, succinate, PCr, creatine) were graphed as arbitrary units and subjected to one-way ANOVA with Tukey $t$-test post hoc analysis for multiple comparisons.

**RNA sequencing.** BMDMs from WT and NDI1 mice were seeded in 12-well plates as described above. BMDMs were pretreated with or without 500 nM piericidin A for 30 min before addition of 100 ng ml⁻¹ ultrapure LPS for 4 h. Samples were lysed with RLT Plus buffer (Qiagen, catalog no. 74134) with β-mercaptoethanol (1%) and homogenized with QIAshredder Spin Columns (Qiagen, catalog no. 79654). RNA was extracted using the RNeasy Plus Mini Kit (Qiagen, catalog no. 74134), according to the manufacturer's protocol plus on-column DNase treatment using the RNase-Free DNase Set (Qiagen, catalog no. 79254). RNA was quantified and quality control performed using the Agilent 4200 TapeStation RNA ScreenTape. mRNA libraries were prepared using NEBNext Ultra Kit with polyA selection (New England BioLabs). Sequencing of libraries was performed using a Next-Seq 500 High output for 75 cycles (Illumina). Raw BCL read files were demultiplexed and FASTQ files were generated using bcl2fastq and trimmed using Trimmomatic[60]. The reads were then aligned to the mouse mm10 reference genome using STAR to generate BAM files[61]. HTSeq was used to count reads in the exons of genes[62]. Likelihood ratio tests for all samples and all detected transcripts and pairwise differential gene expression analyses were carried out using the R package DESeq2 (ref. [63]).

**LPS induction of IL-1β protein in mice.** Crude O5:B55 LPS (Sigma, catalog no. L2880) was prepared at 5 mg ml⁻¹ in PBS. Littermate mice (WT, QPC-KO, QPC-KO/AOX and NDI1) were administered at 100 mg kg⁻¹ or 50 mg kg⁻¹ LPS, as specified in figure legends, via the intraperitoneal (i.p.) route for 2 h. For cyclocreatine treatments, cyclocreatine was made freshly before each experiment at 10 mg ml⁻¹ in PBS and brought to a pH of 7.4. To avoid oxidation of cyclocreatine powder, we limited repeated freeze–thaws and exposure of individual vials to air. Cyclocreatine in the drinking water was prepared at 1% weight/volume and administered overnight ad libitum. Cyclocreatine solution or PBS vehicle was administered i.p. to C57Bl/6J mice at 400 mg kg⁻¹ for 2 h before administration of 50 mg kg⁻¹ crude LPS for 2 h. Whole blood samples were harvested via retro-orbital bleed before euthanasia in a $CO_2$ chamber. Samples were allowed to clot and then centrifuged at 14,000$g$ for 15 min and serum was collected. IL-1β concentration in serum was measured using the IL-1β Quantikine ELISA kit (R&D Systems, catalog no. MLB00C) as per the manufacturer's instructions.

**Cell death assay using LDH release.** BMDMs were plated at $0.15 \times 10^6$ in a 96-well plate and allowed to adhere overnight. Cells were primed with LPS (100 ng ml⁻¹) and the NLRP3 inflammasome activated with Nigericin (20 μM) or CL097 (70 μM) for the times indicated with or without the caspase-1 inhibitor VX-765 (20 μg ml⁻¹). Plates were spun down at 500$g$ for 1 min and cell culture supernatant was transferred to a fresh plate. Assay was performed on supernatant according to manufacturer's instruction using the Cytotoxicity Detection Kit (LDH) (Sigma, catalog no. 11644793001).

**CKB knockdown using lipid nanoparticles/short interfering RNA complex.** Lipid nanoparticles (LNPs) for in vitro creatine kinase b (CKB) knockdown in macrophages were synthesized through rehydration of a thin film of lipid mixture containing 1,2-dioleoyl-3-trimethylammonium-propane, cholesterol and 1,2-distearoyl-sn-glycero-3-phosphoethanolamine-N-[methoxy(polyethylene glycol)-2000] (Avanti Polar Lipids), followed by sonication. Murine CKB siRNA (Sigma) was complexed with LNP in HEPES buffer for 20 min at room temperature. Macrophages were treated with LNP/short interfering RNA (siRNA) (CKB or scrambled negative control) complex in Opti-MEM at an siRNA concentration of 100 nmol l⁻¹ for 6 h at 37 °C. The medium was then replaced with complete RPMI. Cells were cultured for additional 24 or 48 h. At 24 h after transduction, some wells were lifted and quantitative PCR (qPCR) was performed to assess knockdown efficacy. Inflammasome activation was performed as described 48 h after LNP/siRNA treatment. The following CKB siRNA sequences were used: (5′-3′): siRNA1 (forward 5′-GGCAUAUGGCACAAUGACA[dT][dT]-3′; reverse 5′-UGUCAUUGUGCCAUAUGCC[dT][dT]-3′); siRNA2 (forward 5′-GACUUUCCUGGUGUGGAUU[dT][dT]-3′; reverse 5′-AAUCCACACCAGGAAAGUC[dT][dT]-3′); siRNA3 (forward 5′-GAGUGAAACUACUCAUUGA[dT][dT]-3′; reverse 5′-UCAAUGAGUAGUUUCACUC[dT][dT]-3′).

The following primers were used to assess knockdown efficacy: CKB (forward 5′-AGTTCCCTGATCTGAGCAGC-3′; reverse 5′-GAATGGCGTCGTCCAAA GTAA-3′); Actin (forward 5′-TTGCTGACAGGATGCAGAAG-3′; reverse

5′-ACATCTGCTGGAAGGTGGAC-3′). Fold changes in gene expression relative to untreated control were calculated by the $\Delta\Delta C_t$ method using mouse actin as an endogenous control for mRNA expression.

**Membrane potential measurement.** BMDMs were plated at 2 million cells per well in a 12-well plate and allowed to adhere overnight. Cells were primed for 4 h with LPS (100 ng ml⁻¹) in the presence or absence of FCCP (10 μM), piericidin A (Pier) (100 nM), Oligomycin (50 nM) or Myxothiazol (100 nM). TMRE (Abcam, catalog no. ab113852) was added at a concentration of 200 nM for 30 min. Cells were washed with PBS and removed from the plate with Accutase (Fisher Scientific, catalog no. NC9839010) before resuspension in PBS supplemented with 10% NU-Serum IV (Fisher Scientific, catalog no. CB-55004). Data were obtained using a BD FACSymphony A5-Laser Anaylzer (BD Biosciences).

**Statistical analysis.** Statistical analyses were performed in GraphPad Prism v.9 software using statistical tests indicated in the figure legends. Statistical analyses of metabolomics data were performed using Metaboanalyst[64]. Data are presented as mean ± s.e.m. with a minimum of $n = 3$ independent experiments, except Fig. 6b–g, which are presented as mean ± s.d. of four technical replicates. Specific number of replicates are indicated in figure legends. Experiments were neither randomized nor blinded. Statistical significance was determined by a two-tailed $t$-test, a one-sample $t$-test, an ANOVA followed by Tukey's multiple comparison test or an ANOVA followed by Fisher's LSD. Specific tests are indicated in figure legends. Statistical significance was defined as follows: $P < 0.05$. Data distribution was assumed to be normal, but this was not tested formally. No statistical methods were used to predetermine sample sizes, but our samples sizes are similar to those reported in previous publications[19,34,41]. Plated cells were allocated randomly to each treatment group; C57Bl/6 mice were assigned randomly to each treatment group. Data collection and analysis were not performed blind to the conditions of the experiments. Experiments were excluded from analysis if the controls did not work; data from successfully completed experiments were not excluded.

**Reporting Summary.** Further information on research design is available in the Nature Research Reporting Summary linked to this article.

## Data availability
RNA-seq data have been deposited in GEO under the accession code GSE197606. Source data for ProteinSimple (Wes) and metabolomics are provided with this paper. All other data are present in the article and supplementary information files or can be obtained from the corresponding author upon reasonable request.

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

## Acknowledgements
This work was supported by the National Institutes for Health (NIH) (5R35CA197532, 5P01HL071643-15 and 5PO1AG049665) to N.S.C.; L.K.B. was supported by the NIH/National Heart, Lung, and Blood Institute (NHLBI) (T32HL076139-15). J.M. was supported by the NIH (1R01NS115955-01). J.S.S. was supported by the NIH/National Institute of Allergy and Infectious Diseases (NIAID) T32 (T32AI083216). Additional support from the NIH (F30CA250236) to K.V., NIH/NHLBI T32HL076139-17 to T.A.P. and the Ford Foundation to A.E.R. We thank the Robert H. Lurie Cancer Center Flow Cytometry facility and Metabolomics Core supported by National Cancer Institute Cancer Center Support Grant (NCI CCSG) P30 CA060553 for their invaluable assistance. We thank P. Gao at Northwestern University for his expertise in metabolomics. We thank H. Abdala-Valencia at Northwestern University for RNA sequencing. We thank O. Grob at University of Freiburg for his helpful comments and discussion of the paper. Some figure elements were generated with biorender.com.

## Author contributions
L.K.B, J.S.S, J.M. and N.S.C. conceptualized the study, interpreted the data and wrote the manuscript with the input of all authors. K.V. analyzed RNA-seq data. L.K.B, J.S.S, T.A.P., A.R., P.Z. and A.E.R. carried out experiments in the paper. M.S. and H.T.J. provided AOX mice reagent and input in the editing of the paper. A.E.R provided technical expertise with mouse experiments. C.R.R. generated the NDI1 mice.

## Competing interests

The authors declare no competing interests.

## Additional information

**Extended data** is available for this paper at https://doi.org/10.1038/s41590-022-01185-3.

**Correspondence and requests for materials** should be addressed to Navdeep S. Chandel.

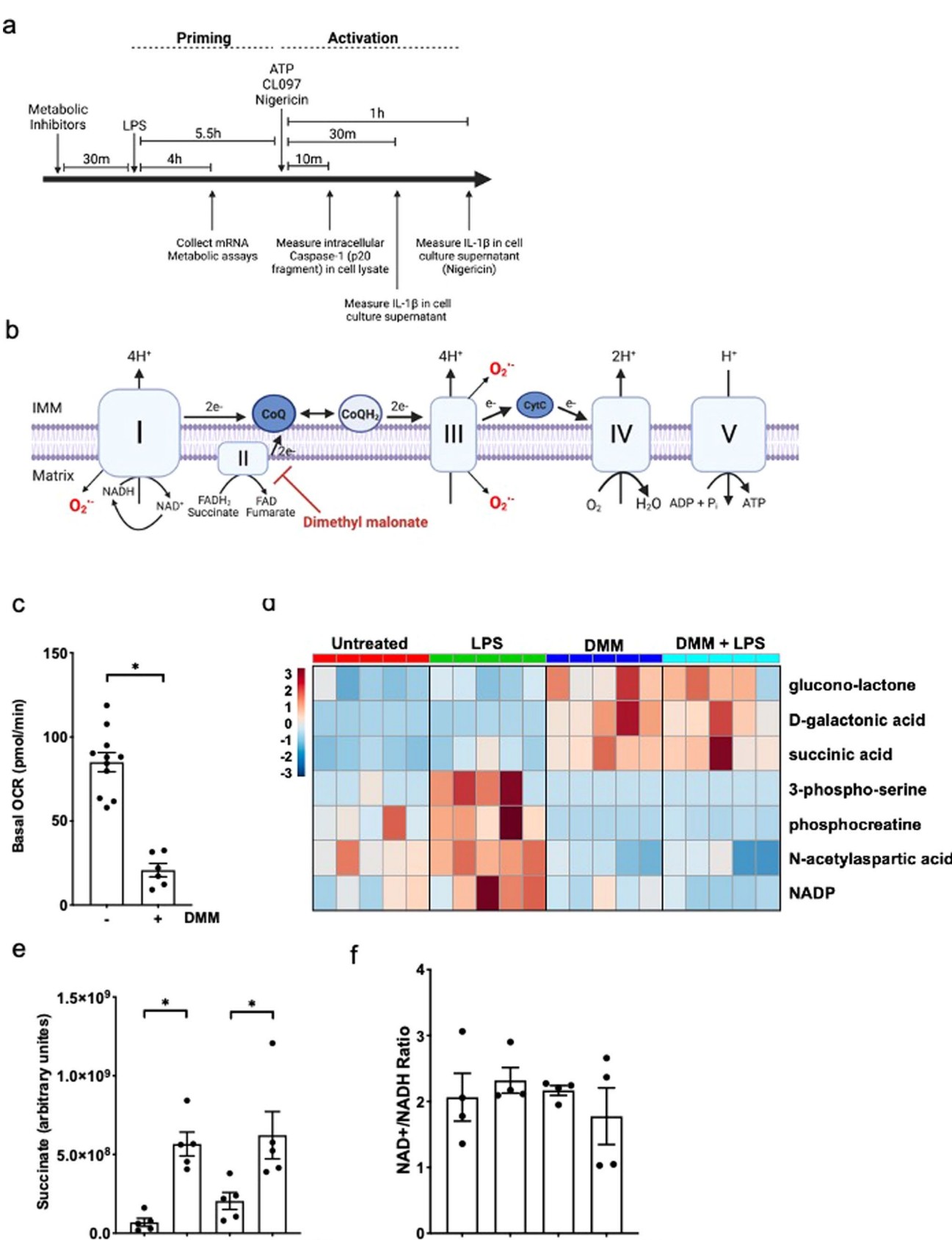

**Extended Data Fig. 1 | See next page for caption.**

**Extended Data Fig. 1 | Dimethyl malonate (DMM) inhibits mitochondrial complex II.** a) Timeline of treatment of BMDMs with metabolic inhibitors, LPS, and inflammasome activators. b) Schematic of the mitochondrial ETC, indicating forward and reverse (RET) electron transport. Dimethyl malonate (DMM) inhibits mitochondrial complex II, preventing succinate oxidation and linked electron transport in either direction. c) OCR in BMDMs after 3 hours treatment with or without 10 mM DMM (Untreated: N = 11; 10 mM DMM: N = 6). d) Heatmap of significantly altered metabolites in BMDMs treated with DMM (10 mM) with or without LPS (100 ng/mL) for 4 hours. The relative abundance of each metabolite is depicted as z score across rows (red, high; blue, low). (N = 5, each treatment). e) Succinate concentration (AU, arbitrary units) in WT BMDMs with or without treatment with LPS (100 ng/mL) and DMM (10 mM) for 4 h (N = 5, each treatment). f) $NAD^+$/NADH ratio in BMDMs after 4 hours treatment with or without LPS (100 ng/mL), with or without DMM (10 mM) (N = 4, each treatment). Data are means +/− SEM. * $p < 0.05$, two-tailed t-test (**c** * $p < 0.0001$), one-way ANOVA with Tukey test for multiple comparisons (**e** * $p = 0.0057$ UT/DMM, * $p = 0.0205$ LPS/LPS + DMM), or one-way analysis of variance (ANOVA) with Fisher's LSD (d). N indicates number of individual mice. Parts of this figure were created with BioRender.com.

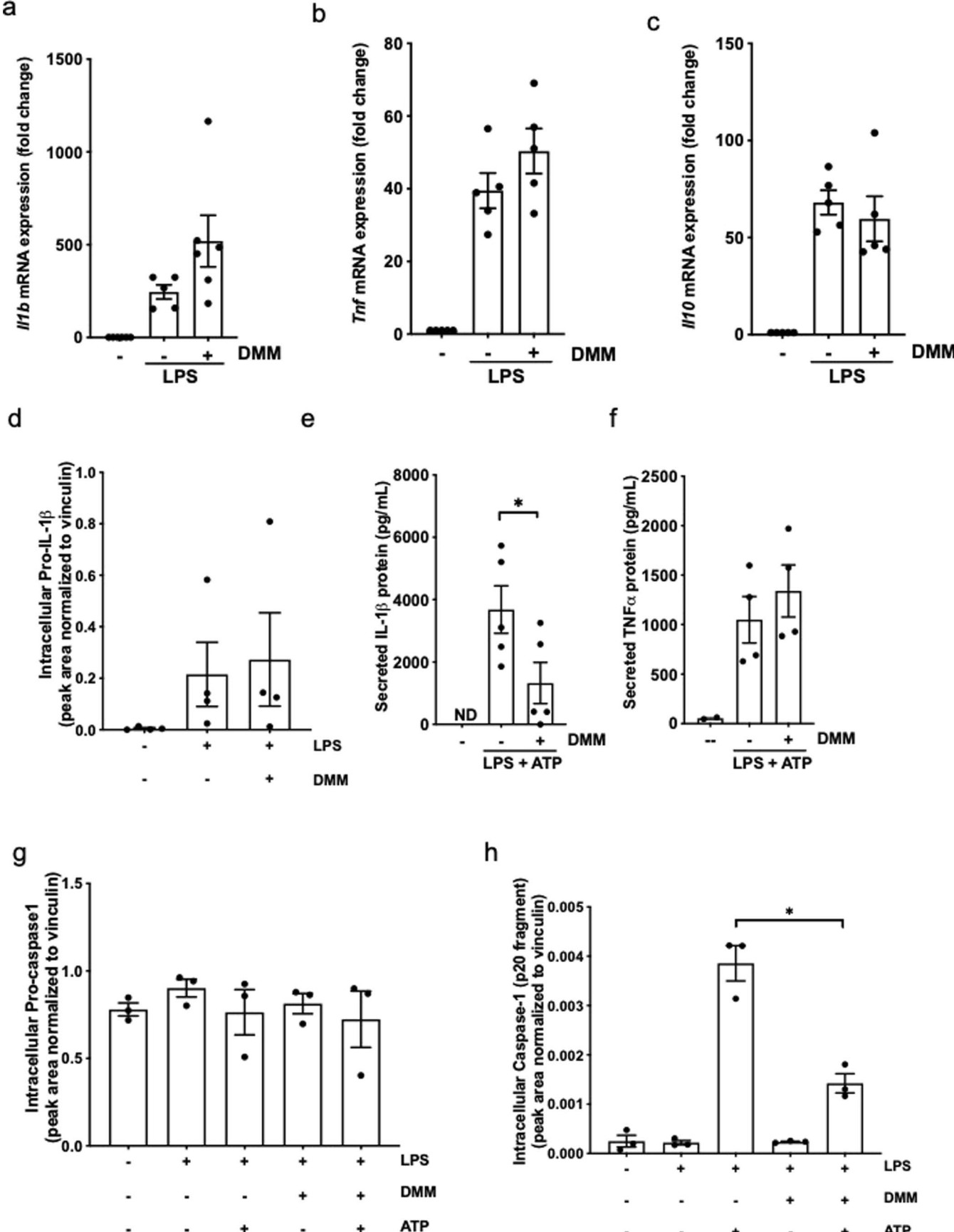

**Extended Data Fig. 2 | See next page for caption.**

**Extended Data Fig. 2 | Dimethyl malonate (DMM) decreases NLRP3 inflammasome activation.** a) *Il1b* mRNA expression ($\Delta\Delta C_t$) in BMDMs treated with LPS (100 ng/mL) for 4 hours, with or without DMM (10 mM) (N = 5 UT, LPS; N = 6 LPS + DMM). b) *Tnf* mRNA expression ($\Delta\Delta C_t$) in BMDMs treated as in a (N = 5 for all treatments). c) *Il10* mRNA expression ($\Delta\Delta C_t$) in BMDMs treated as in a (N = 5 for all treatments). d) Pro-IL-1β protein levels in cell lysates of BMDMs treated with LPS (100 ng/mL), with or without DMM (10 mM). (N = 4 for all treatments). e) IL-1β protein levels in cell culture supernatant from BMDMs treated with LPS (100 ng/mL) and ATP (5 mM), with or without DMM (10 mM). (N = 5 for all treatments). f) TNFα protein levels in cell culture supernatant from BMDMs treated as in e. (N = 3 for all treatments). g) Intracellular pro-caspase-1 protein expression in cell lysates from BMDMs treated with or without LPS (100 ng/mL) and ATP (5 mM), with or without DMM (10 mM). (N = 3 for all treatments). h) Intracellular caspase-1 (p20 fragment) protein expression in cell lysates from BMDMs treated with or without LPS (100 ng/mL) and ATP (5 mM), with or without DMM (10 mM) (N = 3 for all treatments). Data are means +/− SEM. * $p < 0.05$, two-tailed t-test (**e** *$p = 0.0001$), one-way analysis of variance (ANOVA) with a Tukey test for multiple comparisons (**h** *$p < 0.0001$).

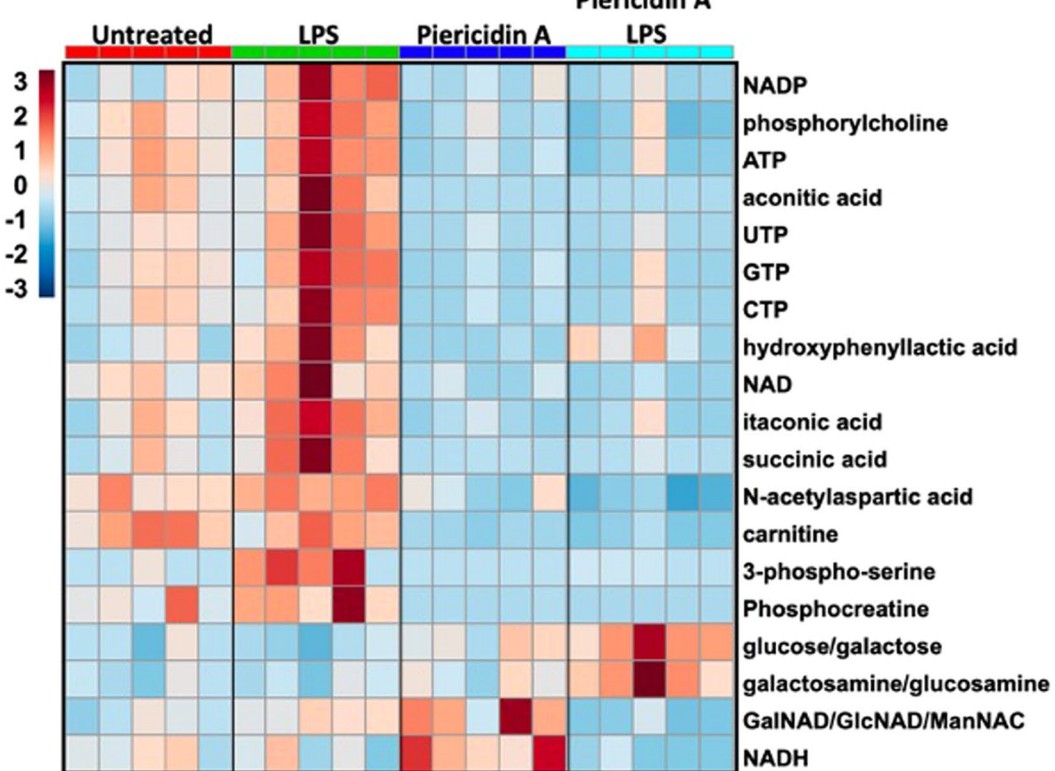

**Extended Data Fig. 3 | See next page for caption.**

**Extended Data Fig. 3 | Piericidin A inhibits mitochondrial complex I.** a) Schematic of the mitochondrial ETC, indicating forward and reverse (RET) electron transport. Piericidin A inhibits mitochondrial complex I, preventing proton pumping, superoxide production, and both forward and reverse electron transport. b) OCR in BMDMs after 1-hour treatment with or without 100 nM or 500 nM piericidin A (N = 8 basal; N = 4 100 nM; N = 6 500 nM). c) NAD$^+$/NADH ratio in BMDMs after 4-hour treatment with or without LPS (100 ng/mL), with or without piericidin A (500 nM) (N = 4 for each condition). d) Heatmap of significantly altered metabolites in BMDMs treated with piericidin A (500 nM) with or without LPS (100 ng/mL) for 4 hours. The relative abundance of each metabolite is depicted as z score across rows (red, high; blue, low). (N = 5 for each condition). e) Succinate concentration (AU, arbitrary units) in WT BMDMs with or without treatment with LPS (100 ng/mL), with or without piericidin A (500 nM) for 4 hours (N = 5, for each condition). Data are means +/− SEM. * p < 0.05, one-way ANOVA with Tukey test for multiple comparisons (**b** *p < 0.0001; **c** *p = 0.0442 UT/Piericidin A, *p = 0.0426 LPS/LPS + Piericidin A; **e** *p = 0.0258 UT/LPS, *p = 0.0043 LPS/LPS + Piericidin), or one-way ANOVA with Fisher's LSD (d). Parts of this figure were created with BioRender.com.

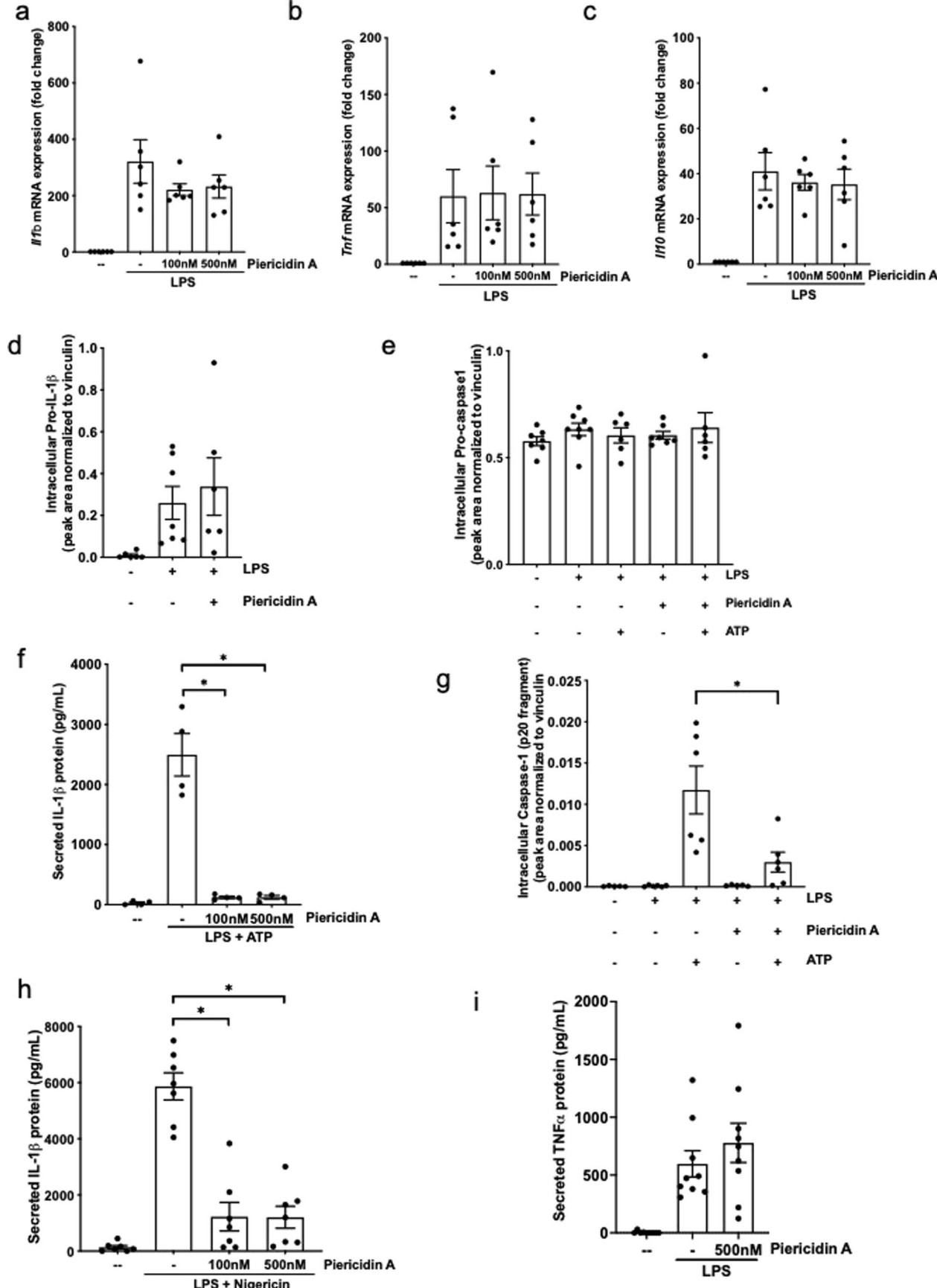

**Extended Data Fig. 4 | See next page for caption.**

**Extended Data Fig. 4 | Piericidin A decreases NLRP3 inflammasome activation.** a) *Il1b* mRNA expression ($\Delta\Delta C_t$) in BMDMs treated with or without LPS (100 ng/mL) for 4 hours, with or without piericidin A (100 nM, 500 nM) (N = 6 for each condition). b) *Tnf* expression ($\Delta\Delta C_t$) in BMDMs treated as in a (N = 5 for all conditions). c) *Il10* mRNA expression ($\Delta\Delta C_t$) in BMDMs treated as in a (N = 5 for all conditions). d) Pro-IL-1β protein expression in cell lysates from BMDMs treated with or without LPS (100 ng/mL), with or without piericidin A (100 nM). (N = 7 LPS; N = 6 UT, LPS + piericidin A; N = 4 LPS + ATP, LPS + piericidin A + ATP). e) Intracellular pro-caspase-1 protein expression in cell lysates from BMDMs treated with or without LPS (100 ng/mL) and ATP (5 mM), with or without piericidin A (100 nM). (N = 8 LPS; N = 7 UT, LPS + piericidin A; N = 6 LPS + ATP, LPS + piericidin A). f) IL-1β protein levels in cell culture supernatant treated with or without LPS (100 ng/mL) and ATP (5 mM), with or without piericidin A (100 nM, 500 nM). (N = 4 for each condition). g) Intracellular caspase-1 (p20 fragment) protein expression in cell lysates from BMDMs treated with or without LPS (100 ng/mL) and ATP (5 mM), with or without piericidin A (100 nM). (N = 6 LPS, LPS + ATP, LPS + piericidin A + ATP; N = 5 UT, LPS + piericidin A.). h) IL-1β protein levels in cell culture supernatant from BMDMs treated with or without LPS (100 ng/mL) and Nigericin (20 μM), with or without piericidin A (100 nM, 500 nM). Subsequently, (N = 7 for each condition). i) TNFα protein levels in cell culture supernatant from BMDMs treated as in H. (N = 9 for all conditions). Data are means +/− SEM. * p < 0.05, one-way ANOVA with Tukey test for multiple comparisons (**f** *p < 0.0001; **g** *p = 0.0027; **h** *p < 0.0001).

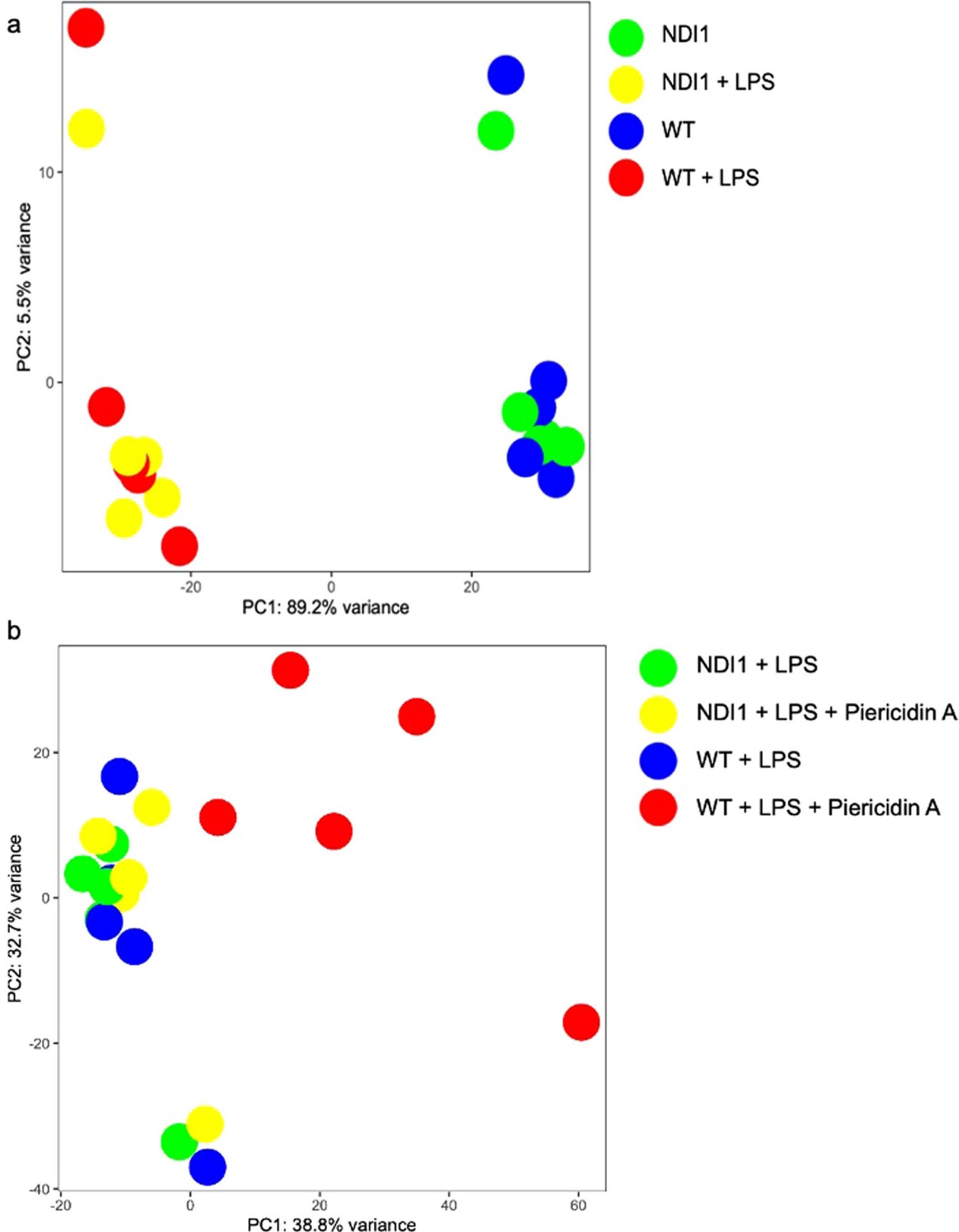

**Extended Data Fig. 5 | See next page for caption.**

**Extended Data Fig. 5 | Piericidin A inhibits mitochondrial complex I to modulate LPS-dependent mRNA expression.** a) Principal component analysis of RNASeq data on WT and NDI1 BMDMs treated, or not, with LPS (100 ng/mL) for 4 hours. Each dot represents RNASeq data from a single sample. (N = 5 for each treatment). b) Principal component analysis of RNASeq data on WT and NDI1 BMDMs treated with LPS (100 ng/mL) for 4 hours, with or without piericidin A (500 nM). Each dot represents RNASeq from a single sample. (N = 5 for each treatment).

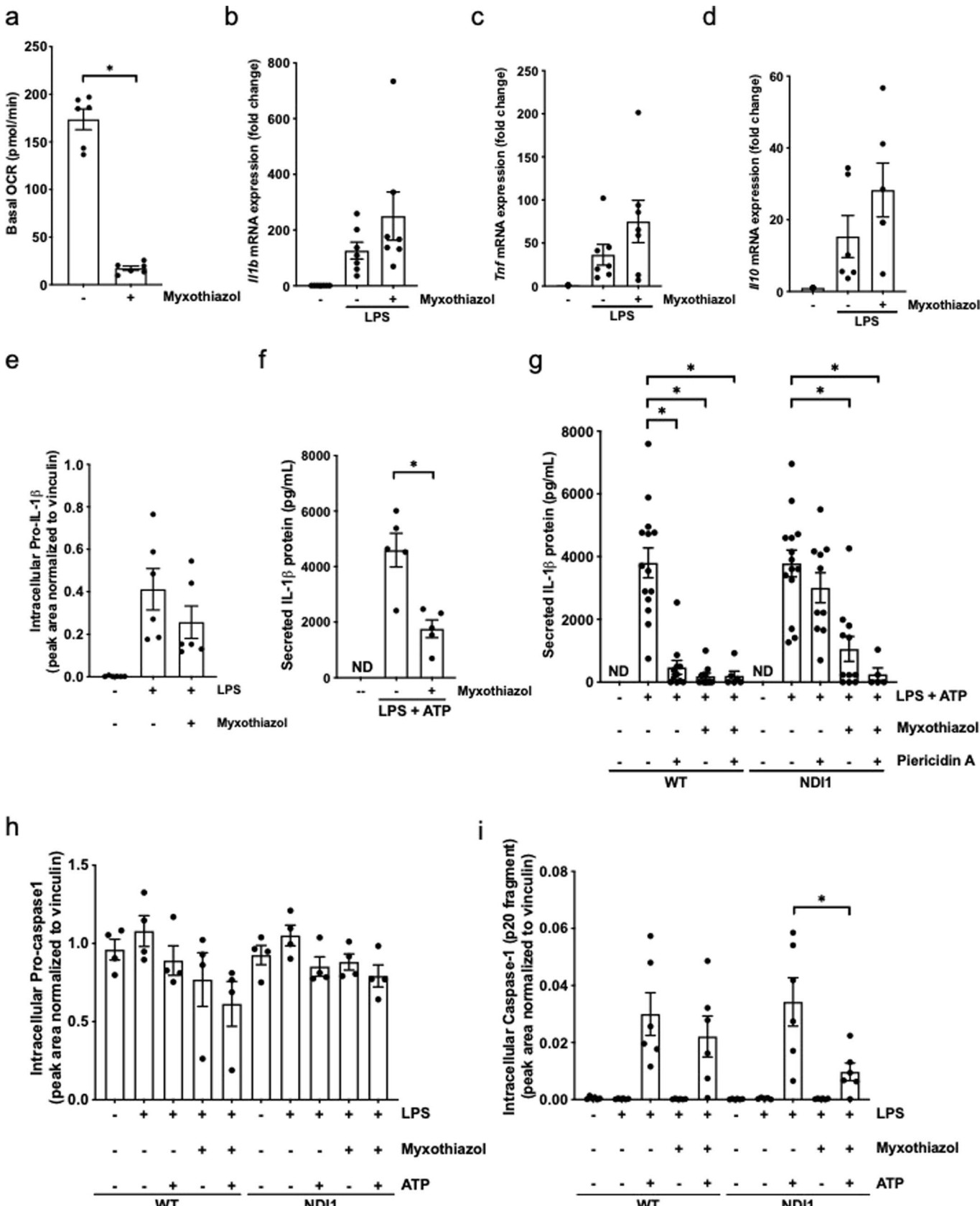

**Extended Data Fig. 6 | See next page for caption.**

**Extended Data Fig. 6 | Mitochondrial complex III inhibitor myxothiazol decreases NLRP3 inflammasome activation.** a) OCR in BMDMs with or without 1-hour treatment with 100 nM myxothiazol (N = 6 for each treatment). b) *Il1b* mRNA expression ($\Delta\Delta C_t$) in BMDMs treated with or without LPS (100 ng/mL), with or without myxothiazol (100 nM) (N = 7 for each treatment). c) *Tnf* mRNA expression ($\Delta\Delta C_t$) in BMDMs treated as in a (N = 7 for each treatment). d) *Il10* mRNA expression ($\Delta\Delta C_t$) in BMDMs treated as in a (N = 6 UT, LPS; N = 5 LPS + myxothiazol). e) Pro-IL-1β protein expression in cell lysates from WT and NDI1 BMDMs treated with LPS (100 ng/mL) and with or without myxothiazol (100 nM) (N = 6 for all treatments). f) IL-1β protein levels in cell culture supernatant from BMDMs treated with LPS (100 ng/mL) for 5.5 hours, with or without myxothiazol (100 nM and ATP (5 mM). (N = 5 for each condition). g) IL-1β protein levels in cell culture supernatant from WT and NDI1 BMDMs treated LPS (100 ng/mL) and ATP (5 mM), with or without myxothiazol (100 nM). (N = 13 both genotypes LPS; N = 10 both genotypes LPS + ATP, LPS + ATP + piericidin A; N = 6 for both genotypes LPS + ATP + myxothiazol + piericidin A). h) Intracellular Pro-caspase-1 protein expression in cell lysates from WT and NDI1 BMDMs treated with LPS (100 ng/mL), with or without myxothiazol (100 nM) and ATP (5 mM)(N = 4 for each treatment and genotype). i) Intracellular caspase-1 (p20 fragment) protein expression in cell lysates from WT and NDI1 BMDMs treated as in h (N = 6 for each condition). Data are means +/− SEM. * $p < 0.05$, two-tailed t-test (**a** *$p < 0.0001$; **f** *$p = 0.0033$), or one-way ANOVA with Tukey test for multiple comparisons (**g** *$p < 0.0001$; **i** *$p = 0.0025$).

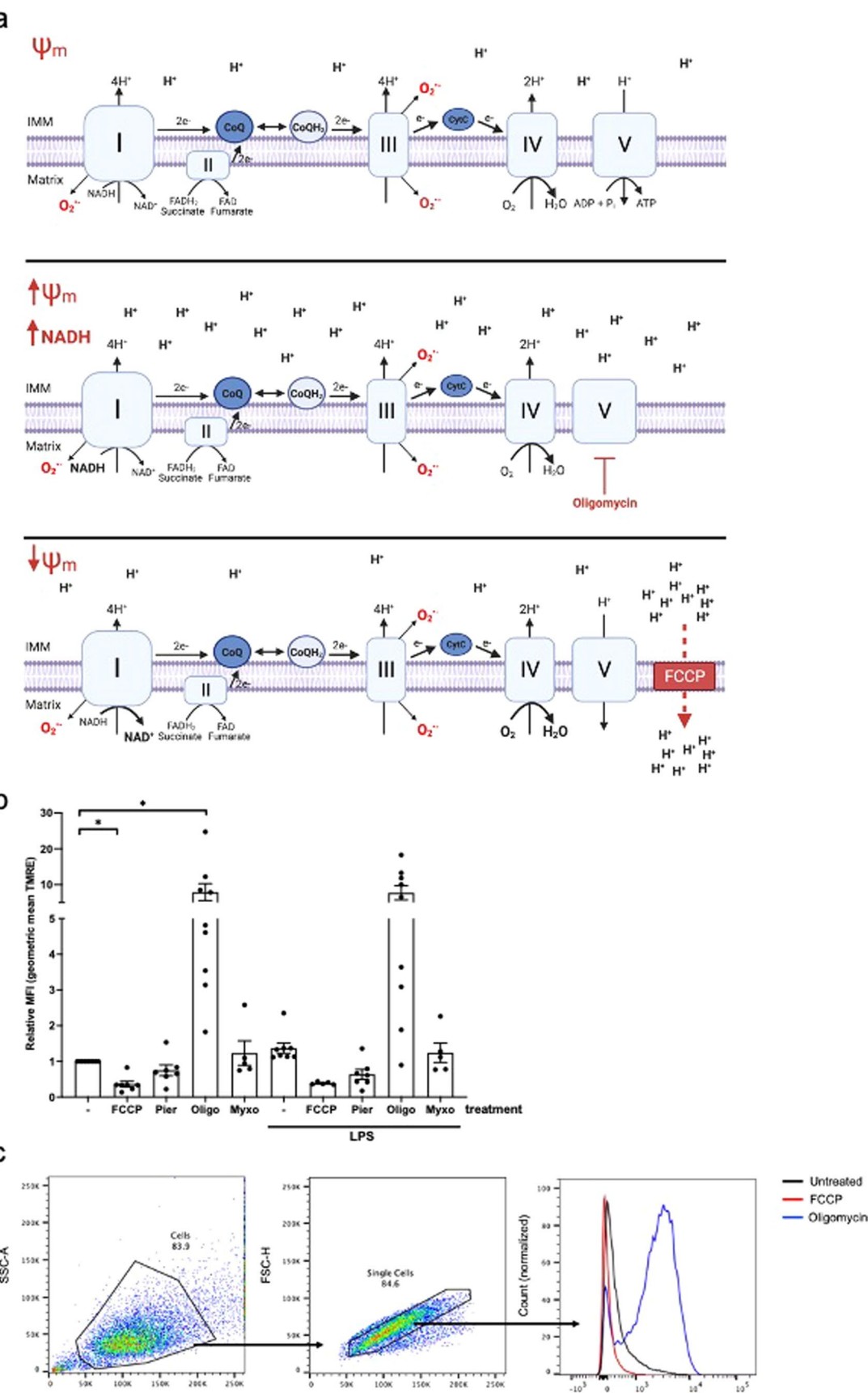

**Extended Data Fig. 7 | See next page for caption.**

**Extended Data Fig. 7 | Oligomycin or FCCP cause an increase or decrease in mitochondrial membrane potential, respectively.** a) Schematic of mitochondrial membrane potential ($\Psi\psi$m) at baseline (top), during in the presence of oligomycin (middle), and in the presence of FCCP (bottom). At baseline, mitochondrial complexes I, III and IV pump protons across the inner mitochondrial membrane to generate and maintain a high membrane potential. Mitochondrial complex V uses this proton motive force to generate ATP from ADP and $_{Pi}$. Oligomycin inhibits mitochondrial complex V, preventing the passage of protons through complex V into the mitochondrial matrix. This causes an increase in the membrane potential as protons build up in the intermembrane space. FCCP is a protonophore and allows for the free passage of protons across the inner mitochondrial membrane. This decreases the membrane potential, preventing ATP generation. b) Relative MFI (geometric mean of TMRE stain, relative to UT control) of BMDMs treated with LPS (100 ng/mL), or not, with FCCP (10 µM), piericidin A (Pier) (100 nM), Oligomycin (50 nM), or Myxothiazol (100 nM) (N = 5 Myxo, FCCP + LPS, Myxo + LPS; N = 7 FCCP, Pier, Pier + LPS; N = 9 Oligo, Oligo + LPS, UT). c) Example gating strategy for b with representative histograms of untreated, FCCP treated, and oligomycin treated BMDMs. Cell counts are standardized to mode. Data are means +/− SEM. * p < 0.05, one-sample t-test (**b** *p < 0.0001 UT/FCCP, p = 0.0116 UT/Oligo). Parts of this figure were created with BioRender.com.

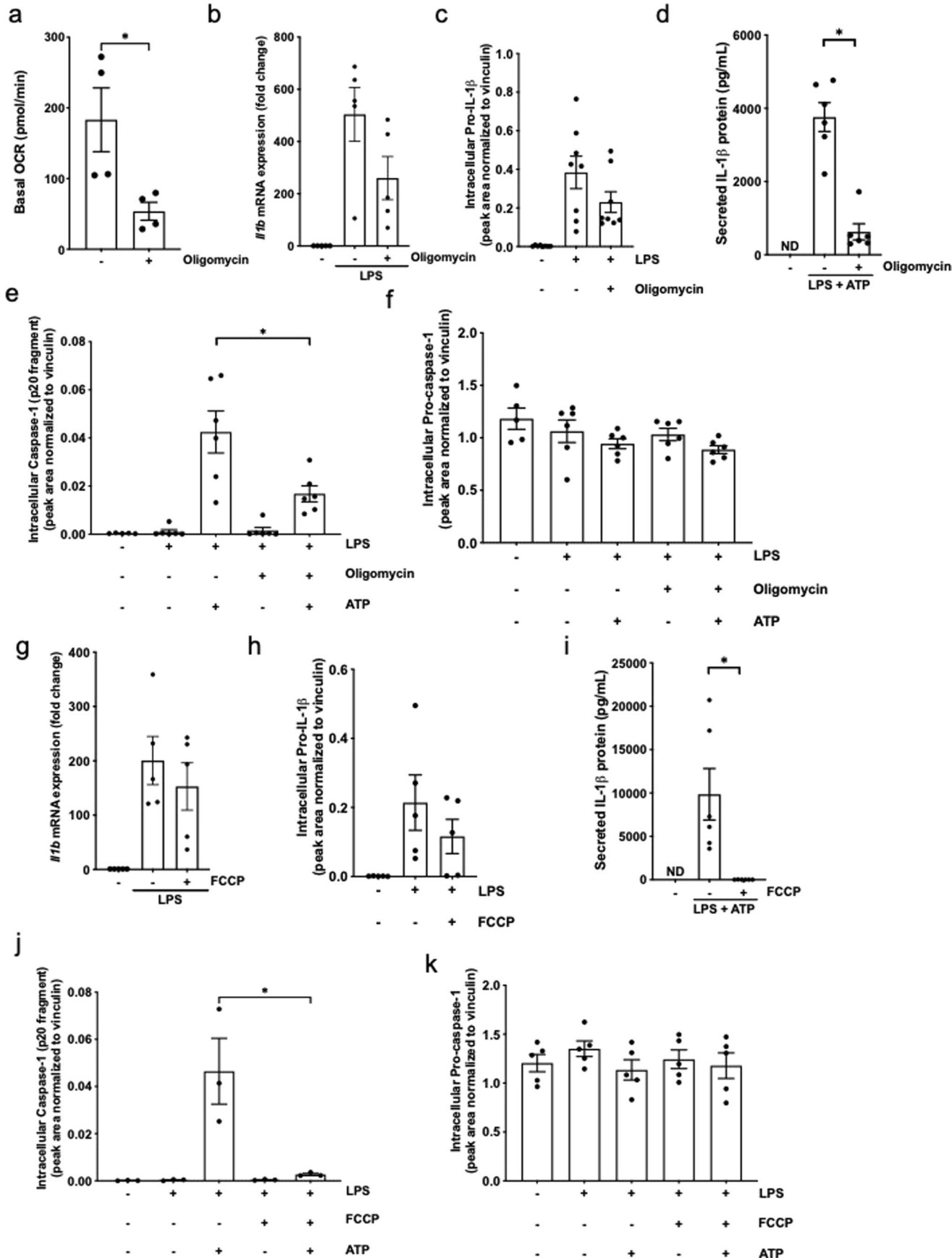

**Extended Data Fig. 8 | See next page for caption.**

**Extended Data Fig. 8 | Oligomycin and FCCP decrease NLRP3 inflammasome activation.** a) OCR in BMDMs from WT mice after one-hour treatment with or without oligomycin (50 nM) (N = 4, each treatment). b) *Il1b* mRNA expression ($\Delta\Delta C_t$) in BMDMs treated with or without LPS (100 ng/mL) for 4 hours with or without oligomycin (50 nM) (N = 5 for each treatment). c) Pro-IL-1β protein expression in cell lysates from BMDMs treated with LPS (100 ng/mL) for 5.5 hours, with or without oligomycin (50 nM). (N = 8 for each treatment). d) IL-1β protein levels in cell culture supernatant of BMDMs treated with LPS (100 ng/mL) and ATP (5 mM), with or without oligomycin (50 nM). (N = 6 for each condition). e) Intracellular caspase-1 protein expression in cell lysates from from BMDMs treated with LPS (100 ng/mL) and ATP (5 mM), with or without oligomycin (50 nM). (N = 6 for each condition). f) Intracellular pro-caspase-1 protein expression in cell lysates from BMDMs treated as in e. (N = 6 for each condition). g) *Il1b* mRNA expression ($\Delta\Delta C_t$) in BMDMs treated with or without LPS (100 ng/mL) for 4 hours with or without FCCP (10μM) (N = 5 for each condition). h) Pro-IL-1β protein expression in cell lysates from BMDMs treated with LPS (100 ng/mL) for 5.5 hours, with or without FCCP (10μM). (N = 5 for each treatment). i) IL-1β protein levels in cell culture supernatant of BMDMs treated with LPS (100 ng/mL) and ATP (5 mM), with or without FCCP (10μM). (N = 6 for both treatments). j) Intracellular caspase-1 (p20 fragment) protein expression in cell lysates from BMDMs treated with LPS (100 ng/mL) and ATP (5 mM), with or without FCCP (10μM). (N = 6 for all treatments). Data are means +/− SEM. *p < 0.05, two-tailed t-test (**a** *p = 0.0328; **d** *p < 0.0001; **i** *p = 0.0077), one-way ANOVA with Tukey test for multiple comparisons (**e** *p = 0.0026; **j** *p = 0.0041).

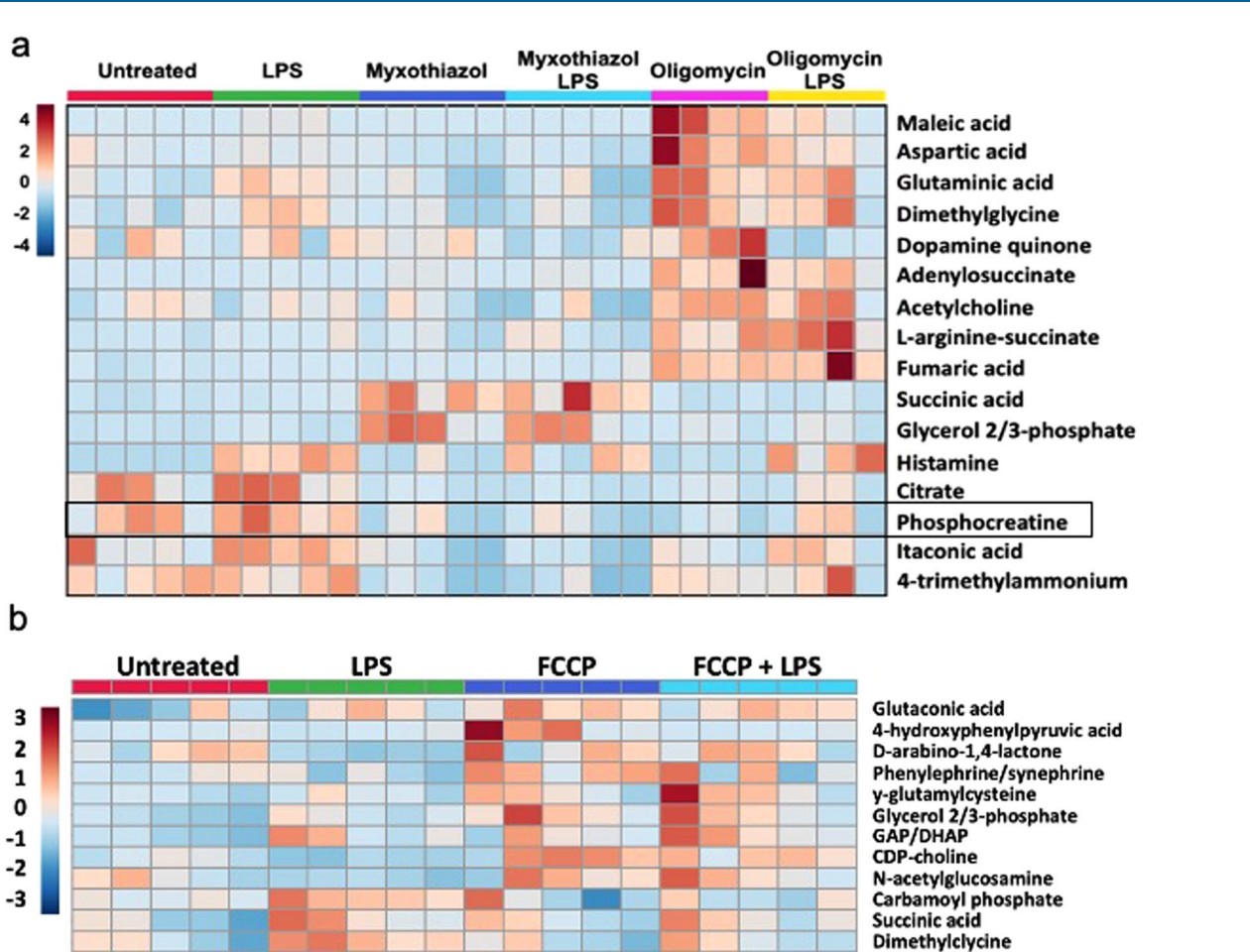

**Extended Data Fig. 9 | See next page for caption.**

**Extended Data Fig. 9 | Mitochondrial ETC inhibitors decrease phosphocreatine levels.** a) Heatmap of top 16 altered metabolites in BMDMs treated with LPS (100 ng/mL) for 4 hours with or without myxothiazol (100 nM) or oligomycin (50 nM). The relative abundance of each metabolite is depicted as z score across rows (red, high; blue, low). (N = 5 UT, LPS, Myxothiazol, Myxothiazol + LPS; N = 4 Oligomycin, Oligomycin + LPS). b) Heatmap of top 50 altered metabolites in BMDMs treated with LPS (100 ng/mL) with or without FCCP (10 μM) for 4 hours. The relative abundance of each metabolite is depicted as z score across rows (red, high; blue, low). (N = 5 or all treatments).

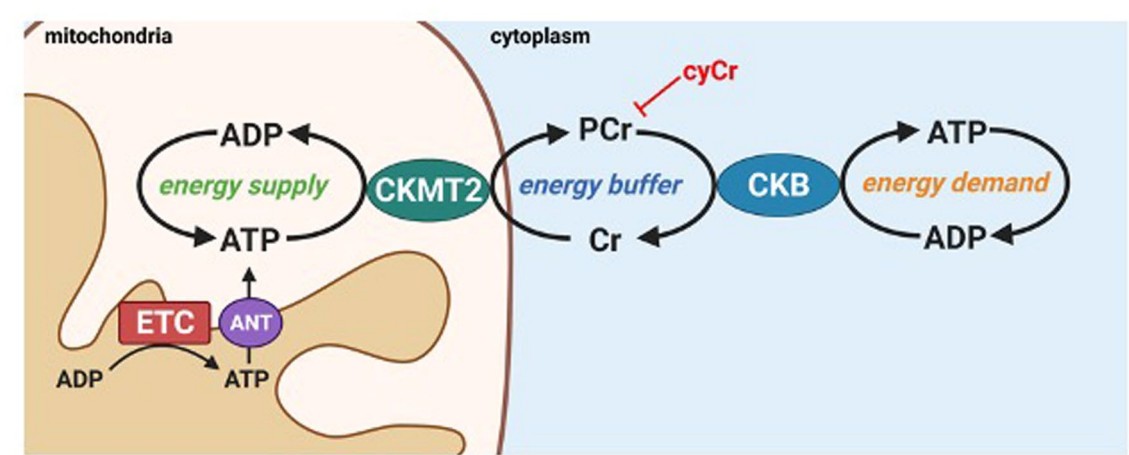

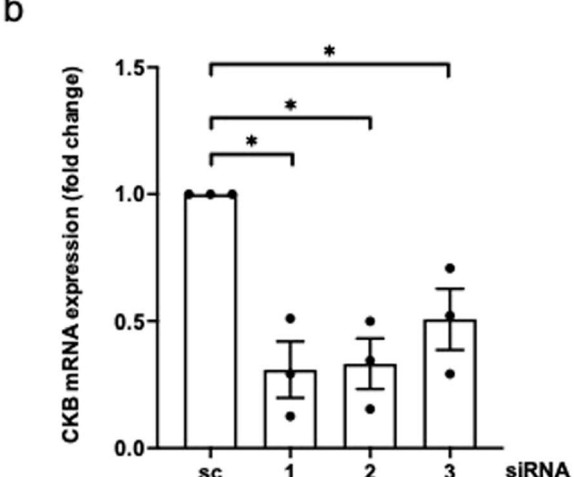

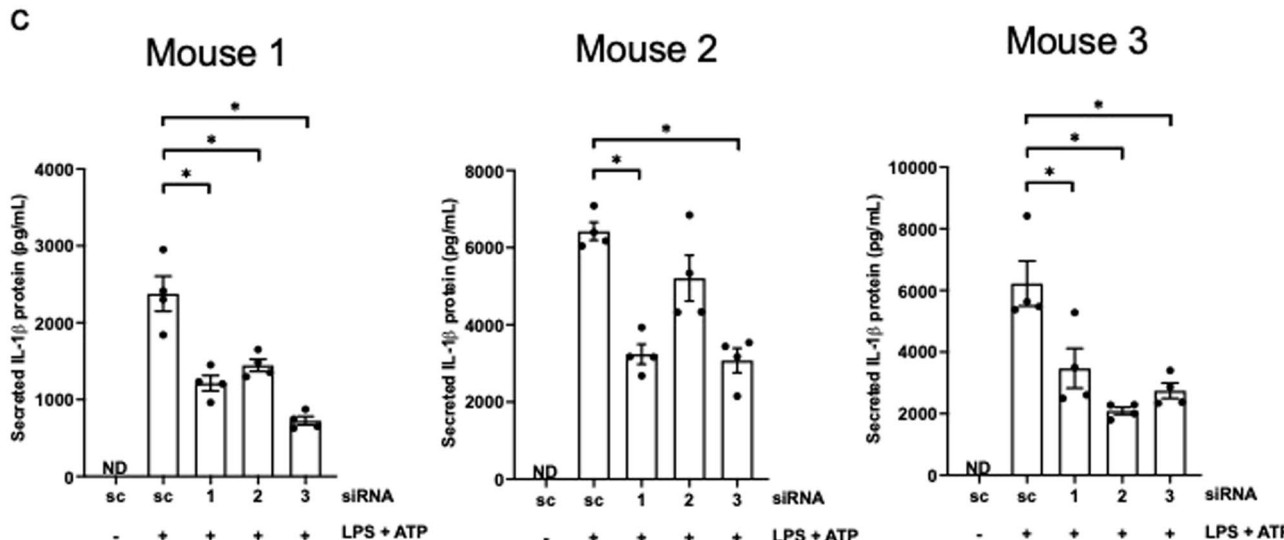

**Extended Data Fig. 10 | See next page for caption.**

**Extended Data Fig. 10 | Creatine kinase (CKB) RNAi decreases secretion of IL-1β protein levels in response to LPS plus ATP.** a) Schematic of the phosphocreatine shuttle. A phosphate group from mitochondria-generated ATP is transferred to creatine (Cr) by CKMT2, generating phosphocreatine (PCr) and ADP. PCr is able to cross the mitochondrial membrane to the cytoplasm. Creatine kinase (CKB) transfers its phosphate group to ADP to generate ATP to meet cellular energetic demands. In this way the generation of PCr provides an energy buffer to quickly generate ATP. Cyclocreatine (cyCr) disrupts this buffer. CyCr is phosphorylated by CKB to produce phosphocyclocreatine, a poor donor of phosphate to ADP for generation of ATP. b) *Ckb* mRNA expression ($\Delta\Delta C_t$) in BMDMs transected with scramble siRNA control (sc) or siRNA against *Ckb* (N = 3, N are biological replicates from 3 independent experiments). c) IL-1β protein concentration in cell culture supernatant of BMDMs transfected with vector control or siRNA against *Ckb* treated with LPS (100 ng/mL) and ATP (5 mM). Each panel represents an independent experiment with N = 4 technical replicates. Data are means +/− SEM. * $p < 0.05$, one-way ANOVA with Tukey test for multiple comparisons (**b** *p = 0.0041 sc/1, *p = 0.0051 sc/2, *p = 0.0274 sc/3; **c** mouse 1: *p = 0.0001 sc+LPS/1+LPS, *p = 0.001 sc+LPS/2+LPS, *p < 0.0001 sc+LPS/3+LPS; mouse 2: *p = 0.0002 sc+LPS/2+LPS, *p = 0.0001 sc+LPS/3+LPS; mouse 3: *p = 0.0062 sc+LPS/1+LPS, *p = 0.0003 sc+LPS/2+LPS, *p = 0.0011 sc+LPS/3+LPS). Parts of this figure were created with BioRender.com.

# Reporting Summary

## Statistics

For all statistical analyses, confirm that the following items are present in the figure legend, table legend, main text, or Methods section.

| n/a | Confirmed | |
|---|---|---|
| ☐ | ☒ | The exact sample size (*n*) for each experimental group/condition, given as a discrete number and unit of measurement |
| ☐ | ☒ | A statement on whether measurements were taken from distinct samples or whether the same sample was measured repeatedly |
| ☐ | ☒ | The statistical test(s) used AND whether they are one- or two-sided *Only common tests should be described solely by name; describe more complex techniques in the Methods section.* |
| ☐ | ☒ | A description of all covariates tested |
| ☐ | ☒ | A description of any assumptions or corrections, such as tests of normality and adjustment for multiple comparisons |
| ☐ | ☒ | A full description of the statistical parameters including central tendency (e.g. means) or other basic estimates (e.g. regression coefficient) AND variation (e.g. standard deviation) or associated estimates of uncertainty (e.g. confidence intervals) |
| ☐ | ☒ | For null hypothesis testing, the test statistic (e.g. *F*, *t*, *r*) with confidence intervals, effect sizes, degrees of freedom and *P* value noted *Give P values as exact values whenever suitable.* |
| ☒ | ☐ | For Bayesian analysis, information on the choice of priors and Markov chain Monte Carlo settings |
| ☒ | ☐ | For hierarchical and complex designs, identification of the appropriate level for tests and full reporting of outcomes |
| ☒ | ☐ | Estimates of effect sizes (e.g. Cohen's *d*, Pearson's *r*), indicating how they were calculated |

*Our web collection on statistics for biologists contains articles on many of the points above.*

## Software and code

Policy information about availability of computer code

**Data collection**
Oxygen consumption data was collected using Wave 2.4 software. Flow cytometry data was collected using FACS DIVA 8.0.3 software. Metabolite data was collected using Xclibur 4.1 software. Immunoblot data were collected using a Wes by ProteinSimple using Compass for SW software 5.0.1. RTPCR data was collected using CFX Manager (version 3.1) by Bio-Rad. RNASeq data was collected using Illumina NextSeq 500 system Raw BCL read files were demultiplexed and converted to FASTQ files using bcl2fastq (Illumina) and trimmed using Trimmomatic (version 0.39). Fluorescence data from the $H_2O_2$ data and ATP assay and colorometric data from BCA and ELISA were collected using SpectraMax M2 (Molecular Devices) and SoftMax Pro (Version 6.4). For metabolomics, high-resolution HPLC–tandem mass spectrometry was performed on a Q-Exactive (ThermoFisher Scientific) in line with an electrospray source and an UltiMate 3000 (ThermoFisher Scientific) and data were collected using Xcalibur 4.1 software.

**Data analysis**
GraphPad Prism 9.0 and MetaboAnalyst 4.0 were used for statistical tests. RNASeq data was analyzed using the R package edgeR6. Metabolite data was analyzed using Tracefinder 4.1 software. Immunoblot data were analyzed using Compass for SW software 5.0.1 (ProteinSimple). Flow cytometry data was analyzed using Flowjo 10.4.2. For RNASeq, reads were then aligned to the mouse mm10 reference genome using STAR to generate BAM files3, HTSeq was used to count reads in the exons of genes, and likelihood ratio tests for all samples and all detected transcripts and pairwise differential gene expression analyses were carried out using the R package DESeq2.

For manuscripts utilizing custom algorithms or software that are central to the research but not yet described in published literature, software must be made available to editors and reviewers. We strongly encourage code deposition in a community repository (e.g. GitHub). See the Nature Portfolio guidelines for submitting code & software for further information.

## Data

Policy information about availability of data

All manuscripts must include a data availability statement. This statement should provide the following information, where applicable:
- Accession codes, unique identifiers, or web links for publicly available datasets
- A description of any restrictions on data availability
- For clinical datasets or third party data, please ensure that the statement adheres to our policy

All data from the manuscript are available from the corresponding author on request. Source data are provided with this paper. Rdata related to this paper is available on the GEO repository (accession number GSE197606)

# Field-specific reporting

Please select the one below that is the best fit for your research. If you are not sure, read the appropriate sections before making your selection.

☒ Life sciences          ☐ Behavioural & social sciences          ☐ Ecological, evolutionary & environmental sciences

For a reference copy of the document with all sections, see nature.com/documents/nr-reporting-summary-flat.pdf

# Life sciences study design

All studies must disclose on these points even when the disclosure is negative.

| | |
|---|---|
| Sample size | All experiments were performed using sample sizes based on standard protocols in the field. We made every effort to avoid excessive or needless use of animals. No statistical tests were used to predetermine sample sizes. We used sample sizes commonly used in literature in the field (Weinberg 2019, Mills 2016, Coll 2015). We used statistical analysis consistent with the sample size for each experiment and found sufficient statistical power with the sample sizes used in our study. |
| Data exclusions | Outliers were determined using the ROUT method, Q = 1%. Experiments were excluded from analysis if the controls did not work; data from successfully completed experiments were not excluded unless they were determined to be outliers. |
| Replication | All experimental data were reliable reproduced in multiple independent experiments as indicated in the figure legends. For in vivo experiments, multiple mice were used in at least two independent cohorts to ensure reproducibility. |
| Randomization | Transgenic mice were predetermined by mouse genotype and therefore could not be randomized. C57Bl/6J mice were randomly assigned to treatment and control groups. All mice were age-matched and littermates. |
| Blinding | Investigators were not blinded. Blinding was not possible as predominately one person was responsible for performing each experiment and carrying out data analysis. |

# Reporting for specific materials, systems and methods

We require information from authors about some types of materials, experimental systems and methods used in many studies. Here, indicate whether each material, system or method listed is relevant to your study. If you are not sure if a list item applies to your research, read the appropriate section before selecting a response.

### Materials & experimental systems

| n/a | Involved in the study |
|---|---|
| ☐ | ☒ Antibodies |
| ☒ | ☐ Eukaryotic cell lines |
| ☒ | ☐ Palaeontology and archaeology |
| ☐ | ☒ Animals and other organisms |
| ☒ | ☐ Human research participants |
| ☒ | ☐ Clinical data |
| ☒ | ☐ Dual use research of concern |

### Methods

| n/a | Involved in the study |
|---|---|
| ☒ | ☐ ChIP-seq |
| ☐ | ☒ Flow cytometry |
| ☒ | ☐ MRI-based neuroimaging |

## Antibodies

| | |
|---|---|
| Antibodies used | Antibodies used for immunoblot: anti-Vinculin (Cell Signaling, #13901, clone E1E9V; 1:500 dilution); anti -IL-1beta (R&D systems, #AF-401-NA; 1:200 dilution); anti-caspase-1 (p20)(Adipogen, AG-20B-0042-C100, clone Casper-1; 1:250 dilution); anti-ASC (Novus Biologics, NBP1-78977SS; 1:50 dilution); anti-NLRP3 (Novus Biologics, NBP2-03948SS, clone 25N10E9; 1:100 dilution) |
| Validation | The antibodies used in this study were tested by the manufacturer |

| Validation | -anti-Vinculin (Cell Signaling, #13901, clone E1E9V). This antibody can be found in 85 citations. The manufacturer also provides antibody testing data: https://www.cellsignal.com/products/primary-antibodies/vinculin-e1e9v-xp-rabbit-mab/13901.<br><br>-anti-IL-1beta (R&D systems, #AF-401-NA). This antibody can be found in 150 citations. The manufacturer also provides antibody testing data: https://www.rndsystems.com/products/mouse-il-1beta-il-1f2-antibody_af-401-na<br><br>-anti-Caspase-1 (p☐(clone Casper-1, Adipogen; catalog number AG-20B-0042-C100). This antibody can be found in 28 citations. The manufacturer also provides antibody testing data: https://adipogen.com/ag-20b-0042-anti-caspase-1-p20-mouse-mab-casper-1.html<br><br>-anti-ASC (Novus Biologics, NBP1-78977SS). This antibody can be found in 26 publications. The manufactucurer also provides antibody testing data: https://www.novusbio.com/products/asc-tms1-antibody_nbp1-78977#supportresearch<br><br>-anti-NLRP3 (Nobus Biologics, NBP2-03948SS, clone 25N10E9). The antibody can be found in 1 publication. The manufactuer also provides antibody testing data (https://www.novusbio.com/products/nlrp3-nalp3-antibody-25n10e9_nbp2-03948#protocols-faqs) |
|---|---|

# Animals and other organisms

Policy information about studies involving animals; ARRIVE guidelines recommended for reporting animal research

| Laboratory animals | Both male and female C57Bl/6J mice were used. Mice were 8-14 weeks old. Rosa26NDI1-lsl/wt and Rosa26AOX-lsl/wt mice were mixed C57Bl/6 J/N. Rosa26NDI1-lsl/wt mice were genotyped using the following primers: Rosa26 Fwd 5' – GAGTTCTCTGCTGCCTCCTG; Rosa26 Rev 5' – CCGACAAAACCGAAAATCTG; and WPRE B Fwd 5' – GACGAGTCGGATCTCCCTTT. Rosa26AOX-lsl/wt mice were genotyped using the following primers: AOX lsl Fwd 5'-GCGATGCAAGATGGAGGGTA-3'; AOX lsl Rev 5'-TGAATCCAACCGTGGTCTCG-3'; Rosa26 Fwd 5'-GACCTCCATCGCGCACTCCG-3; and Rosa26 Rev 5 -CTCCGAGGCGGATCACAAGC-3. VAV-iCre mice were genotyped using the following primers: Fwd 5'-AGATGCCAGGACATCAGGAACCTG-3' and Rev 5' ATCAGCCACACCAGACACAGAGATC-3'. QPC floxed and wildtype alleles were genotyped using the following primers: QPC Fwd 5'-CTTCCGCTCCTCCCGGAAGT; QPC Rev 5'- TTCCCAAACTCGCGGCCCATG. LysM mice were genotyped using the following primers: LysM-Cre 5'- CRECCCAGAAATGCCAGATTACG; LysM-Pro 5'- GCATTGCAGACTAGCTAAAGGCAG; LysM ex-1 5'- GTCCGCCAGGCGGACTCCATAG |
|---|---|
| Wild animals | This study did not involve wild animals |
| Field-collected samples | This study did not involve samples collected from the field. |
| Ethics oversight | All mouse work was done in accordance with Northwestern University Institutional Animal Care and Use Committee (IACUC). |

Note that full information on the approval of the study protocol must also be provided in the manuscript.

# Flow Cytometry

## Plots

Confirm that:

☒ The axis labels state the marker and fluorochrome used (e.g. CD4-FITC).

☒ The axis scales are clearly visible. Include numbers along axes only for bottom left plot of group (a 'group' is an analysis of identical markers).

☒ All plots are contour plots with outliers or pseudocolor plots.

☒ A numerical value for number of cells or percentage (with statistics) is provided.

## Methodology

| Sample preparation | Sample preparation is described in detail in the methods section of the manuscript.<br><br>BMDMs were plated at 2 million cells/well in a 12-well plate and allowed to adhere overnight. TMRE was added at a concentration of 200nM for 30 minutes. Cells were washed with PBS and removed from the plate with Accutase before resuspension in PBS supplemented with 10% NU-Serum IV. |
|---|---|
| Instrument | BD FACSymphony A5-Laser Anaylzer |
| Software | BD FASC Diva was used for collection of the data. All data was analyzed using FlowJo software. |
| Cell population abundance | Cells were not sorted |
| Gating strategy | Example gating strategy is provided in Ext Fig 6. FSC-A vs SSC-A was used to determine cell populations from debris. From this population, FSC-A vs FSC-H was used to determine single cells. Geometric mean of TMRE was used to determine relative MFI of treated samples compared to untreated samples. |

☒ Tick this box to confirm that a figure exemplifying the gating strategy is provided in the Supplementary Information.

