## [Peer Review File · Nature Immunology]

Peer Review Information

Journal: Nature Immunology

Manuscript Title: Mitochondrial electron transport chain is necessary for NLRP3 inflammasome activation

Corresponding author name(s): Navdeep S. Chandel

Editorial Notes:

Transferred manuscripts This manuscript has been previously reviewed at another journal that is not operating a transparent peer review scheme. This document only contains reviewer comments, rebuttal and decision letters for versions considered at Nature Immunology.

Redactions – published data Parts of this Peer Review File have been redacted as indicated to remove third-party material.

Reviewer Comments & Decisions:

Decision Letter, initial version:
--

Subject: Decision on Nature Immunology submission NI-LE31928-T

Message: 13th Apr 2021

Dear Professor Chandel,

Thanks for transferring your manuscript "Mitochondria-dependent phosphocreatine production sustains NLRP3 inflammasome activation" from Nature along with your 'predictive rebuttal' outlining how you plan to address the reviewer concerns.

We are happy with your proposed plan to enhance the inflammasome end of the paper. Please be aware that we require positive comments from reviewer 3 and reviewer 4, as they are the inflammasome experts here, whereas reviewer 1 and 2 are more mitochondrial metabolism experts. As such, we are pleased that you are prepared to look at other inflammasomes and provide clearer evidence of Nlrp3 activation. One sticking point here is that we are not sure that your response to reviewer 4 regarding the use of imiquimod is going to be met with a positive response. The reviewer was fairly adamant that this is a critical issue, and it is not really clear to us why you cannot provide at least some information about imiquimod response. However, if this is the only thing lacking in a

revised manuscript we would at least return to the reviewers for further comments on the matter. Please note that although we would approach all 4 Nature reviewers, some might not agree to re-review for Nature Immunology. In that event, some mediation would be required from other reviewers.

If you choose to revise your manuscript taking into account all reviewer and editor comments, please highlight all changes in the manuscript text file in Microsoft Word format.

* If you have not done so already please begin to revise your manuscript so that it conforms to our Letter format instructions at <http://www.nature.com/ni/authors/index.html>. Refer also to any guidelines provided in this letter.

The Reporting Summary can be found here:
<https://www.nature.com/documents/nr-reporting-summary.pdf>

You may use the link below to submit your revised manuscript and related files:
[REDACTED]

If you wish to submit a suitably revised manuscript we would hope to receive it within 4 months. If you cannot send it within this time, please let us know. We will be happy to consider your revision so long as nothing similar has been accepted for publication at Nature Immunology or published elsewhere.

Nature Immunology is committed to improving transparency in authorship. As part of our efforts in this direction, we are now requesting that all authors identified as 'corresponding author' on published papers create and link their Open Researcher and Contributor Identifier (ORCID) with their account on the Manuscript Tracking System (MTS), prior to acceptance. ORCID helps the scientific community achieve unambiguous attribution of all scholarly contributions. You can create and link your ORCID from the home page of the MTS by clicking on 'Modify my Springer Nature account'. For more information please visit www.springernature.com/orcid.

Thank you for the opportunity to review your work.

Sincerely,

Nicholas Bernard
Consulting Editor
Nature Immunology

Author Rebuttal to Initial comments

We thank the reviewers for their constructive comments.

We would like to highlight that our paper clarifies the necessity of mitochondrial electron transport chain (ETC) in inflammasome activation. Our work is best put in the context of the current models proposed in the field. This is illustrated in a recent 2021 review in [REDACTED]. Initially, [the] group described the CL097 mechanism [REDACTED]. CL097 works in a K⁺ efflux independent manner while extracellular ATP or nigericin is dependent on K⁺ efflux to activate NLRP3 inflammasome. They suggested that CL097 could potentially inhibit quinone oxidoreductases NQO2 or mitochondrial

complex I to generate ROS. They did not provide any genetic data to support their conclusions. Furthermore, multiple other groups suggested that mitochondrial ROS is a key input NLRP3 inflammasome activation including the late Jurg Tschopp (PMID: 21124315). However, our genetic and

pharmacologic data finds no evidence that mitochondrial ROS is necessary for NLRP3 inflammasome activation by extracellular ATP or CL097. This is a major finding!

Mitochondrial ETC inhibitors are not sufficient to trigger NLRP3 inflammasome activation in LPS-primed BMDMs. Nevertheless, our genetic data does point to the necessity of CL097 inhibition of mitochondrial complex I to activate NLRP3 inflammasome in LPS-primed BMDMs, suggesting that CL097 targets mitochondrial complex I and some other unknown target(s) to activate the NLRP3 inflammasome. Finally, LPS stimulates mitochondrial dependent PCr production that is used to sustain NLRP3 inflammasome activation (see schematic above).

It is important to note that we have provided our NDI1 mice and shared these results with Professor Olaf Groß to further understand how CL097 activates inflammasome. Furthermore, we have shared and discussed the current manuscript with him. He is appropriately acknowledged in the manuscript.

Here we highlight key issues that we addressed to bolster our conclusions:

- (1) We have used nanoparticle-mediated delivery of siRNA against cytosolic creatine kinase B (CKB) enzyme to further bolster our conclusions about the necessity of phosphocreatine to support NLRP3 inflammasome. Note, BMDMs like other primary macrophages are difficult to infect with lentiviruses. We routinely use AAV, retro- and lenti-viruses to deliver genes but have had limited success. Thus, we chose a nanoparticle strategy with the expertise of two new co-authors, Aida Rashidi and Peng Zhang. Also, we demonstrate that cyclocreatine *in vivo* diminishes LPS induction of IL-1b in the serum (Fig. 4i).
- (2) We have clarified our protocol in the text. We stimulate cells for 5.5 hours with LPS followed by addition of extracellular ATP for 30 minutes. We measured intracellular cleavage of caspase-1 (p20 fragment) in cell lysates at 10 minutes after extracellular ATP administration. Note, we don't detect LDH release at 10 minutes after ATP stimulation. At 30 minutes after extracellular ATP administration, we measured secreted IL-1b in the supernatant. Many of the previous studies that examined IL-1b protein levels including Mills et al. Cell 2016 (PMID: 27667687) stimulated BMDMs with LPS for 24 hours (priming) and did not add second signal like extracellular ATP. Thus, ETC inhibition at succinate dehydrogenase (SDH) could result in decreased priming at 24 hours with LPS but our data clearly demonstrate at shorter times of priming the ETC inhibitor do not decrease priming by LPS.

(3) An important conclusion of our study is that glycolysis alone cannot support NLRP3 inflammasome activation. The dogma in the field is that LPS stimulates glycolysis (Warburg Effect) and this is the dominant metabolic pathway. We demonstrated that BMDMs treated with piericidin A during LPS stimulation have high levels of ECAR (glycolytic index) concomitant with reduced OCR. Thus, glycolysis in the absence of mitochondrial ATP is not sufficient to trigger NLRP3 inflammasome activation. Collectively, these data indicate that extracellular ATP activation of NLRP3 inflammasome depends on mitochondrially derived ATP, initially generated by forward respiratory electron flow, and supplied via the phosphocreatine shuttle.

(4) FCCP and oligomycin decrease and increase mitochondrial membrane potential, respectively (Extended Fig. 7; see schematic on the right). Furthermore, FCCP allows ETC to function without generating ATP. FCCP allows efficient NADH oxidation. By contrast, oligomycin decreases both ETC function and ATP generation. Oligomycin inhibition of ETC function results in mitochondrial NADH buildup. Our results indicate that FCCP and oligomycin both decrease extracellular LPS activation of NLRP3 inflammasome without altering LPS priming (Extended Fig. 8). Importantly, they both decrease phosphocreatine levels (Extended Fig. 9). These results indicate that mitochondrial membrane potential or mitochondrial NADH levels are not key inputs in NLRP3 inflammasome activation.

(5) The reviewers had an excellent suggestion about understanding R837 and it is related compound CL097 (Groß et al, Immunity 2016) as NLRP3 inflammasome activators in a K⁺ efflux independent manner. By contrast, the widely used NLRP3 inflammasome activators, extracellular ATP or nigericin, depend on K⁺ efflux. It has been proposed that R837 and CL097 inhibit the quinone oxidoreductases NQO2 and mitochondrial Complex I to trigger ROS production that stimulates NLRP3 inflammasome activation. We tested the necessity of mitochondrial complex I inhibition for CL097 dependent inflammasome activation by using our NDI1 expressing BMDMs. CL097 caused cell death within 20 minutes in an active caspase-1 dependent manner in LPS-primed BMDMs (Fig. 5a). However, CL097 decreased OCR in an active caspase-1 independent manner, indicating that the decrease in OCR was not due to cell death (Fig. 5b). NDI1 expression prevented CL097- or piericidin A-induced decrease in OCR indicating that CL097 indeed inhibits mitochondrial complex I (Fig 5c, Extended Fig. 12a). NDI1 expression prevented CL097-induced secretion of IL-1 β and intracellular caspase-1 cleavage in LPS-primed BMDMs (Fig. 5d, e). CL097 did not alter intracellular pro-caspase-1 levels (Fig. 5f). Next, we tested whether mitochondrial

complex I inhibitor piericidin A or other ETC inhibitors are sufficient to trigger inflammasome like CL097 in LPS primed BMDMs. None of the ETC inhibitors triggered secretion of IL-1 β (Fig. 5g) indicating that CL097, in addition to inhibiting mitochondrial complex I, has other targets that are necessary for NLRP3 inflammasome activation, perhaps endolysosomal effects (PMID: 27692612). Although piericidin A cannot serve as signal 2, we tested whether administration of piericidin A or cyclocreatine during LPS priming would diminish CL097 activation of NLRP3 inflammasome like extracellular ATP. Indeed, both piericidin A and cyclocreatine diminished secreted IL-1 β levels and intracellular caspase-1 cleavage upon CL097 triggering of the NLRP3 inflammasome (Extended Fig. 12b-g). Thus, mitochondrial generated ATP during LPS priming is also necessary for CL097 activation of NLRP3 inflammasome.

CL097 inhibition also triggers ROS production (PMID: 27692612). Thus, we tested whether increasing mitochondrial ROS in NDI1 expressing BMDMs, which are resistant to CL097, would rescue NLRP3 inflammasome activation. Antimycin is a well described generator of mitochondrial superoxide production at complex III. By contrast, myxothiazol inhibits mitochondrial superoxide production at complex III. Indeed, antimycin rescued intracellular caspase-1 cleavage (Fig. 5h, i) and secreted IL-1 β levels (Fig. 5j) in NDI1 expressing LPS-primed BMDMs treated with CL097. Surprisingly, myxothiazol also increased secreted IL-1 β levels in NDI1-expressing LPS-primed BMDMs treated with CL097 (Fig.

j). Moreover, oligomycin and FCCP, which have opposite effects on mitochondrial membrane potential and superoxide production, also increased secreted IL-1 β levels in NDI1-expressing BMDMs primed with LPS and treated with CL097 (Fig. 5j). FCCP, unlike oligomycin and other ETC inhibitors, allows for efficient NAD⁺ regeneration (PMID: 26232225). These results suggest that the rescue effects by ETC inhibitors and FCCP is ROS independent. To directly test whether suppressing or scavenging mitochondrial superoxide could prevent CL097 or extracellular ATP activation of NLRP3 inflammasome, we administered mitochondrial targeted superoxide scavenger, Mito-TEMPO, and superoxide production suppressors S3QEL (S1) or S3QEL (S3). S1 and S3 can suppress mitochondrial complex I or III generated superoxide production, respectively. Neither Mito-Tempo, S1 or S3 prevented CL097 or extracellular ATP activation of NLRP3 inflammasome (Extended Fig. 12h-k). We used MitoTempo, S1 and S3 concentrations that do not inhibit OCR (data not shown) and have previously shown efficacy in other cell systems (PMID: 28842493, PMID: 31627168, PMID: 26368590, PMID: 33148642). Finally, we tested whether increasing ROS production could rescue the secreted IL-1 β levels in NDI1 expressing LPS primed BMDMs treated with CL097. Paraquat, a known generators of superoxide production, failed to increase secreted IL-1 β levels (Extended Fig. 12l). Thus, our data indicate that CL097 requires inhibition of mitochondrial complex I to trigger NLRP3 inflammasome activation through an unidentified mitochondria dependent mechanism (Fig. 5K).

- (6) Michael Karin and colleagues have suggested that the requirement of mitochondria is the release of mtDNA. Initially, they utilized TFAM floxed mice, which we provided. TFAM is necessary for mtDNA replication. Recently they utilized cytidine monophosphate kinase 2 (CMPK2), which is rate limiting for mtDNA synthesis, to further bolster their conclusions around mtDNA synthesis. It is important to note that mtDNA encodes key subunits of ETC. Thus, the loss of TFAM or CMPK2 causes mtDNA depletion resulting in ETC inhibition. Our data does not preclude mtDNA hypothesis but the use of TFAM or other strategies to deplete mtDNA is challenging to interpret as TFAM depletes both ETC proteins and mtDNA.
- (7) Based on these observations we have reworded the title of the manuscript: “Mitochondrial electron transport chain is necessary for NLRP3 inflammasome activation”

Response to Reviewers

Referee #1 (Remarks to the Author):

The precise mechanism of how mitochondrial function contributes to NLRP3 activation following activation of macrophages is of considerable interest and importance. Here the authors use a series of elegant genetic and pharmacological models to nicely dissect out the potential roles of ROS from mitochondria from ATP production from the oxidative phosphorylation machinery. They show nicely using these elegant models that NLRP3 inflammasome activation is independent of mitochondrial ROS production, while many other facets of this signaling pathway remain intact. This is an elegant and important result that will be of great interest to researchers in immunometabolism. However, I feel that this finding is better suited to a more specialized immunology journal than Nature. The authors then go on to address this gap by proposing a mechanism by which mitochondrial function is required for NLRP3 activation by generating phosphocreatine in the IMS which is then exported to the cytosol, there converted back to ATP which is required to activate the inflammasome. I found this section of the work to be quite preliminary and far less solid than the rest of the paper. Below, I have outlined some points related to this for the authors' consideration.

We thank the reviewer positive comments about the use of “elegant models”.

Major points

1 The measurements of ATP by RFU is most ideal. My view is that you need to measure quantify the ATP and also the ADP, and more importantly present these as ATP/ADP ratio. Similarly, the Phosphocreatine/creatine ratio should also be expressed.

We have expressed Phosphocreatine/creatine ratio (Fig. 4a). We do not think ATP/ADP ratio would provide additional insight into the mechanism.

2. The model seems to be that ATP export from mitochondria alone is either not occurring in these cells or is low and has to be supplemented by PCr export. If this is the case, then knocking out either mitochondrial creatine kinase (CK) and/or cytosolic creatine kinase is essential. To me the data showing a drop in CrP may just be a consequence of inhibiting mitochondrial ATP production, and does not indicate the CrP is on the key pathway.

This is a key experiment, and we thank the reviewer for the suggestion. We have used nanoparticle delivered RNAi against cytosolic creatine kinase enzyme CKB in primary BMDMs. Our data demonstrates RNAi against CKB diminishes intracellular caspase-1 cleavage and secreted IL-1b (Fig. 4g, h and Extended Fig. 10).

3. In LPS-stimulated macrophages there is a large upregulation of glycolysis and it is generally thought (whether this is true or not can be argued) that most ATP production is from glycolysis and that mitochondrial function is essential, but that mitochondrial ATP production is not. If that is the case, then it is surprising that oligomycin has an effect on cytosolic ATP levels. What is happening to glycolytic ATP production?

An important conclusion of our study is that glycolysis alone cannot support inflammasome NLRP3 activation. The dogma in the field is that LPS stimulates glycolysis (Warburg Effect) and this is the dominant metabolic pathway (PMID: 27396447). We demonstrate that BMDMs treated with piericidin A during LPS stimulation have high levels of ECAR (glycolytic index) concomitant with reduced OCR, yet they still have reduced inflammasome activation (Extended Fig 11g and Fig. 1d). Thus, glycolysis generated ATP in the absence of mitochondrial ATP is not sufficient to trigger NLRP3 inflammasome activation. Collectively, these data indicate that extracellular ATP activation of NLRP3 inflammasome depends on mitochondrially derived ATP, initially generated by forward respiratory electron flow, and supplied via the phosphocreatine shuttle.

4. The use of cyclocreatine to act as CK substrate that forms cyclocreatine phosphate is not in my view compelling evidence for a role for CrP. It is more likely that cyclocreatine is just acting as a drain on ATP levels forming a dead-end product, cyclocreatine phosphate. On other words, this is acting just as another ATP depleting agent and affecting the inflammasome for that reason. You would probably get a similar result with arsenate for similar reasons. It would have been good to measure the formation of cyclocreatine phosphate. I note that while it is true that creatine phosphate is a better substrate for eth phosphorylation of ADP by CK than cyclocreatine phosphate, it does still occur. This is why cyclocreatine is being explored as a potential therapeutic for subjects with creatine transport deficiencies. This means that the interpretation of the effects of cyclocreatine should be cautious and more controls carried out.

We could not measure the formation of cyclocreatine phosphate as the reviewer suggested by mass spectrometry. However, our RNAi data against cytosolic CKB is consistent with cyclocreatine data.

Referee #2 (Remarks to the Author):

The manuscript by Billingham and colleagues investigated the impact of individual mitochondrial transport chain (ETC) complexes on the NLRP3 inflammasome activation in BMDMs under pro-inflammatory conditions. The authors used chemical inhibitors to impair specific ETC complexes and studied the impact on metabolism and inflammasome activation, specifically IL1b levels. Further, they used genetically engineered cell models expressing *S. cerevisiae* NADH dehydrogenase (NDI1) or alternate oxidase (AOX) and demonstrated that NLRP3 activation is independent of complex I or III ROS production. Finally, they demonstrated that the phosphocreatine (PCr) shuttle is critical for mitochondrial ATP generation and inflammasome activation. The findings may provide useful information on the complex inflammatory response and how altered ETC activity influences NLRP3 inflammasome activation.

The distinct impact on different ETC complexes as well as PCr shuttle on the immune response and NLRP3 inflammasome activation is interesting. I think the demonstration of ATP regeneration as the critical important function of NLRP3 inflammasome activation is well executed. It is important to confirm whether ATP, succinate, NAD, NADPH, or some random amino acid is the key driver given the many studies emerging. However, at some level it leaves us back to where we started – ATP. The experiments are well designed to address these points.

We thank the reviewer's positive comments.

Some points could use clarification.

1. NLRP3 activation experiments were performed with 5mM exogenous ATP supplemented to the media, which I presume is typical. How does this influence energy charge? How does its inclusion impact metabolite levels?
- 2.

The classic stimuli to activate NLRP3 inflammasome is the administration of extracellular ATP that binds to the ionotropic P2X7 receptor. ATP, PCr or any other metabolites with phosphate groups do not cross membranes.

2. In the end, the PCr results are simply about cell activation in a dish rather than the relevant microenvironment (which could change the needs of cells). Could the authors demonstrate the mechanistic link to PCr *in vivo* through starvation or supplementation? This would be an exciting finding and put results in better physiological context. Some stronger interventions on this front (e.g. ANT) would help given its importance to the conclusion.

This is an excellent suggestion. Indeed, we used cyclocreatine *in vivo*. Our results that indicate that cyclocreatine decreases LPS induction of IL-1b in serum.

3. The inflammasome is also activated under hypoxic conditions suggesting a mechanism that is less dependent on mitochondrially derived ATP. Some discussion of the relevance to current findings is warranted.

Hypoxia encountered in tissues is 0.5-1.5% O₂ range. Cell can respire and generate mitochondrial ATP until 0.5%. The km of cytochrome c oxidase is 0.1%. O₂ (PMID: 8702521). Thus, hypoxic cells can generate mitochondrial ATP and respire (PMID: 24301801). Mitochondrial ATP will be able to sustain NLRP3 inflammasome even at low oxygen concentration. The cells would have to be close to 0% O₂ for mitochondrial ATP to be limiting (PMID: 12376345). We have added this to the discussion to the paper.

4. Can the authors speculate on what specific cellular process becomes ATP-starved? Is caspase activation very demanding in cells (on a molar level) with respect to ATP consumption? "Cellular processes" are very demanding but this is quite general.

There is an excellent review that summarizes that ATP drives NLRP3 inflammasome activation (PMID: 33036374). Our model is that ETC inhibitors deplete mitochondrial ATP and PCr and activate glycolysis during LPS priming. The addition of ETC inhibitors or cyclocreatine limits ATP to levels that are not able to sustain NLRP3 inflammasome activation. Importantly, elevated glycolysis even under ETC inhibitors is not sufficient to maintain NLRP3 inflammasome activation.

5. Mitochondria can also impact negative potential regulators of the inflammasome such as NO? Was this axis of regulation impacted at all?

LPS can stimulate iNOS dependent NO that could decrease oxygen consumption rate by competitively inhibiting cytochrome c oxidase. However, previously we demonstrated that NO under normoxia does not effectively inhibit cytochrome c oxidase (PMID: 11861645).

Referee #3 (Remarks to the Author):

The manuscript by Billingham et al. reports that maintenance of the cytoplasmic ATP pool is required to sustain activation of the NLRP3 inflammasome. They discovered that inhibitors of electron transport chain (ETC) complexes inhibited the NLRP3 inflammasome, and this inhibition was rescued by expression of *Saccharomyces cerevisiae* NADH dehydrogenase (NDI1) or *Ciona intestinalis* alternative oxidase (AOX). Metabolomics analysis revealed that the amount of phosphocreatine in cells was depleted by inhibition of ETC complexes. Overall, the authors claim that ATP generated by the OxPhos pathway is required for activation of the NLRP3 inflammasome and that phosphocreatine is an alternative source of ATP for the process. While the study has potentially identified a new link between phosphocreatinine and NLRP3 inflammasome activation, the experimental design is not sufficient to support their conclusions, and the study fails to account for previously established biochemical mechanisms.

We thank the reviewer for instructive comments and have addressed the major concerns

Major concerns:

1. Some of the data in this paper contradict previously published findings. 1a. Mills et al, Cell 2016, showed that DMM decreases LPS-induced Il1b expression, while here the authors find that DMM did not decrease this expression and had no effect on inflammasome priming (Extended Figure 2a).

Mills et al. paper measured LPS induction of IL1b mRNA expression at 24-48 hours. We measured IL1b mRNA expression after 4 hours of LPS stimulation. Inflammasome activation usually requires a few hours of priming followed by 30 minutes of ATP administration. It is likely that long-term inhibition of ETC prevents priming i.e., IL-1b mRNA expression. It is not clear why Mills et al used such long duration with LPS rather than the conventional duration of administering LPS for a few hours. Furthermore, Mills et al. only looked at LPS priming without providing signal 2 to activate NLRP3 inflammasome.

1b. R837 and CL097 (Groß et al, Immunity 2016) have been shown to inhibit the OxPhos pathway but activate the NLRP3 inflammasome.

We spent a considerable amount of our efforts understanding whether CL097 requires mitochondrial complex I for NLRP3 activation, see figure 5 as well as point 5 of this rebuttal on page 2. Our genetic data demonstrate that CL097 requires mitochondrial complex I through a ROS independent manner. We have shared our data and NDI1 mice with Professor Olaf Groß to further study this mechanism.

1c. Similarly, FCCP, another drug targeting the OxPhos pathway, increases IL-1. secretion (Jabaut et al, Free Radic Biol Med 2013). If the mechanism proposed in this study is correct, these inhibitors should be inhibiting the NLRP3 inflammasome.

Indeed, our current data demonstrates that FCCP administration inhibits NLRP3 inflammasome activation (Extended Fig. 8).

2. Whether the ETC inhibitors used here are blocking NLRP3 activation rather than priming is not convincingly established. Although the authors use several different ETC inhibitors, previous studies have shown that these inhibitors can suppress the priming signal for the NLRP3 inflammasome. For example, DMM has been shown to decrease LPS-induced Il1b expression (Mills et al, Cell 2016). Furthermore, DMM inhibits Ikb degradation during LPS stimulation (Zmijewski et al, Am J Physiol Lung Cell Mol Physiol 2009), further suggesting that DMM affects NFkB signaling and would block NLRP3 inflammasome priming.

The authors checked for the mRNA expression of Il1b to determine whether there was a priming defect (Extended Figure 2a, DMM; Extended Figure 4a, piericidin A; Extended Figure 6b, myxothiazol), but additional analyses are needed to confirm that there is no priming defect here. The authors should evaluate Nlrp3 and Il1b expression as well as protein expression levels for NLRP3, IL-1., and the NF-kB signaling pathway over a time course to provide confirmation. Additionally, oligomycin has a strong inhibitory effect on Il1b expression (Mills et al, Cell 2016), but the authors have not evaluated this.

The previous studies evaluated effects of DMM at 24 hrs. (Mills et al. Cell 2016) after LPS stimulation while we assessed mRNA expression at 4 hrs. after LPS stimulation. But the reviewer made an excellent suggestion to examine the protein levels of key components of NLRP3 inflammasome. Indeed, we measured key components of NLRP3 inflammasome (Fig. 2f, g and h). They did not change in presence of ETC inhibitor. Furthermore, oligomycin did not decrease IL-1b mRNA or pro-IL-1b protein levels (Extended Fig. 8b and c).

3. It is not clear why exogenously provided ATP is not sufficient to overcome the mitochondrial ATP to activate NLRP3 (Figure 4b and 4c), as ATP can cross the cell membrane.

ATP and phosphometabolites do not cross the cell membrane. Extracellular ATP stimulates the cell membrane ionotropic P2X7R receptor to cause K⁺ efflux.

4. The authors need to genetically validate the role of creatinine kinase in NLRP3 inflammasome activation.

BMDMs are notoriously difficult to knockdown genes of interest. Thus, we can use nanoparticle delivered RNAi against cytosolic creatinine kinase enzyme (CKB) in primary BMDMs to address this concern. A nanoparticle using scrambled RNAi was used as a control. Our results indicate that CKB RNAi decreases intracellular caspase-1 cleavage and secreted IL-1b (Fig.4g, h; Extended Fig. 10).

5. The authors should check the effect of ETC inhibition and cyclocreatine-mediated inhibition on other inflammasomes such as AIM2 and NLRC4 to verify that this is specific to the NLRP3 inflammasome and is not directly affecting another core component of the inflammasome.

Our careful reading of the literature indicates each inflammasome has a complicated biological mechanism and we feel this is beyond the scope of our current study. It would also distract to the general audience that is interested in NLRP3 dependent inflammation and metabolism. Furthermore, we put our maximal efforts in understanding CL097 mechanism as it relates to ETC inhibition.

6. What happens to expression of core components of the NLRP3 inflammasome (NLRP3, ASC, CASP1) with cyclocreatine treatment? Since translation is the most energy intensive process in the cell, and cyclocreatine treatment for one hour depletes ATP (Figure 5b), it is possible that the observed phenotype is due to a decrease in the amounts of these proteins.

This was an excellent suggestion! We measured these core components. Our results indicate ETC inhibitors don't change these core components (see Fig. 2f, g and h).

7. Aspects of this study lack novelty. For instance, the requirement of ATP hydrolysis for NLRP3 inflammasome activation is already well established.

This is a bit unfair. There have been multiple reports in the literature that demonstrate that ETC inhibitors can increase or decrease NLRP3 inflammasome activation. Many of these findings contradict each other primarily because they don't show the specificity of the compounds used to inhibit ETC function. We have used novel reagents (NDI1, QPC and AOX mice) to show the specificity of ETC inhibitors in primary BMDMs along with unbiased metabolomics that allowed us to conclude that it is mitochondrial ATP sustained by PCr that is necessary for inflammasome activation. Importantly, LPS stimulated macrophages are thought to be primarily glycolytic. In our model, ETC inhibition is not compensated by glycolysis. Furthermore, CL097 experiments decoded the necessity of mitochondrial complex I inhibition independent of ROS production for NLRP3 inflammasome activation. See pages 1 and 2 that highlights how we have clarified the current model in the field.

Minor concerns:

1. The time point used in this study is sometimes confusing. Some figures show 4 h LPS stimulation, while some show 6 h. Is there an explanation for this?

We use 4-hour time point to assess gene expression and metabolism changes prior to inflammasome activation. After, 5.5 hours of LPS priming we add ATP. Subsequently, we assess cleaved caspase-1 cell lysate at 10 minutes post ATP and IL-1b in the supernatant 30 minutes post ATP.

2. The addition of western blots to show the protein expression would greatly strengthen the manuscript, particularly to show pro-and cleaved caspase-1.

Caspase 1 and pro-caspase 1 protein expression data is included in the manuscript (Fig. 2d and e). This data was generated by performing a Simple Western with Wes, as opposed to a traditional Western blot. The Wes by ProteinSimple conducts automated Western blots in the absence of a gel or a blot. This new technology automates the entire Western blotting process including the separation of proteins, antibody incubation, signal detection, and data analysis. Thus, the Wes can be used as an analytical tool that provides quantifiable data. Unlike a traditional Western blot, a Wes uses probes or capillaries to load the sample, separation/stacking matrix, and antibodies from a specialized plate. The proteins within the probe are separated by molecular weight when a voltage is applied and immobilized following ultraviolet light. A chemiluminescent substrate is added after incubation with the primary and secondary antibodies. A charge-coupled device (CCD) camera records the light given off by the chemiluminescent reaction in a series of images over time, with fresh substrate added before each exposure. The Compass software then analyzes the Wes data and presents the results as an electropherogram (charge-based analysis) and in band-view (size-based analysis where bands are displayed in each lane similar to a traditional Western). Because the size-based analysis/band-view is a computer-generated image and not quantitative, we decided to report the quantifiable data from the Wes; that is, for each sample, the peak area of our protein of interest (i.e. caspase 1 or pro-caspase 1) divided by the peak area of our loading control vinculin. We presented this charge-based analysis data as a bar graph. We can add this detail in methods section.

3. The authors state that “NDI1-expressing mice do not have altered IL-1. protein in vivo 2 hours post LPS administration (Fig. 2a) or at baseline (data not shown), indicating that expression of NDI1 itself is not inflammatory.” However, the evidence is not strong enough to support this conclusion. The metabolomics data in Figure 1f showed that NDI1 expression changed the basal expression of some metabolites such as NAD and UDP-D-glucose, suggesting some pathways are affected by the NDI1 expression. To strengthen the conclusion here, analysis of the basal expression of some representative inflammatory cytokines such as IL-6, TNF, and IFN-. is necessary.

We have TNF cytokine data in vivo already. It does not change due to NDI expression in vivo. We did not include this data because we want to exclusively focus on one aspect of inflammation, NLRP3 inflammasome driven. We will reword sentence to appropriately reflect this.

4. Electrons from FADH₂ can enter the ETC at complex II. Therefore, inhibition of ETC complex I would not affect FADH₂ electron transport. It would be helpful to measure the mitochondrial membrane potential with piericidin A treatment.

We have measured mitochondrial membrane potential after piericidin A and ETC inhibitors.

5. In Extended Figure 5, the untreated NDI1 and WT cells should be included as controls. **We provide this RNAseq. data (Extended Fig. 5).**

6. Figure 3g lacks a description for the dots and squares in the graph.

We can add a description for the dots and squares in the figure legends (new Figure 3h).

7. The amount of nigericin used in this study (20 mM) is excessive. Do authors expect similar results with more conventional concentrations of 2 – 20 μ M?

That is a typo. It is meant to be 20 μ M.

Referee #4 (Remarks to the Author):

This manuscript investigates the interesting and timely question of whether and how mitochondria support NLRP3 inflammasome signalling. Several studies over a decade have implicated mitochondrial functions and dysfunctions (e.g. mitoROS, cytosolic release of mitoDNA) in mediating NLRP3 activation. Here, the authors use what appear to be elegant genetic approaches, coupled with pharmacological interventions to tease apart specific functions for mitochondria in NLRP3 activation. They conclude that mitochondrial dysfunction (e.g. ROS, mitochondrial damage leading to release of DAMPs including DNA) is not required for NLRP3 activation, and rather, they assert that the sole function of mitochondria during NLRP3 signalling is to produce ATP (hydrolysis of which is a well-known requirement for NLRP3 signalling). While approaches employed in parts of this study are sophisticated (e.g. mitochondrial perturbations) others are crude (e.g. measures of NLRP3 activation/activity), and the take-home message of the study is not novel (i.e. NLRP3 requires ATP for signaling; phosphocreatine supplies the

cell with ATP). With substantial revision, the rigour of the study may be improved to support the author's conclusions that mitochondrial dysfunction is not required for NLRP3 signalling; however, it is this Reviewer's opinion that such a finding would be more appropriately reported in a more specialist journal.

The reviewer finds mitochondrial ATP argument not novel. The dogma in the field is that LPS stimulated macrophages are glycolytic, yet this pathway cannot compensate for ETC inhibition. Also, there are so many conflicting reports on ETC inhibitors and NLRP3 inflammasome activation based on using pharmacologic interventions. Our combination of specific pharmacologic ETC inhibitors with genetics ablating ETC clearly demonstrates that wide variety of ETC inhibitors can decrease NLRP3 inflammasome activation through ATP production in response to extracellular ATP (K⁺ efflux dependent). Furthermore, we thank the reviewer for helping us focus on CL097 and ETC inhibition. These experiments further clarify the role of ETC in response to stimuli like CL097 that don't use K⁺ efflux as the primary mechanism.

Major concerns

1. The authors are experts in mitochondrial function and these aspects of the study appear to be elegantly and appropriately performed. The inflammasome aspects of the study, however, lack expertise and depth; this makes interpretation of the data very difficult. Throughout the study, the authors use two assays to measure NLRP3 inflammasome signalling: (1) they quantify IL-1b released into the cell culture medium, and (2) they measure 'active caspase-1' through what appears to be a western blot assay followed by quantification (although this is not very clear, and blots are not supplied). There are several issues with these assays as they are applied here. IL1b ELISAs are not a direct measure of NLRP3 signalling – these assays are not very specific for the cleaved (p17) form of this cytokine that is generated by caspase-1, and can detect pro-IL-1b that is released during cell death (e.g. induced by inflammasome activators or mitochondrial poisons). As a very downstream measure of inflammasome signalling, IL-1b release requires several pathways (e.g. trafficking to the plasma membrane, exit via GSDMD pores) that may be altered by experimental mitochondrial perturbations such as the authors employ. For the caspase-1 western blots, it is unclear which band the authors have quantified – they state it is the "active form". p20 is the most abundant cleavage product, so I assume this was what was measured – it is a common misconception is that this is active caspase-1 (actually it is a p33/p10 tetrameric species). What is even more strange is that the authors appear to have quantified caspase-1 fragments in cell extracts instead of cell supernatants – it is very well established that p20 and p10 caspase-1 fragments are released by the cell almost immediately after they are generated by the inflammasome, when the cell ruptures. These fragments are usually undetectable in cell extracts. Cell death (by mitochondrial poisons or inflammasome agonists) is not measured anywhere in the manuscript, and can be a major confounder of both of the outputs measured here. Instead of using (very downstream) outputs such as IL-1b or caspase-1 cleavage to report on NLRP3

activation status, for a journal such as Nature, this Reviewer would expect more direct outputs to have been measured, for example (1) BRET-based measurements of NLRP3 structure (e.g. methods pioneered by the Pelegrin lab) that allow direct measurement of NLRP3 in its closed, inactive state versus open, active state; or (2) microscopic quantification of NLRP3-induced ASC specks. Either of these methods would allow NLRP3 activity to be measured directly, rather than relying on signalling events very downstream in the cascade, that may be unintentionally altered by mitochondrial perturbations. Suggest using either of these approaches in an experimental system where caspase-1 activity is disabled (e.g. by genetic ablation or the caspase-1 inhibitor VX765) to prevent confounding effects of caspase-1 driven pyroptosis on inflammasome and mitochondrial outputs.

The goal of the paper was to simply assess whether ETC is necessary for NLRP3 inflammasome activation. We respectfully disagree with the use of other methods including BRET based NLRP3 activation assays would bolster the conclusion. However, the reviewer is correct that the dominant species of active caspase-1 dimers elicited by inflammasomes are in fact full-length p46 and a transient species, p33/p10 and the final terminated product p20/p10 fragment. We will rephrase caspase-1 data to indicate as intracellular cleaved caspase-1 (p20 fragment). Throughout the paper, we measured intracellular caspase-1 (p20 fragment) in cell lysate 10 minutes after extracellular ATP stimulation. The secreted IL-1 β in the supernatant was measured at 30 minutes. Thus, IL-1 β we are measuring is due to downstream effects of inflammasome activation.

Protein levels have been assessed by performing a Simple Western with Wes, as opposed to a traditional Western blot. The Wes by ProteinSimple conducts automated Western blots in the absence of a gel or a blot. This new technology automates the entire Western blotting process including the separation of proteins, antibody incubation, signal detection, and data analysis. Thus, the Wes can be used as an analytical tool that provides quantifiable data. Unlike a traditional Western blot, a Wes uses probes or capillaries to load the sample, separation/stacking matrix, and antibodies from a specialized plate. The proteins within the probe are separated by molecular weight when a voltage is applied and immobilized following ultraviolet light. A chemiluminescent substrate is added after incubation with the primary and secondary antibodies. A charge-coupled device (CCD) camera records the light given off by the chemiluminescent reaction in a series of images over time, with fresh substrate added before each exposure. The Compass software then analyzes the Wes data and presents the results as an electropherogram (charge-based analysis) and in band-view (size-based analysis where bands are displayed in each lane similar to a traditional Western). Because the size-based analysis/band-view is a computer-generated image and not quantitative, we decided to report the quantifiable data from the Wes; that is, for each sample, the peak area of our protein of interest (i.e. caspase 1 or pro-caspase 1) divided by the peak area of our loading control vinculin. We presented this charge-based analysis data as a bar graph. We can add this detail in methods section.

2. Related to point 1 -NLRP3 signalling requires both priming and triggering signals (in this study, signal 1 is LPS while signal 2 is usually ATP). The authors acknowledge that LPS upregulates pro-IL-1b levels, but they fail to acknowledge that LPS employs multiple mechanisms to poise the cell for NLRP3 activation by signal 2. For example, LPS upregulates NLRP3 mRNA and protein levels, as well as triggering a myriad of posttranslational modifications (e.g. to NLRP3) that 'ready' NLRP3 for activation. While the authors do check that mitochondrial perturbations do not affect pro-IL-1b induction, throughout the manuscript they fail to check whether these affect NLRP3 protein levels or other priming functions. Further, mitochondrial inhibitors are applied 1-3 h BEFORE adding LPS, which is then applied for 3-6 h prior to inflammasome activators (e.g. ATP). This experimental design means that mitochondrial inhibitors are extremely likely to affect LPS signalling and inflammasome priming. When studying signal 2 of inflammasome activation is important to add inhibitors AFTER priming (e.g. in the final 30 mins or so before the signal 2 stimulus) to minimise the likelihood of unintended effects on LPS signalling and inflammasome priming.

We measured key components of NLRP3 inflammasome (Fig. 2f-h), which did not change due to ETC inhibition.

3. Related to point 1 – of all the studies that implicate mitochondrial dysfunction and mitoROS in activating NLRP3, the strongest study is Reference 10 cited by the authors (Gross, Immunity, 2016). In that study, imiquimod (R837) was found to function as a mitochondrial poison, which activates NLRP3 through an unusual mechanism that does not require K⁺ efflux. Why did the authors not employ R837 as a key inflammasome stimulus throughout their study? This is the primary agonist that activates NLRP3 through mitochondrial perturbation, so its omission in the current manuscript is extremely surprising (and alarming). One cannot make blanket conclusions about mitochondrial dysfunction being unimportant for NLRP3 activation if imiquimod remains untested.

This was an excellent suggestion. We spent much of our time testing the necessity of mitochondrial ETC inhibition for CL097 activation of NLRP3 inflammasome. Please see pages 1 and 2 for an extensive discussion of our findings. We have shared our findings with Professor Groß (Immunity 2016) and our NDI1 mice to further examine how CL097 can activate NLRP3 inflammasome.

4. It is unclear to this Reviewer whether the experimental mitochondrial perturbations (genetic, pharmacological) made in this study have general effects on cell functions that support agonist sensing or inflammasome signalling. It would have been reassuring to see that mitochondrial perturbations did not affect signalling by an ATP-independent inflammasome (e.g. AIM2). Since the NAIP/NLRC4

inflammasome harbors ATP-ase activity, one would presume that this inflammasome would require mitochondrial ATP for activation, similar to NLRP3? Even more interesting is the question of whether the NLRP1 inflammasome requires mitochondrial ATP to enable NLRP1 oligomerisation prior to NLRP1 self-cleavage and release of the C-terminal CARD for signalling? Agonists for other inflammasomes (e.g. transfected dsDNA or flagellin) should have been employed throughout the study as important controls.

Our careful reading of the literature indicates each inflammasome has a complicated biological mechanism and we feel this is beyond the scope of our current study. It would also distract to the general audience that is interested in NLRP3 dependent inflammation and metabolism. Furthermore, we put our efforts in understanding CL097 mechanism as it relates to ETC inhibition.

5. There are several data replication issues. Figure captions do not clearly state the level of replication of each data set, often just referring to “n=(a number)”. Please reword to indicate the number of independent biological replicates (and not technical replicates). Where statistics were performed, were they performed on biological replicates (as they should be) and NOT technical (in-plate) replicates? For many in vitro figures, the number of data points within a panel differs widely between conditions in the same data set – this suggests that these conditions were not performed side by side in each experiment, which would substantially undermine the rigour of these experiments. Extended fig 1b legend indicates the untreated sample is N=1 while DMM is N=6; then why is there 11 data points for untreated, and how can one do statistics on one biological replicate?

They are all biological replicates, and we will explicitly state this for clarity. Also, Extended fig 1b legend is a typo. It is N=11 not N=1.

6. Important controls (e.g. unstimulated, LPS alone, mitochondrial inhibitor alone or in combination with LPS) are lacking in many figures, making data difficult to interpret. This is true of both in vitro and in vivo data.

ETC inhibitors cannot serve as a second signal to induce IL-1b secretion during LPS stimulation (Fig. 5g). Cyclocreatine does not induce IL-1b in the serum *in vivo*. In the absence of a second signal in vitro (ATP, Nigericin, CL097), does not activate the NLRP3 inflammasome and thus IL-1b is not processed or released from the cells. See graphs below. Similarly, we do not detect caspase-1 cleavage in the absence of LPS or second signal; these controls have been added throughout the paper where applicable for clarity.

In vitro supernatant IL-1b

In vivo serum IL-1b

- The LPS priming time differs widely between experiments (from 3 to 6 h) for no apparent reason; all experiments should have used a standard priming time. 4 h LPS is a standard priming length in the field.

We use 4-hour time point to assess gene expression and metabolism changes prior to inflammasome activation. After, 5.5 hours of LPS priming we add ATP. Subsequently, 10 minutes later we assess cleaved caspase-1 and at 30 minutes IL-1b in the supernatant. This has been added to figure legends and methods section for clarification

- Some experimental data has such an enormous amount of experimental variation (e.g. Extended figure 4c and 4f) that conclusions really cannot be made. For example, Ext Fig 4F looks like 3 experiments showed no difference while 3 experiments did show a difference.

We have used appropriate statistics to make appropriate conclusions. We added further experimental (increasing N) data to bolster conclusions.

9. Why does Piericidin A decrease H2O2 production in ND1 cells in Fig 1? The authors stated that ND1 cells were used as they are resistant to Piericidin A and cannot make ROS?

NDI1 by itself cannot generate ROS. By contrast, mitochondrial complex I can generate ROS. Our BMDMs harboring both NDI1 and a functional mitochondrial complex I can generate ROS through endogenous complex. In the presence of piericidin, endogenous complex I is inhibited thus cannot not generate ROS or regenerate NAD+. NDI1 plus piericidin does not allow endogenous complex I to generate ROS but still allows for NAD+ regeneration.

Minor points

10. Related to point 1, there are many statements about inflammasome biology that are incorrect or outdated. For example, two sentences in the abstract: -“as well as increasing pro-IL-1b expression at the mRNA level, NLRP3 oligomerisation leads to the cleavage of caspase-1 ...” suggests that NLRP3 oligomerisation induces pro-IL-1b mRNA, which is incorrect (pro-IL-1b mRNA is upregulated by signal 1 stimuli such as LPS).

We will correct this awkward sentence.

-“NLRP3 inflammasome activation occurs in proximity to mitochondria-associated membranes” – this is a very outdated view. More recent articles showing NLRP3 is activated at the dTGN are more generally accepted in the field.

Thank you for suggesting this. We will correct this mistake.

11. In Figure 5b it seems that the first data point has been normalised to 1 (ie. this condition has no variance), and yet standard t-tests (which assume equal variance between samples) have been used to compare this condition to others. A one sample t-test would have been more appropriate.

We have performed a one-sample t-test for comparisons to the first data group.

12. Extended figure 1d units is misspelt “unites”

We will correct this mistake.

13. Extended figure 3b looks like it must have been cropped off accidentally at the top

Hmm. We don’t see this in our version. But we will make sure all figures are visible.

14. Why does the concentration of piericidin A vary so much between experiments (100 to 500 nM)?

Initially, we tried various concentrations of piericidin A to see which concentration is effective in decreasing OCR (Figure 1C). But the key is that we rescue the effects of piericidin at 100 or 500 nM with NDI1 expression.

15. NLRP3 mis-spelt “NRLP3” in several places

We will correct this mistake.

16. For all heat map data – what does the scale mean? Is it fold induction/repression, linear/log scale, etc?

We have used well accepted convention to represent heatmaps. It is a linear distribution of induction or repression of gene expression or metabolites based on the z-score.

17. Extended figure 4 legend seems to be scrambled compared to figure – c caption seems to apply to 4d, d caption seems to apply to 4e, e caption seems to apply to 4c.

We will correct this mistake and clarify captions.

18. Extended figure 5 seems to be missing an analysis of important controls: WT vs NDI1 unstimulated, and inhibitor only

We provide this RNAseq. data (Extended Fig. 5).

19. Page 9 Inflammasome activation section in methods statement “BMDM were treated with LPS to activate the inflammasome” cannot be true, as LPS primes for (but does not activate) the NLRP3 inflammasome.

We apologize that we forgot to add ATP or nigericin in the sentence. Yes, we know that the LPS is only priming.

20. Page 9 Inflammasome activation section in methods – the authors appear to not appreciate that inflammasome agonists trigger inflammasome-mediated cell death. They indicate that they use a BCA assay to normalise for protein in cell lysates. This does not make sense for an experimental system that induces cell death in some (inflammasome-signalling) samples but not (inflammasome-resting) others

We apologize for not providing the details regarding measurement of cleaved caspase-1. Protein lysate was collected rapidly after ATP treatment (10 minutes) for activation, while majority of cells were still adhered to the plate and alive. We don't detect LDH release at this point. Quantification of protein concentrations demonstrated consistent concentrations across all treatment samples. The same amount and concentration of all samples was loaded in each independent experiment. To validate this, peak areas of vinculin and pro-caspase1 were consistent across samples in majority of independent experiments.

Decision Letter, first revision:

Subject: Decision on Nature Immunology submission NI-LE31928A

Message: 18th Jan 2022

Dear Professor Chandel,

Your Letter, "Mitochondrial electron transport chain is necessary for NLRP3 inflammasome activation" has now been seen by 3 referees. Reviewer 2 (metabolism) and 3

(inflammosomes) are the original reviewers from Nature. Unfortunately, we could not get reviewer 1 or reviewer 4 to re-review the paper for us in the time frame required, so instead we recruited a new reviewer (#5 inflammosomes and metabolism) to comment on your response to those reviewers.

You will see from their comments copied below that the inflammosme reviewers continue to have problems with the inflammasome data. Although they recognize that progress has been made, they maintain that further data are required as controls and to provide clearer mechanistic insight. In light of these comments, we cannot accept the manuscript for publication, but would be very interested in considering a revised version that addresses these serious concerns.

We hope you will find the referees' comments useful as you decide how to proceed. If you wish to submit a substantially revised manuscript, please bear in mind that we will be reluctant to approach the referees again in the absence of major revisions.

We agree with the reviewer's comments and would expect all reviewer comments to be addressed experimentally where requested, including to look at additional inflammasomes as a control and at CK as requested by the new reviewer 5

If you choose to revise your manuscript taking into account all reviewer and editor comments, please highlight all changes in the manuscript text file

* If you have not done so already please begin to revise your manuscript so that it conforms to our Letter format instructions at <http://www.nature.com/ni/authors/index.html>. Refer also to any guidelines provided in this letter.

The Reporting Summary can be found here:
<https://www.nature.com/documents/nr-reporting-summary.pdf>

-- that unprocessed scans are clearly labelled and match the gels and western blots

presented in figures.

-- that control panels for gels and western blots are appropriately described as loading on sample processing controls

-- all images in the paper are checked for duplication of panels and for splicing of gel lanes.

[REDACTED]

If you wish to submit a suitably revised manuscript we would hope to receive it within 6 months. If you cannot send it within this time, please let us know. We will be happy to consider your revision so long as nothing similar has been accepted for publication at Nature Immunology or published elsewhere.

Nature Immunology is committed to improving transparency in authorship. As part of our efforts in this direction, we are now requesting that all authors identified as 'corresponding author' on published papers create and link their Open Researcher and Contributor Identifier (ORCID) with their account on the Manuscript Tracking System (MTS), prior to acceptance. ORCID helps the scientific community achieve unambiguous attribution of all scholarly contributions. You can create and link your ORCID from the home page of the MTS by clicking on 'Modify my Springer Nature account'. For more information please visit www.springernature.com/orcid.

Thank you for the opportunity to review your work.

Sincerely,

Nick Bernard, PhD
Senior Editor
Nature Immunology

Reviewers' Comments:

Reviewer #2:

Remarks to the Author:

The authors have addressed my concerns.

Reviewer #3:

Remarks to the Author:

The article by Billingham et al. investigates whether mitochondrial electron transport chain (ETC) complexes impact NLRP3 inflammasome activation. Genetic approaches and pharmacological interventions were used to conclude that it is not ROS produced by the mitochondria but rather ATP production that is important for NLRP3 inflammasome activation. While some of my original concerns were addressed in this revision, there are some major issues that remain.

Major comments:

1. In my initial review, I noted several instances where results obtained in this manuscript differ from the published literature. Contradictory results include 1) DMM experiments (Ext Fig 2), which the authors have addressed by pointing out methodological differences, 2) CL097's previously shown inhibition of the oxphos pathway, and 3) FCCP experiments (Ext Fig 8). The authors do little to address contradictions (2) and (3) with additional experimental evidence and offer no real convincing argument or experimental justification as to how their experimental method is different/superior and why their model/results are correct for these aspects of the study. This remains a significant issue with the interpretation of their results.

2. Also in my initial review, I suggested checking the effect of ETC inhibition and cyclocreatine-mediated inhibition on other inflammasomes to verify that this effect is specific to the NLRP3 inflammasome and not due to an effect on another core inflammasome component. The authors' response was that they wanted to focus singularly on NLRP3 inflammasome-driven inflammation. However, checking the effect of ETC and cyclocreatine-mediated inhibition on other inflammasomes could help determine whether these treatments have any effect on other common mechanisms for inflammasome activation. Additionally, measuring non-inflammasome related cytokine levels could strengthen the connection of NDI1 to priming of NLRP3 inflammasomes. I would argue that these additional aspects of investigation are essential and do contribute to the understanding of NLRP3 inflammasome-driven inflammation.

Minor comments:

1. Per my suggestion, the authors measured the expression of components of NLRP3 inflammasomes (Figs 2f-2h) and drew conclusions about ETC inhibitors. However, in Fig 2g, NLRP3 expression is similar in wild-type cells treated with and without LPS, which is not consistent with the expected phenotype.

2. In several in vitro experiments, there are large differences in samples sizes between analyses performed within the experiment. For example, in Ext Fig 8h (measuring pro-IL-1 β expression), n=5 for each treatment, but in Fig 8c (also measuring pro-IL-1 β expression), n=8 in each treatment. This raises a question of whether these experiments were performed side by side or using separate experimental set ups, which may have influenced the results.

3. The authors based much of the novelty of the manuscript on the utilization of NDI1, QPC and AOX mice. They should therefore include information about the number of backcrosses that have been done, as the phenotype observed could possibly be due to the genetic background of the mice.

Reviewer #5:

Remarks to the Author:

This is an intriguing paper which re-examines the role of mitochondrial ROS in Nlrp3 activation. The authors find no role for ROS but instead they find mitochondrial ATP is needed to drive cytosolic ATP production via Creatine Kinase. This is an interesting and important observation for the fields of Nlrp3 and immunometabolism. I have however the following issues that would need to be addressed.

1. The key finding (other than correcting the literature on the role of mitochondrial ROS) concerns Creatine Kinase. This should be in the title of the paper, as from the current title its not clear to the reader what has actually been discovered here.
2. As CK is therefore central here I would like to see more data to support its role, specifically the use of a small molecule inhibitor which should block NLRP3 activation.
3. I agree with Reviewer 4 who has requested more assays. They authors should really examine NLRP3-proximal signals here and Asc oligomerisation would be an obvious and straightforward assay to perform. Equally the suggestion of assaying AIM2 is a good one, as that is ATP - independent. These act as controls but also provide further evidence to support the key finding here.
4. What do the authors think is the actual role of ATP here? Is it to act as a substrate for NLRP3 or is it for a kinase or other process dependent on a kinase for the regulation of NLRP3? This to me is a key question here - the authors could speculate on this point but if they had experiments to address this, it would greatly strengthen the manuscript. This is why reviewer 4 was asking for a BRET-based assay as that can measure how ATP binding and hydrolysis affects NLRP3.

Author Rebuttal, first revision:

Reviewers' Comments:

Reviewer #2:

Remarks to the Author:

The authors have addressed my concerns.

Reviewer #3:

Remarks to the Author:

The article by Billingham et al. investigates whether mitochondrial electron transport chain (ETC) complexes impact NLRP3 inflammasome activation. Genetic approaches and pharmacological interventions were used to conclude that it is not ROS produced by the mitochondria but rather ATP production that is important for NLRP3 inflammasome activation. While some of my original concerns were addressed in this revision, there are some major issues that remain.

Major comments:

1. In my initial review, I noted several instances where results obtained in this manuscript differ from the published literature. Contradictory results include 1) DMM experiments (Ext Fig 2), which the authors have addressed by pointing out methodological differences, 2) CL097's previously shown inhibition of the oxphos pathway, and 3) FCCP experiments (Ext Fig 8). The authors do little to address contradictions (2) and (3) with additional experimental evidence and offer no real convincing argument or experimental justification as to how their experimental method is different/superior and why their model/results are correct for these aspects of the study. This remains a significant issue with the interpretation of their results.

CL097 works in a K⁺ efflux-independent manner while extracellular ATP or nigericin is dependent on K⁺ efflux to activate the NLRP3 inflammasome. Dr. Grob Immunity paper suggested that CL097 could potentially inhibit quinoneoxidoreductases like NQO2 or mitochondrial complex I to generate ROS. He did not provide any genetic evidence to support either mechanism. We have been in contact with Dr. Grob and shared our NDI1 mice. Our genetic data point to CL097 inhibiting complex I but not activating the NLRP3 inflammasome through ROS. As expected, he is thrilled that we genetically validated part of his model. He will be using our NDI1 mice to further probe the effects of CL097.

Original comment from the reviewer: "Similarly, FCCP, another drug targeting the OxPhos pathway, increases IL-1B secretion (Jabaut et al, Free Radic Biol Med 2013). If the mechanism proposed in this study is correct, these inhibitors should be inhibiting the NLRP3 inflammasome." We wrote that our data show the opposite. FCCP inhibited NLRP3 inflammasome. The one major difference is that their paper Figure 7 shows inflammasome activation in the presence of FCCP using serum amyloid A as a stimulus rather than LPS + nigericin, ATP or CL097. They stimulated for 24 hours. It is well known that FCCP can cause intracellular cytosolic acidification after long-term treatment¹ Our use of piercidin +/- NDI1 and myxothiazol +/- AOX are the best combination of genetic and pharmacologic experiments, justifying our protocol. Also, we use canonical widely utilized stimulus and time points in the field.

2. Also in my initial review, I suggested checking the effect of ETC inhibition and cyclocreatinemediated inhibition on other inflammasomes to verify that this effect is specific to the NLRP3 inflammasome and not due to an effect on another core inflammasome component. The authors' response was that they wanted to focus singularly on NLRP3 inflammasome-driven inflammation. However, checking the effect of ETC and cyclocreatine-mediated inhibition on other inflammasomes could help determine whether these treatments have any effect on other common mechanisms for inflammasome activation. Additionally, measuring non-inflammasome related cytokine levels could strengthen the connection of NDI1 to priming of NLRP3 inflammasomes. I would argue that these additional aspects of investigation are essential and do contribute to the understanding of NLRP3 inflammasome-driven inflammation.

We have stimulated the AIM2 inflammasome using poly (dA:Dt) with piericidin A and assessed caspase-1 cleavage. See figure on next page. Also, we have done TNF α secretion with ELISA with ETC inhibitors. See figures below.

Caspase-1 (p20 fragment) data in AIM2 stimulated BMDMs

Intracellular caspase-1 (p20 fragment) protein expression (peak area, normalized to vinculin peak area) in cell lysates from WT BMDMs treated for 4 hours with LPS (100ng/mL) prior to 2 hours of transfection with poly(dA:dT) (5 μ /mL) (N=5 for all treatments).

Intracellular pro-caspase-1 protein expression (peak area, normalized to vinculin peak area) in cell lysates from WT BMDMs treated for 4 hours with LPS (100ng/mL) prior to 2 hours of transfection with poly(dA:dT) (5 μ /mL) (N=5 for all treatments).

TNF α production with piericidin A or DMM in BMDMs

Minor comments:

1. Per my suggestion, the authors measured the expression of components of NLRP3 inflammasomes (Figs 2f-2h) and drew conclusions about ETC inhibitors. However, in Fig 2g, NLRP3 expression is similar in wild-type cells treated with and without LPS, which is not consistent with the expected phenotype.

This is a typo on NLRP3 and ASC. Old graph versus new graph is here:

2. In several in vitro experiments, there are large differences in samples sizes between analyses performed within the experiment. For example, in Ext Fig 8h (measuring pro-IL-1 β expression), n=5 for each treatment, but in Fig 8c (also measuring pro-IL-1 β expression), n=8 in each treatment. This raises a question of whether these experiments were performed side by side or using separate experimental set ups, which may have influenced the results.

These were all similar experimental setup (hrs.) with LPS priming. These are all independent biological replicates from different isolations of BMDMs. Importantly, they are NOT technical replicates.

3. The authors based much of the novelty of the manuscript on the utilization of NDI1, QPC and AOX mice. They should therefore include information about the number of backcrosses that have been done, as the phenotype observed could possibly be due to the genetic background of the mice.

These are all C67B6 mice with appropriate littermate controls and previously published strains. This has been clarified in the methods section.

Reviewer #5:

Remarks to the Author:

This is an intriguing paper which re-examines the role of mitochondrial ROS in Nlrp3 activation. The authors find no role for ROS but instead they find mitochondrial ATP is needed to drive cytosolic ATP production via Creatine Kinase. This is an interesting and important observation for the fields of Nlrp3 and immunometabolism. I have however the following issues that would need to be addressed.

1. The key finding (other than correcting the literature on the role of mitochondrial ROS) concerns Creatine Kinase. This should be in the title of the paper, as from the current title its not clear to the reader what has actually been discovered here.

It is important to mention that the history of mitochondrial role in and inflammasome activation is a quite contentious. The initial reports from late Professor Tschopp demonstrated mitochondrial complex I inhibitor rotenone, or complex III inhibitor antimycin, induced mtROS and was sufficient to activate the NLRP3 inflammasome (Nature 2010). Professor Gabriel Nunez pointed out that mitochondria are dispensable for NLRP3 inflammasome activation (Immunity 2013). Multiple studies link ROS to NLRP3 inflammasome activation. Furthermore, Professor Michael Karin using our TFAM floxed mice suggested mtDNA is a key input for NLRP3 inflammasome activation (Nature 2018). TFAM is necessary to maintain mtDNA levels. In the absence of TFAM, mtDNA is depleted. Our studies do not address or refute this mechanism. However, it is important to note that experimental strategies that deplete mtDNA like loss of TFAM also disable ETC function (MitoATP production) and thereby diminish mitochondrial ATP production. Thus, it is possible depletion of mtDNA by TFAM ablation impairs NLRP3 inflammasome activation in part due to diminished mitochondrial ATP mechanism. Presently, the use of TFAM floxed mice cannot distinguish between mtDNA or ATP as the key input. However, our interventions do not diminish mtDNA levels but inhibit ETC function. Thus, our genetic/pharmacological combination has clarified much of these past disparate findings and our current title is justified.

2. As CK is therefore central here I would like to see more data to support its role, specifically the use of a small molecule inhibitor which should block NLRP3 activation.

We have used RNAi against CK to complement cyclocreatine data as a tool to inhibit CK function.

3. I agree with Reviewer 4 who has requested more assays. They authors should really examine NLRP3-proximal signals here and Asc oligomerisation would be and obvious and straightforward assay to perform. Equally the suggestion of assaying AIM2 is a good one, as that is ATP - independent. These act as controls but also provide further evidence to support the key finding here.

See on the previous page that AIM2 dependent caspase-1 cleavage is not affected by complex I inhibitor Piericidin A. I am not sure what is gained from doing ASC

oligomerization assay. Our main goal was to answer a simple question using genetics. “Is mitochondrial ETC necessary for NLRP3 inflammasome activation” As a follow-up, whether ROS is the key signaling input. We have answered these two key questions. Moreover, as pointed out in the next comment, ATP hydrolysis is needed for NLRP3 inflammasome activation. We don’t think ASC oligomerization would provide any further insight into why ATP is necessary for NLRP3 inflammasome activation.

4. What do the authors think is the actual role of ATP here? Is it to act as a substrate for NLRP3 or is it for a kinase or other process dependent on a kinase for the regulation of NLRP3? This to me is a key question here - the authors could speculate on this point but if they had experiments to address this, it would greatly strengthen the manuscript. This is why reviewer 4 was asking for a BRET-based assay as that can measure how ATP **binding and hydrolysis affects NLRP3**.

As we wrote: “NLRP3 requires ATP hydrolysis for inflammasome activation^{2,3}. The widely used NLRP3 inhibitor MCC950 interacts with the Walker B motif within the NLRP3 NACHT domain to prevent ATP hydrolysis”. See references 2 and 3 below:

An important conclusion of our study is that glycolysis alone cannot support NLRP3 inflammasome activation. The dogma in the field is that LPS stimulates glycolysis (Warburg Effect) and this is the dominant metabolic pathway. We demonstrated that BMDMs treated with piericidin A during LPS stimulation have high levels of ECAR (glycolytic index) concomitant with reduced OCR. Thus, glycolysis in the absence of mitochondrial ATP is not sufficient to support NLRP3 inflammasome activation. Collectively, these data indicate that extracellular ATP activation of NLRP3 inflammasome depends on mitochondrially derived ATP, initially generated by forward respiratory electron flow, and supplied via the phosphocreatine shuttle.

References:

1. Berezhnov, A. V. *et al.* Intracellular pH Modulates Autophagy and Mitophagy. *J. Biol. Chem.* **291**, 8701–8708 (2016).
2. Duncan, J. A. *et al.* Cryopyrin/NALP3 binds ATP/dATP, is an ATPase, and requires ATP binding to mediate inflammatory signaling. *Proc. Natl. Acad. Sci. U. S. A.* **104**, 8041–8046 (2007).
3. Coll, R. C. *et al.* MCC950 directly targets the NLRP3 ATP-hydrolysis motif for inflammasome inhibition. *Nat. Chem. Biol.* **15**, 556–559 (2019).

Decision Letter, second revision:

Subject: Your manuscript, NI-A31928B

Message: Our ref: NI-A31928B

10th Feb 2022

Dear Dr. Chandel,

Thank you for your patience as we've prepared the guidelines for final submission of your Nature Immunology manuscript, "Mitochondrial electron transport chain is necessary for NLRP3 inflammasome activation" (NI-A31928B). Please carefully follow the step-by-step instructions provided in the attached file, and add a response in each row of the table to indicate the changes that you have made. Please also check and comment on any additional marked-up edits we have proposed within the text. Ensuring that each point is addressed will help to ensure that your revised manuscript can be swiftly handed over to our production team.

When you upload your final materials, please include a point-by-point response to any remaining reviewer comments and please make sure to upload your checklist.

In recognition of the time and expertise our reviewers provide to Nature Immunology's editorial process, we would like to formally acknowledge their contribution to the external peer review of your manuscript entitled "Mitochondrial electron transport chain is necessary for NLRP3 inflammasome activation". For those reviewers who give their assent, we will be publishing their names alongside the published article.

Nature Immunology offers a Transparent Peer Review option for new original research manuscripts submitted after December 1st, 2019. As part of this initiative, we encourage our authors to support increased transparency into the peer review process by agreeing to have the reviewer comments, author rebuttal letters, and editorial decision letters published as a Supplementary item. When you submit your final files please clearly state in your cover letter whether or not you would like to participate in this initiative. Please note that failure to state your preference will result in delays in accepting your manuscript for publication.

Cover suggestions

As you prepare your final files we encourage you to consider whether you have any images or illustrations that may be appropriate for use on the cover of Nature Immunology.

Nature Immunology has now transitioned to a unified Rights Collection system which will allow our Author Services team to quickly and easily collect the rights and permissions required to publish your work. Approximately 10 days after your paper is formally accepted, you will receive an email in providing you with a link to complete the grant of rights. If your paper is eligible for Open Access, our Author Services team will also be in touch regarding any additional information that may be required to arrange payment for your article.

Please note that *Nature Immunology* is a Transformative Journal (TJ). Authors may publish their research with us through the traditional subscription access route or make their paper immediately open access through payment of an article-processing charge (APC). Authors will not be required to make a final decision about access to their article until it has been accepted. [Find out more about Transformative Journals](https://www.springernature.com/gp/open-research/transformative-journals).

If you have any questions about costs, Open Access requirements, or our legal forms, please contact ASJournals@springernature.com.

Authors may need to take specific actions to achieve [compliance](https://www.springernature.com/gp/open-research/funding/policy-compliance-faqs) with funder and institutional open access mandates. For submissions from January 2021, if your research is supported by a funder that requires immediate open access (e.g. according to [Plan S principles](https://www.springernature.com/gp/open-research/plan-s-compliance)) then you should select the gold OA route, and we will direct you to the compliant route where possible. For authors selecting the subscription publication route our standard licensing terms will need to be accepted, including our [self-archiving policies](https://www.springernature.com/gp/open-research/policies/journal-policies). Those standard licensing terms will supersede any other terms that the author or any third party may assert apply to any version of the manuscript.

Please use the following link for uploading these materials: [REDACTED]

Best regards,

Elle Morris
Senior Editorial Assistant
Nature Immunology
Phone: 212 726 9207
Fax: 212 696 9752
E-mail: immunology@us.nature.com

On behalf of

Nick Bernard, PhD
Senior Editor
Nature Immunology

Reviewer #1:

Remarks to the Author:

Although the authors haven't carried out all the experiments I requested I am happy with the rebuttal and the experiment on AIM2. They speculate on what the ATP is needed for here which is to bind NLRP3 to allow it's activation which is reasonable, given what we know about NLRP3.

Reviewer #3:

Remarks to the Author:

The article by Billingham et al. investigates whether mitochondrial electron transport chain (ETC) complexes impact NLRP3 inflammasome activation. Genetic approaches and pharmacological interventions were used to conclude that it is not ROS produced by the mitochondria but rather ATP production that is important for NLRP3 inflammasome activation. The authors have addressed my minor comments, but my primary concern about the specificity of the mechanism remains.

Major comments:

1. The authors have explained some of the reasoning behind the contradictions between their study and previous literature in their point by point response. However, much of this information is still absent from the article. While I still remain unconvinced by the author's logic in explaining these contradiction, at a minimum these discussions should be included in the manuscript to provide readers with the appropriate context for interpreting these experimental results.

2. To address the specificity of the mechanism, the authors included quantification of

caspace-1 p20 fragment production in response to poly(dA:dT) stimulation as a measure of AIM2 inflammasome activation (new Ext. Fig. 12). However, the level of caspace-1 p20 production is extremely low in this experiment, at least 10-fold lower than in the NLRP3 inflammasome activation experiments (for example, Fig. 2D), suggesting the stimulation used is not optimal to activate the AIM2 inflammasome. Additionally, there appears to be a trend toward a decrease in p20 formation upon the addition of piericidin A, and with the overall low level of activation achieved in the positive control, it is difficult to conclusively rule out that there is no difference between these conditions and that the mechanism is specific to NLRP3. Significant additional work needs to be done to address the point of specificity.

Final Decision Letter:

Subject: Decision on Nature Immunology submission NI-A31928C

Message: In reply please quote: NI-A31928C

Dear Dr. Chandel,

I am delighted to accept your manuscript entitled "Mitochondrial electron transport chain is necessary for NLRP3 inflammasome activation" for publication in an upcoming issue of Nature Immunology.

Over the next few weeks, your paper will be copyedited to ensure that it conforms to Nature Immunology style. Once your paper is typeset, you will receive an email with a link to choose the appropriate publishing options for your paper and our Author Services team will be in touch regarding any additional information that may be required.

Please note that Nature Immunology is a Transformative Journal (TJ). Authors may publish their research with us through the traditional subscription access route or make their paper immediately open access through payment of an article-processing charge (APC). Authors will not be required to make a final decision about access to their

article until it has been accepted. [Find out more about Transformational Journals](https://www.springernature.com/gp/open-research/transformational-journals).

Your paper will be published online soon after we receive your corrections and will appear in print in the next available issue. Content is published online weekly on Mondays and Thursdays, and the embargo is set at 16:00 London time (GMT)/11:00 am US Eastern time (EST) on the day of publication. Now is the time to inform your Public Relations or Press Office about your paper, as they might be interested in promoting its publication. This will allow them time to prepare an accurate and satisfactory press release. Include your manuscript tracking number (NI-A31928C) and the name of the journal, which they will need when they contact our office.

About one week before your paper is published online, we shall be distributing a press release to news organizations worldwide, which may very well include details of your work. We are happy for your institution or funding agency to prepare its own press release, but it must mention the embargo date and Nature Immunology. Our Press Office will contact you closer to the time of publication, but if you or your Press Office have any enquiries in the meantime, please contact press@nature.com.

Also, if you have any spectacular or outstanding figures or graphics associated with your manuscript - though not necessarily included with your submission - we'd be delighted to consider them as candidates for our cover. Simply send an electronic version (accompanied by a hard copy) to us with a possible cover caption enclosed.

You can now use a single sign-on for all your accounts, view the status of all your manuscript submissions and reviews, access usage statistics for your published articles

and download a record of your refereeing activity for the Nature journals.

Please note that we encourage the authors to self-archive their manuscript (the accepted version before copy editing) in their institutional repository, and in their funders' archives, six months after publication. Nature Research recognizes the efforts of funding bodies to increase access of the research they fund, and strongly encourages authors to participate in such efforts. For information about our editorial policy, including license agreement and author copyright, please visit www.nature.com/ni/about/ed_policies/index.html

Sincerely,

Nick Bernard, PhD
Senior Editor
Nature Immunology